# Peroxisome-derived ether lipids regulate lysosomal exocytosis

Liang Chen [ID][1], Danielle Henn [ID][1], Zhongzheng Dong [ID][1], Jiaxuan Liang[1], Aleksander Wielenga[1], Goncalo Vale [ID][2,3], Bala Burugula [ID][4], Junyi Zou[5], Yamuna Krishnan [ID][5], Jeffrey G McDonald [ID][3,6], Jacob Kitzman [ID][4] & Ming Li [ID][1✉]

## Abstract

**Lysosomes and peroxisomes are essential for cellular homeostasis, yet how their activities are coordinated remains poorly understood. Here, we identify peroxisome-derived ether lipids as key regulators of lysosomal function. A genome-wide CRISPR/Cas9 screen in LYSET-deficient mucolipidosis V cells revealed that disruption of ether lipid synthesis genes or peroxins markedly reduces lysosome accumulation and restores degradative capacity. Genetic or pharmacological inhibition of ether lipid synthesis enhanced lysosomal exocytosis and promoted the clearance of undigested material independently of mannose-6-phosphate trafficking. Conversely, supplementation with the ether lipid precursor hexadecylglycerol increased lysosome abundance, while reducing their degradative capacity. These findings uncover a peroxisome-lysosome metabolic axis, in which ether lipids act as bidirectional regulators of lysosomal number and function independently of the lysosomal master regulator TFEB. Our findings reveal how peroxisome-localized lipid metabolism modulates lysosomal homeostasis, and suggest potential new strategies to combat lysosomal and peroxisomal disorders.**

**Subject Categories** Membranes & Trafficking; Metabolism; Organelles

## Introduction

The intricate interplay between cellular organelles is a hallmark of life's complexity, ensuring that diverse metabolic processes are seamlessly integrated. Among these organelles, lysosomes and peroxisomes are both indispensable for cellular homeostasis, yet they have traditionally been studied as distinct entities with specialized, non-overlapping functions. Lysosomes, often referred to as the cell's recycling centers, contain more than 60 hydrolytic enzymes that degrade and recycle macromolecules (Saftig and Klumperman, 2009; Settembre and Perera, 2023). These enzymes rely on precise trafficking through the mannose-6-phosphate (M6P) pathway (Coutinho et al, 2012; Ghosh et al, 2003), mediated by the Golgi-localized GlcNAc-1-phosphotransferase (GNPT) complex and its regulatory protein, LYSET/GCAF (Pechincha et al, 2022; Richards et al, 2022; Zhang et al, 2022). Disruptions in this pathway cause severe lysosomal storage disorders (LSDs), i.e., Mucolipidosis (ML) type II/III/V, characterized by profound developmental abnormalities, neurodegeneration, and organ dysfunction (Ain et al, 2021; Platt et al, 2012; Platt et al, 2018). Despite recent advances in therapeutic approaches, such as enzyme replacement and gene therapy, these disorders remain incurable, underscoring the need for innovative strategies to address lysosomal dysfunction.

Many LSDs, in which lysosomes are severely deficient in degradative capacity, exhibit elevated lysosome numbers and increased abundance of their resident proteins (Arevalo et al, 2022; He et al, 2022; Leal et al, 2022; Xu et al, 2014). While this upregulation is often interpreted as a compensatory mechanism to cope with lysosomal dysfunction, the precise molecular mechanisms driving this phenomenon remain poorly understood.

Peroxisomes are metabolic hubs critical for reactive oxygen species (ROS) detoxification, innate immunity, and lipid metabolism (Kim et al, 2025; Kumar et al, 2024; Schrader et al, 2016; Smith and Aitchison, 2013; Wanders and Waterham, 2006). In particular, they synthesize ether lipids, including plasmalogens, which play pivotal roles in membrane architecture and cellular signaling (Braverman and Moser, 2012). Ether lipids are synthesized through a multi-step pathway initiated in the peroxisome and completed in the endoplasmic reticulum (ER). They are known to stabilize membranes, protect against oxidative stress, and are implicated in neuroprotection (Braverman and Moser, 2012; Dean and Lodhi, 2018). Deficiencies in ether lipid synthesis lead to severe developmental syndromes like Rhizomelic Chondrodysplasia Punctata (RCDP) and are associated with neurodegenerative diseases (Braverman and Moser, 2012). Perturbations in ether lipid metabolism have been observed in several lysosomal and neurodegenerative disorders, including Gaucher disease, Sandhoff disease (Lecommandeur et al, 2020) and Parkinson's disease (Guedes et al,

[1]Department of Molecular, Cellular, and Developmental Biology, University of Michigan, Ann Arbor, MI 48109, USA. [2]Department of Internal Medicine, University of Texas Southwestern Medical Center, Dallas, TX 75390, USA. [3]Center for Human Nutrition, University of Texas Southwestern Medical Center, Dallas, TX 75390, USA. [4]Department of Human Genetics, University of Michigan Medical School, Ann Arbor, MI 48109, USA. [5]Department of Chemistry, University of Chicago, Chicago, IL 60637, USA. [6]Department of Molecular Genetics, University of Texas Southwestern Medical Center, Dallas, TX 75390, USA. ✉E-mail: mlium@umich.edu

2017; Lopez de Frutos et al, 2022). These findings suggest a potential but poorly understood connection between ether lipid homeostasis and lysosomal dysfunction.

In this study, we conducted a genome-wide screen to uncover pathways that could restore lysosomal function in cells lacking LYSET, a key regulator of the M6P pathway whose deficiency causes Mucolipidosis type V. Strikingly, three peroxisome genes involved in ether lipid synthesis, as well as over a dozen peroxins, emerged as critical regulators of lysosomal function. Disruption of ether lipid biosynthesis in MLV cells not only abolished disease-associated lysosomal accumulation but also partially restored lysosomal digestive capacity. Mechanistically, lowering the levels of ether lipids enhances lysosomal exocytosis, thereby reducing the toxic accumulation of undigested macromolecules. These findings reveal a previously unrecognized link between peroxisomal lipid biosynthesis and lysosome function, potentially opening new therapeutic avenues for treating LSDs, where strategies targeting ether lipid metabolism could enhance lysosomal function and cellular clearance.

# Results

## Lysosome accumulation in M6P-deficient cells occurs independently of mTORC1- and TFEB/TFE3-driven lysosomal biogenesis

Among the most severe LSDs are those caused by defects in the M6P pathway, which is critical for the proper trafficking of lysosomal hydrolases (Ghosh et al, 2003). ML Type II, III, and V, which result from disruptions in this pathway, are characterized by the loss of most luminal enzymes from the lysosome (Ghosh et al, 2003; Pechincha et al, 2022; Richards et al, 2022; Zhang et al, 2022). To better understand how M6P deficiency impacts lysosomal quantity and function, we generated LYSET-deficient (representing ML type V) (Pechincha et al, 2022; Richards et al, 2022; Zhang et al, 2022) and GNPTAB-deficient (representing ML type II/III) (Tiede et al, 2005; Velho et al, 2019) HEK293T cells and analyzed their lysosomal phenotypes. LAMP2 immunostaining and flow cytometry of LysoTracker red-stained cells revealed a striking increase in lysosome numbers in the knockout (KO) cells compared to WT (Fig. 1A–C). We further assessed lysosomal function in these KO cells using Dye Quenched (DQ)-BSA digestion assay (Frost et al, 2017). DQ-BSA is a derivative of bovine serum albumin conjugated to a self-quenched fluorophore. It is internalized through endocytosis and trafficked to lysosomes, where proteolytic cleavage relieves quenching and produces fluorescence. Thus, DQ-BSA effectively reports lysosomal degradative capacity. As expected, the KO cells exhibited a ~70% reduction in DQ-BSA fluorescence, since most luminal hydrolases are absent from lysosomes (Fig. 1D,E).

One prevailing hypothesis is that LSD-associated lysosome accumulation is driven by the inhibition of the mechanistic target of rapamycin complex (mTORC) in response to cellular stress due to lysosome dysfunction, leading to dephosphorylation and nuclear translocation of Transcription Factor EB (TFEB) and its homolog, TFE3, the master regulators of lysosomal biogenesis (Sardiello et al, 2009; Settembre et al, 2012). Upon dephosphorylation, TFEB/TFE3 translocate into the nucleus to activate the Coordinated Lysosomal Expression and Regulation (CLEAR) network, which upregulates

lysosomal and autophagy-related genes to stimulate lysosome biogenesis (Palmieri et al, 2011; Samie and Xu, 2014; Sardiello et al, 2009).

To test whether the increased lysosome abundance in LYSET-deficient cells is due to TFEB/TFE3 regulation, we generated TFEB/TFE3 double knockout (DKO) cells and subsequently deleted LYSET in this background (Fig. 1F). Surprisingly, LYSET deletion in DKO cells still resulted in increased lysosome abundance, as indicated by LysoTracker and LAMP2 staining, comparable to LYSET KO cells (Fig. 1G–J), suggesting a TFEB/TFE3-independent mechanism. Further, LYSET deletion did not affect TFEB subcellular localization in HeLa cells, arguing against mTORC1 inhibition in this context (Fig. EV1A). Similarly, reverse transcription quantitative PCR (RT-qPCR) analysis of CLEAR network gene expression showed no significant differences between LYSET KO and WT cells (Fig. EV1B), confirming that TFEB/TFE3 activity remained unchanged.

We next examined mTORC1 activity, which regulates TFEB nuclear translocation via phosphorylation. Phosphorylation levels of mTORC1 substrates S6K and 4EBP1 were slightly elevated in LYSET KO cells (Fig. EV1C,D), suggesting that mTORC1 activity was similar or slightly enhanced. Moreover, mTORC1 activity in LYSET KO cells appeared more resistant to EBSS starvation compared to WT cells (Fig. EV1C,D). Similar elevations in mTORC1 activity have been reported in other LSDs, such as Niemann Pick Type C (Davis et al, 2021) and Cystinosis (Berquez et al, 2023; Luciani and Devuyst, 2024).

In summary, our findings indicate that lysosomal accumulation in M6P-deficient cells is not due to mTORC1 inactivation or TFEB/TFE3-mediated lysosome biogenesis. This suggests an alternative, yet unidentified mechanism underlying the increased lysosome numbers observed in severe LSDs.

## A genome-wide CRISPR screen reveals peroxisome-mediated upregulation of lysosome numbers in M6P-deficient cells

Taking advantage of the ease of quantifying lysosome numbers in single cells by flow cytometry after LysoTracker staining, we designed a pooled genome-wide CRISPR/Cas9 knockout screen to identify genes and pathways that could suppress lysosomal accumulation in MLV cells. In this screen, mutants exhibiting low lysosome numbers (i.e., carrying a mutation that reverses the disease-associated lysosomal accumulation phenotype) are sorted via flow cytometry, and the underlying genes are identified through Illumina sequencing (Fig. 2A).

To achieve this, we transduced LYSET KO cells with Cas9 and the Brunello CRISPR library (Doench et al, 2016), creating a genome-wide KO library that targets 19,114 human genes. The cells were stained with LysoTracker Red, and the bottom 5% of the population, representing cells with fewer lysosomes, were sorted by fluorescence-activated cell sorting (FACS). The cells were allowed to recover and subjected to two additional rounds of sorting. After three rounds, lysosome levels in the sorted population were nearly restored to WT levels, as confirmed by both LysoTracker staining and LAMP2 immunofluorescence (Fig. 2B,C).

Illumina sequencing of sorted cells revealed 177 enriched genes (log2|FC| > 1, Dataset EV1). Both STRING (Franceschini et al, 2013; Szklarczyk et al, 2023) and Gene Ontology (GO)-term analysis (Gene Ontology, 2021; Zhou et al, 2019) of these genes

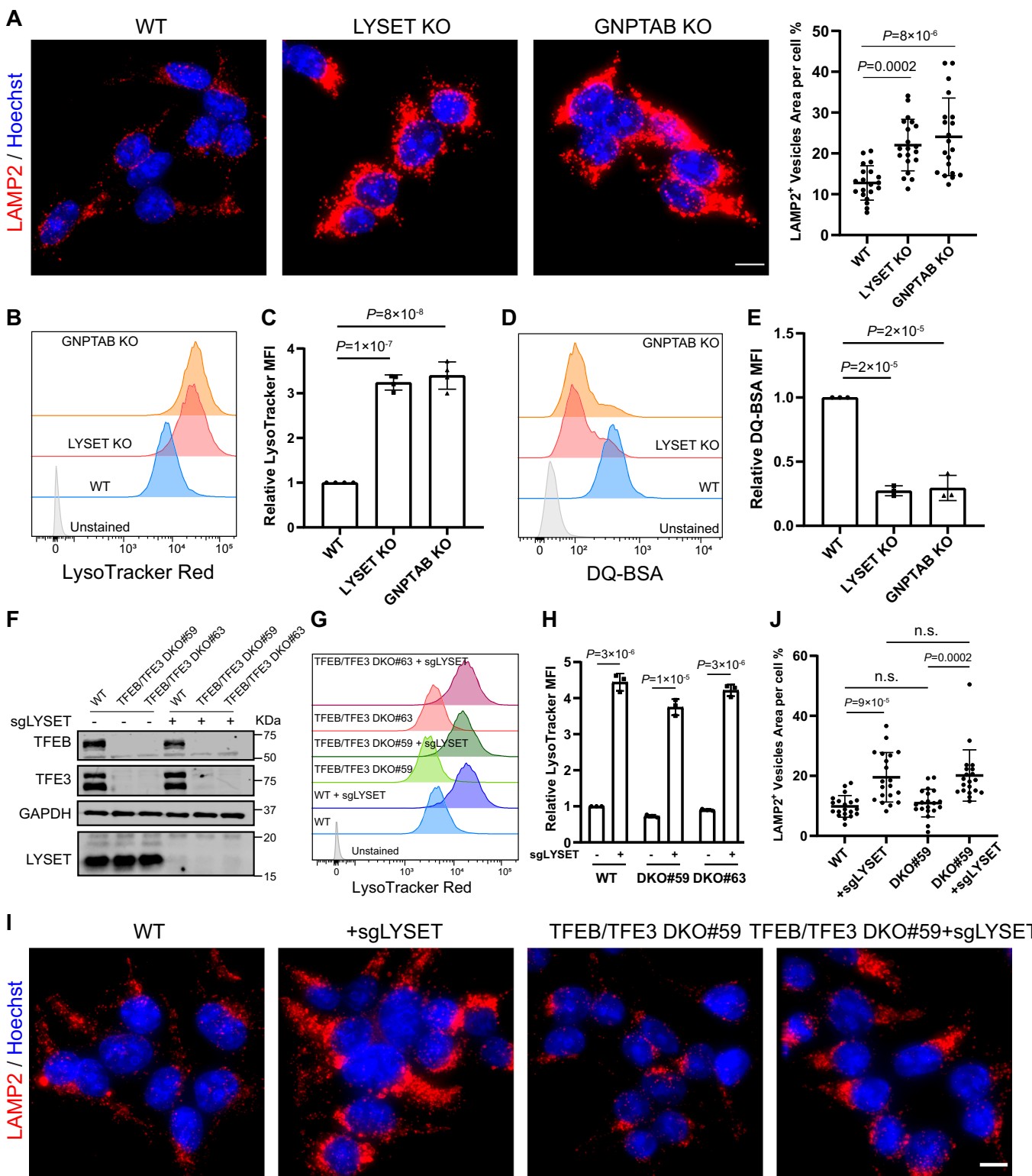

highlighted two major functional groups: 1) Peroxisome membrane transport and function, in particular ether lipid synthesis, including *FAR1, AGPS, GNPAT, ABCD3*, and 11 peroxins involved in peroxisomal protein import, and 2) Macroautophagy and autophagosome assembly, including *ATG9A, ATG10, ATG13, ATG14,*

*ATG101, RB1CC1, PIK3C3, WDR45,* and *EPG5* (Figs. 2D and E-V2A,B). The role of macroautophagy in lysosomal number upregulation will be explored in a separate study; this paper focuses on the contribution of peroxisomes to lysosomal accumulation.

◄ **Figure 1. TFEB/TFE3-independent lysosome accumulation in mucolipidosis cells.**

(A) LAMP2 immunostaining images and their quantification in WT, *LYSET* KO, and *GNPTAB* KO HEK293T cells. Data were presented as mean ± s.d.; *n* = 20 cells from three biological replicates; *P* values were calculated by one-way ANOVA with multiple comparisons. Scale bar: 10 μm. (B) LysoTracker intensity analysis. WT, *LYSET* KO, and *GNPTAB* KO HEK293T cells were stained with 50 nM LysoTracker Red for 30 min before being quantified by flow cytometry. (C) Normalized Mean Fluorescence Intensity (MFI) of (B). Data were presented as mean ± s.d.; *n* = 4 biological replicates. *P* values were calculated using one-way ANOVA with multiple comparisons. (D) WT, *LYSET KO*, and *GNPTAB* KO HEK293T cells were loaded with 5 μg/mL DQ-BSA for 6 h before being quantified by flow cytometry. (E) Normalized MFI of (D). Data were presented as mean ± s.d.; *n* = 3 biological replicates. *P* values were calculated using one-way ANOVA with multiple comparisons. (F) Western blot analysis of the indicated proteins in HEK293T cells, including WT, two independent *TFEB/TFE3* DKO clones (#59, #63), and *LYSET* KO cells generated on top of these cell lines. (G) LysoTracker intensity analysis of indicated cell lines in (F). (H) Normalized MFI of (G). Data were presented as mean ± s.d.; *n* = 3 biological replicates. *P* values were calculated by one-way ANOVA with multiple comparisons. (I) LAMP2 immunostaining of the indicated HEK293T cell lines. Scale bar: 10 μm. (J) Quantification of (I). Data were presented as mean ± s.d.; *n* = 20 cells from three biological replicates; *P* values were calculated by one-way ANOVA with multiple comparisons. Source data are available online for this figure.

Among the top 10 hits, six were peroxisome-related genes: *FAR1*, *AGPS*, *GNPAT*, *PEX5*, *PEX7*, and *PEX19* (Fig. 2E), highlighting a strong connection between peroxisomes and lysosome accumulation in *LYSET* KO cells. 4–6 independent guides targeting these top-hit genes consistently showed enrichment across three rounds of selection (Fig. EV2C), highlighting the robustness and reliability of the screen. Notably, FAR1, AGPS, and GNPAT function in the early steps of the same ether lipid biosynthetic pathway (Fig. EV2D).

Ether lipids are a subclass of phosphoglycerides with the lipid at the sn-1 position attached through an ether bond, instead of an ester bond (Fig. 2F). They can be divided into plasmanyl and plasmenyl ether lipids, depending on the type of bond linking the hydrocarbon chain to the glycerol backbone at the sn-1 position (Figs. 2F and EV2D) (Braverman and Moser, 2012; Dean and Lodhi, 2018). Plasmanyl lipids have a saturated ether bond, while plasmenyl lipids (also known as plasmalogens) have a vinyl ether bond at that position (Fig. 2F). Enzymatically, plasmanyl lipids can be converted into plasmenyl lipids/plasmalogens by TMEM189 (also known as PEDS1), a desaturase that introduces the vinyl-ether bond (Gallego-Garcia et al, 2019; Wainberg et al, 2021; Werner et al, 2020) (Fig. EV2D). Among ether lipids, plasmalogens are the most abundant, accounting for approximately 20% of total phospholipids in mammalian cells. They are particularly enriched in the brain, heart, kidney, and certain immune cells(Braverman and Moser, 2012).

As stated above, deficiencies in ether lipid biosynthesis result in severe developmental disorders named RCDP diseases (Braverman and Moser, 2012). There are five subtypes of RCDP, caused by mutations in *PEX5*, *PEX7*, *FAR1*, *AGPS*, and *GNPAT* (Baroy et al, 2015; Buchert et al, 2014; Duker et al, 2017; Purdue et al, 1997; Wanders et al, 1994). All five genes are among the top 10 hits, underscoring the critical role of ether lipids in lysosomal regulation.

PEX5, PEX7, and PEX19 are peroxisomal receptors that recognize peroxisomal targeting signal (PTS)-containing proteins, facilitating their import into peroxisomes. Specifically, PEX5 transports PTS1 cargoes, PEX7 transports PTS2 cargoes, and PEX19 transports membrane PTS (mPTS) proteins (Kumar et al, 2024). These receptors, along with downstream peroxins, are essential for importing proteins into peroxisomes and establishing the functional organelle (Kumar et al, 2024).

We hypothesize that these identified peroxins are critical because they are required to import the ether lipid synthesis enzymes into the peroxisome. Sequence analysis revealed that GNPAT, AGPS, and FAR1 contain PTS1, PTS2, and mPTS signals, respectively, suggesting

they rely on PEX5, PEX7, and PEX19 for trafficking (Fig. EV3A) (Honsho et al, 2013; Kumar et al, 2024). Indeed, deletion of *PEX19*, *PEX5*, or *PEX7* caused their respective substrates (FAR1, GNPAT, and AGPS) to be absent from peroxisomes (Fig. EV3B–F), reduced protein levels of FAR1 and GFP-GNPAT (Fig. EV3G,H), or altered post-import cleavage of AGPS-EGFP (Fig. EV3I,J), which is consistent with prior studies (Honsho et al, 2013; Mizuno et al, 2013; Skowyra and Rapoport, 2022).

Together, we infer that the peroxisomal genes highlighted in our screen converge on the ether lipid synthesis pathway, prompting us to investigate how ether lipids regulate lysosome numbers.

## Ether lipids as negative regulators of lysosome quantity in M6P-deficient cells

To confirm the importance of ether lipids in lysosomal upregulation, we knocked out *FAR1*, the rate-limiting enzyme in ether lipid synthesis (Honsho et al, 2013), in both *LYSET* and *GNPTAB* KO cells. Using the ratiometric lysosomal pH biosensor FIRE-pHLy (Chin et al, 2021), we verified that *FAR1* deletion does not alter luminal pH (Fig. EV4A), ensuring that pH-sensitive LysoTracker measurements could be accurately compared between these genetic backgrounds. LAMP2 immunostaining and LysoTracker staining confirmed that lysosome numbers were significantly reduced to nearly WT levels in both *LYSET/FAR1* DKO and *GNPTAB/FAR1* DKO cells (Fig. 3A–D and G–J). Reintroducing 3xFLAG-FAR1 into these DKO cells restored lysosome numbers (Fig. 3A–D and G–J). Notably, this lysosome reduction phenotype was not observed in *FAR1* KO cells alone, indicating that the reversal is specific to M6P-deficient cells (Fig. EV4B,C). Similar to FAR1, deletion of another key enzyme in ether lipid synthesis, *AGPS*, in *LYSET* KO cells significantly reduced both LAMP2 and LysoTracker signals, which were restored upon complementation with the *AGPS* gene (Fig. 3E–F and K–L). A similar reduction in lysosome number following *FAR1* disruption was also observed in *LYSET* KO immortalized mouse embryonic fibroblasts (iMEFs) and U2OS cells (Fig. EV5A–D), indicating a conserved effect of ether lipid synthesis on lysosomal accumulation across species and cell lines.

To determine whether the reversal phenotype is specifically due to the loss of plasmalogens, we analyzed LysoTracker intensity in cells lacking both *LYSET* and *PEDS1*, a key gene whose deletion abolishes plasmalogen production and leads to the accumulation of plasmanyl ether lipids (Fig. EV2D) (Werner et al, 2020). However, no reduction in LysoTracker signal was observed in two independent *LYSET/PEDS1* DKO lines (Fig. 3M,N), indicating that

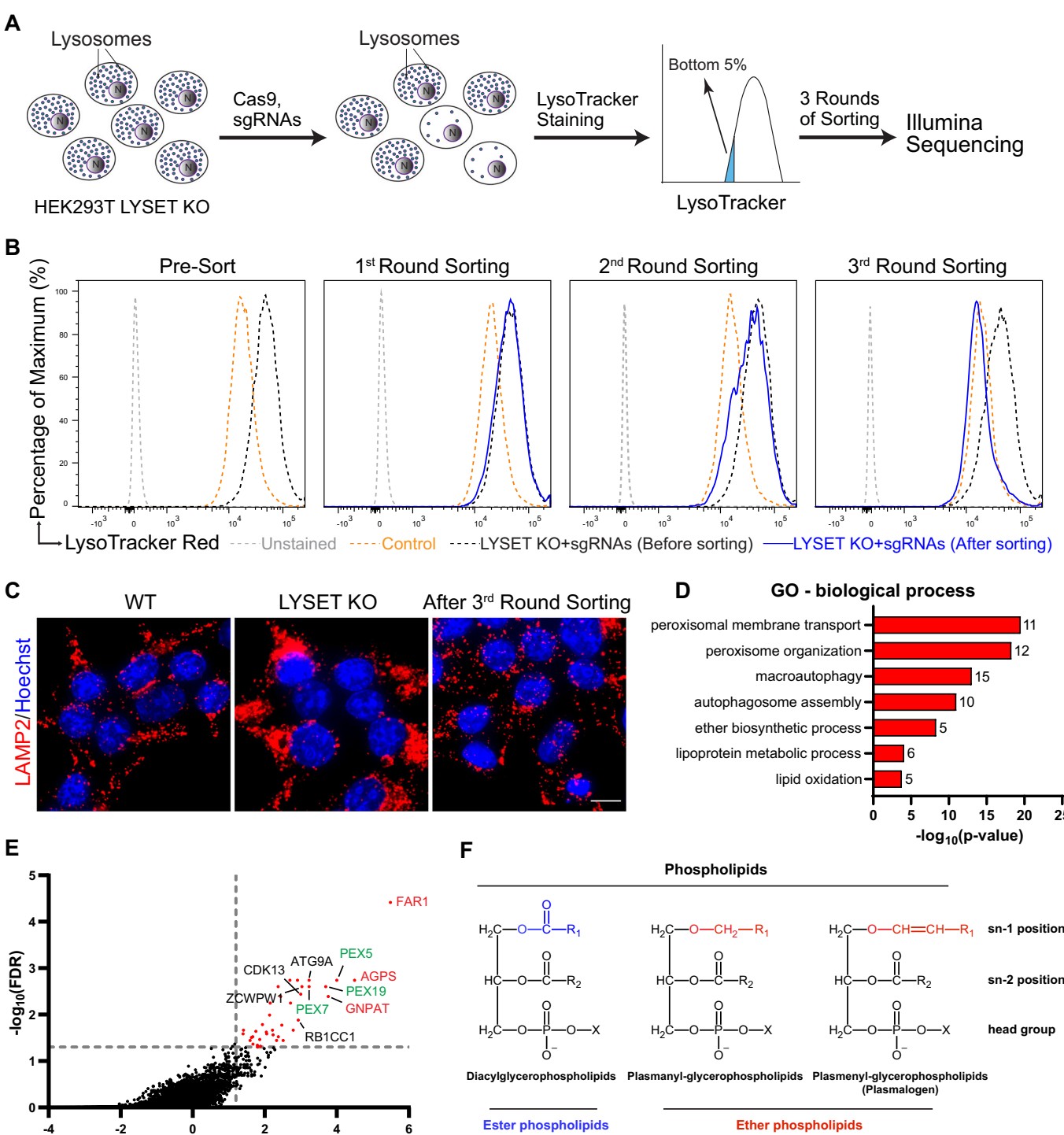

**Figure 2. A genome-wide CRISPR-Cas9 knockout screen identifies peroxisome and autophagy genes as key regulators of lysosome accumulation.**

(A) The design of the CRISPR-Cas9 knockout screen in *LYSET* KO cells. (B) Flow cytometry profiles across each round of sorting. (C) LAMP2 immunostaining images in WT, *LYSET* KO, and 3rd round sorted cells. Scale bar: 10 µm. (D) GO term analysis of enriched genes from the CRISPR-Cas9 knockout screening. Genes with log2(fold change) > 1 were used for analysis. *P* values were calculated based on the cumulative hypergeometric distribution. The number of genes in each pathway was indicated on the right side of each column. (E) Gene enrichment plot following the third round of sorting. Dashed lines indicate the thresholds for enrichment ($\log_2$[fold change] = 1.2) and statistical significance (FDR = 0.05). The top 10 hits are labeled. Peroxisomal proteins are highlighted in red (ether lipid synthesis enzymes) or green (peroxins). (F) Structural differences between ester and ether phospholipids (highlighted in blue or red at the sn-1 position). Ether phospholipids include both plasmanyl and plasmenyl variants. Source data are available online for this figure.

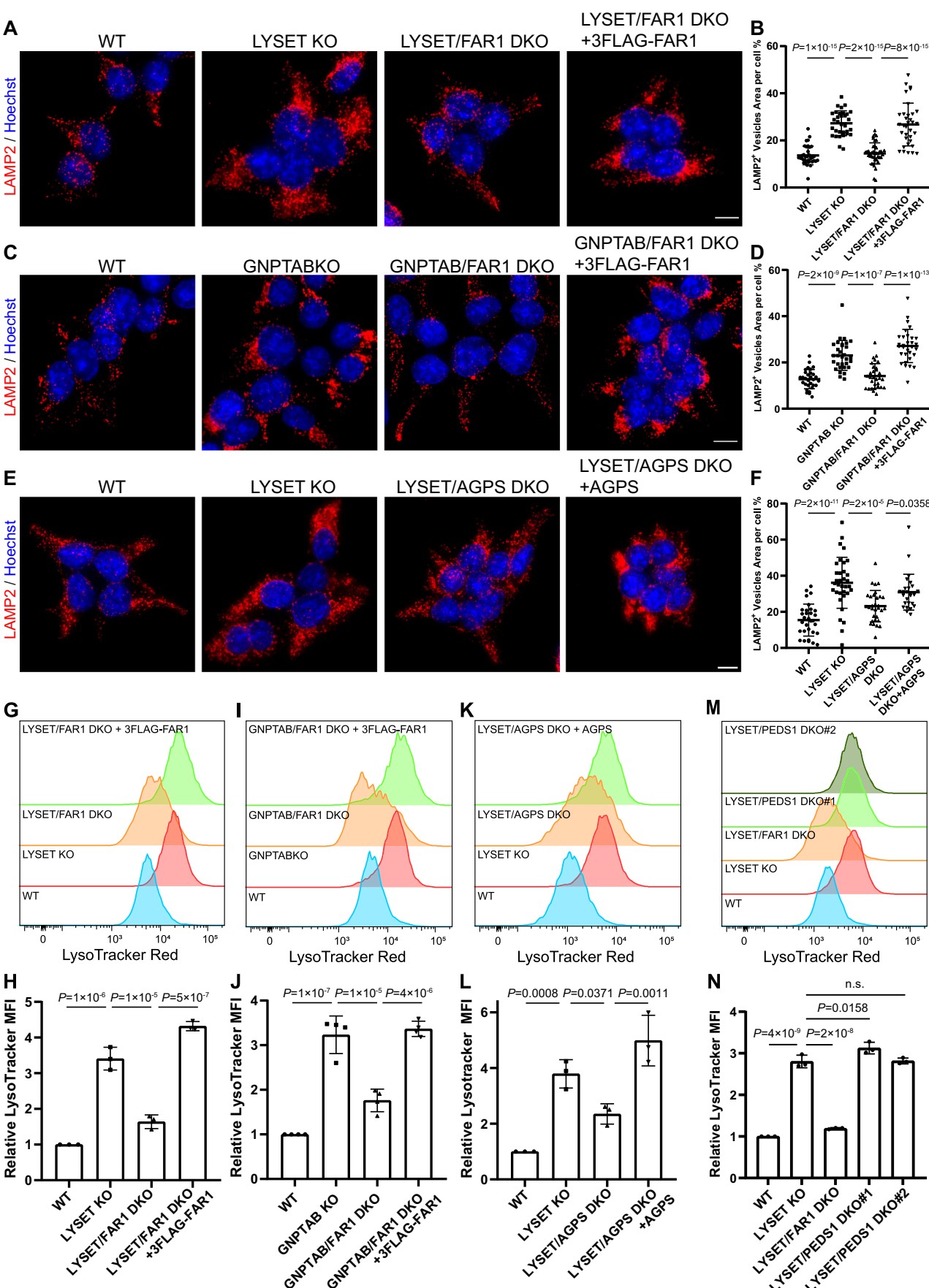

**Figure 3.   Knocking out ether lipid synthesis genes reduces lysosome numbers in M6P-deficient cells.**

(A) LAMP2 immunostaining in WT, *LYSET* KO, *LYSET/FAR1* DKO, and *LYSET/FAR1* DKO HEK293T cells stably expressing 3xFLAG-FAR1. Scale bar: 10 μm. (B) Quantification of (A). Data were presented as mean ± s.d.; *n* = 32 cells from three biological replicates; *P* values were calculated by one-way ANOVA with multiple comparisons. (C) LAMP2 immunostaining in WT, *GNPTAB* KO, *GNPTAB/FAR1* DKO, and *GNPTAB/FAR1* DKO HEK293T cells stably expressing 3xFLAG-FAR1. Scale bar: 10 μm. (D) Quantification of (C). Data were presented as mean ± s.d.; *n* = 30 cells from three biological replicates; *P* values were calculated by one-way ANOVA with multiple comparisons. (E) LAMP2 immunostaining in WT, *LYSET* KO, *LYSET/AGPS* DKO, and *LYSET/AGPS* DKO HEK293T cells stably expressing AGPS. Scale bar: 10 μm. (F) Quantification of (E). Data were presented as mean ± s.d.; *n* = 27 cells from three biological replicates; *P* values were calculated by one-way ANOVA with multiple comparisons. (G) LysoTracker Red staining (50 nM, 30 min) and flow cytometry analysis of the cell lines used in (A). (H) Normalized MFI of (G). Data were presented as mean ± s.d.; *n* = 3 biological replicates; *P* values were calculated by one-way ANOVA with multiple comparisons. (I) LysoTracker Red staining (50 nM, 30 min) and flow cytometry analysis of the cell lines used in (C). (J) Normalized MFI of (I). Data were presented as mean ± s.d.; *n* = 4 biological replicates; *P* values were calculated by one-way ANOVA with multiple comparisons. (K) LysoTracker Red staining (50 nM, 30 min) and flow cytometry analysis of the cell lines used in (E). (L) Normalized MFI of (K). Data were presented as mean ± s.d.; *n* = 3 biological replicates; *P* values were calculated by one-way ANOVA with multiple comparisons. (M) LysoTracker Red staining (50 nM, 30 min) and flow cytometry analysis of WT, *LYSET* KO, *LYSET/FAR1* DKO, and two independent clones of *LYSET/PEDS1* DKO HEK293T cells. (N) Normalized MFI of (M). Data were presented as mean ± s.d.; *n* = 3 biological replicates; *P* values were calculated by one-way ANOVA with multiple comparisons. Source data are available online for this figure.

ether lipids other than plasmalogens are critical for lysosome accumulation in M6P-deficient cells. Together, these results demonstrate that loss of ether lipids, but not plasmalogens specifically, reduces lysosome numbers in M6P-deficient cells, underscoring the critical role of ether lipid homeostasis in lysosomal regulation.

## Inhibition of ether lipid synthesis improves lysosomal proteolysis in M6P-deficient cells

Given that knocking out ether lipid synthesis genes significantly reduced lysosome numbers in M6P-deficient cells, we next investigated whether it had any effect on lysosomal digestive function. To address this, we first assessed the steady-state protein levels of two lysosomal degradation substrates, LAPTM4A and LC3B-II (Zhang et al, 2021; Zhang et al, 2022). Strikingly, in *LYSET/FAR1* DKO cells, we observed a significant reduction in full-length LAPTM4A and LC3B-II levels (Fig. 4A,B). To measure degradation kinetics, we treated the cells with cycloheximide (CHX) to inhibit protein synthesis and monitored the degradation of LAPTM4A and LC3B-II over time. These proteins exhibited faster degradation kinetics in DKO cells, suggesting either enhanced lysosomal function or an increased rate of exocytosis compared to *LYSET* KO cells (Fig. 4C,D). Using the DQ-BSA assay that relies specifically on lysosomal proteolytic activity to activate fluorescence, we confirmed a significant improvement in lysosomal proteolytic activity in *LYSET/FAR1* DKO cells compared to *LYSET* KO (Fig. 4E,F). Importantly, knocking out *FAR1* in *LYSET*-deficient iMEF and U2OS cells also partially rescued lysosomal digestive function, underscoring the conservation of this effect across cell types and species (Fig. EV5E–H).

To further support these findings, we generated *LYSET/AGPS* DKO cells. Knocking out *AGPS* also partially restored lysosomal function in *LYSET* KO cells, as evidenced by reduced levels of LAPTM4A and LC3B-II and improved DQ-BSA digestion. Reintroducing the *AGPS* gene into DKO cells reversed these effects (Fig. EV4D–H).

Besides genetic ablation, we also tested the effect of a small-molecule AGPS inhibitor, ZINC-69435460 (referred to as AGPSi) (Piano et al, 2015). Treatment with AGPSi in *LYSET* KO cells reduced LAPTM4A and LC3B-II levels in a dose-dependent manner (Fig. 4G,H). A similar reduction was observed in *GNPTAB* KO cells (Fig. 4I,J). Consistent with these findings, the DQ-BSA

assay revealed improved lysosomal digestion in AGPSi-treated cells (Fig. 4K,L). Similarly, AGPSi treatment in *LYSET* KO iMEF cells restored lysosomal digestion function (Fig. EV5I–K).

In summary, we conclude that inhibiting ether lipid synthesis, either genetically or pharmacologically, significantly improves lysosomal proteolysis function in M6P-deficient cells.

## Enhanced clearance of lysosomal storage materials after ether lipid synthesis inhibition

Given the improvement in lysosomal proteolysis function upon blocking ether lipid synthesis, we tested its effect on lysosomal storage. To isolate lysosomes, cells were pulsed with dextran-coated magnetite nanobeads (DexoMAG) for 8 h, followed by a 16-h chase in a normal medium to allow DexoMAG uptake and trafficking to lysosomal compartments (Hancock-Cerutti et al, 2022). The cells were then homogenized and processed through a magnetic column to collect the DexoMAG-enriched lysosomes (Fig. 5A). Purified lysosomes showed significant enrichment with minimal contamination from the Golgi apparatus, early endosomes, mitochondria, peroxisomes, or ER (Fig. 5B).

As expected, silver staining revealed substantial protein accumulation in *LYSET* KO lysosomes due to the lack of luminal hydrolases. However, in *LYSET/FAR1* DKO lysosomes, the accumulation was significantly reduced, nearing WT levels (Fig. 5C). To quantify the lysosomal proteome composition, we performed stable isotope labeling on WT, *LYSET* KO, and *LYSET/FAR1* DKO cells, followed by lysosome purification and mass spectrometry analysis (i.e., SILAC analysis, Fig. 5D). Lysosomes from *LYSET* KO cells displayed a drastic reduction in hydrolases, such as CTSA, CTSB, CTSV, and HEXA, consistent with defects in the M6P pathway (Fig. 5E). Moreover, approximately 2000 proteins were accumulated in *LYSET* KO lysosomes (Log2(FC) > 1 and −log10(*P* value) > 1.3), the majority of which we classified as "putative substrates". These substrates included plasma membrane receptors, proteasome components, nuclear pore complexes, autophagy-related proteins (e.g., LC3, SQSTM1), and lysosomal membrane proteins (e.g., LAPTM4A, LAPTM4B, Fig. 5E, Dataset EV2).

Comparing the lysosomal proteomes of *LYSET/FAR1* DKO and *LYSET* KO cells revealed a significant reduction for nearly all "putative substrates" in DKO cells, as indicated by a leftward shift in the volcano plot (Fig. 5F, Dataset EV2). Notably, the levels of

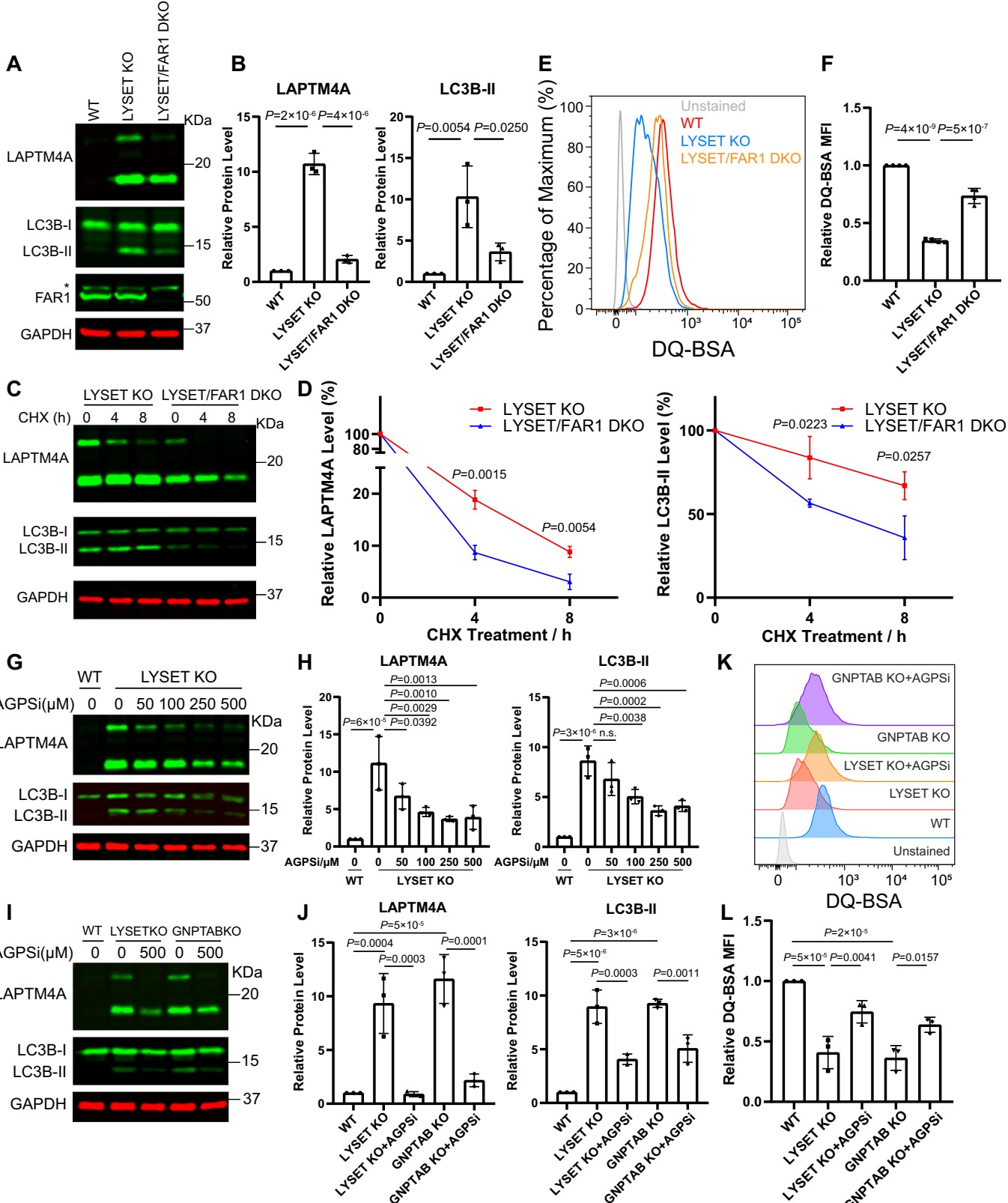

**Figure 4. Inhibition of ether lipid synthesis partially rescues lysosomal digestion in M6P-deficient cells.**

(A) Steady-state levels of LAPTM4A and LC3B-II in WT, *LYSET* KO, and *LYSET/FAR1* DKO HEK293T cells. * Indicates a non-specific band. (B) Quantification of (A), including protein levels of full-length LAPTM4A and LC3B-II. Data were presented as mean ± s.d.; $n = 3$ biological replicates; *P* values were calculated by one-way ANOVA with multiple comparisons. (C) Degradation kinetics of LAPTM4A and LC3B-II in *LYSET* KO and *LYSET/FAR1* DKO HEK293T cells after adding 100 μg/mL CHX. (D) Quantification of (C), including protein levels of full-length LAPTM4A and LC3B-II. Data were presented as mean ± s.d.; $n = 3$ biological replicates; *P* values were calculated by two-tailed unpaired t-test. (E) WT, *LYSET* KO, and *LYSET/FAR1* DKO HEK293T cells were loaded with 5 μg/mL DQ-BSA for 6 h, and the fluorescence intensities were measured by flow cytometry. (F) Normalized MFI of (E), Data were presented as mean ± s.d.; $n = 4$ biological replicates; *P* values were calculated by one-way ANOVA with multiple comparisons. (G) Steady-state levels of LAPTM4A and LC3B-II in WT and *LYSET* KO HEK293T cells treated with increasing concentrations of the AGPS inhibitor for 48 h. (H) Quantification of (G). Data were presented as mean ± s.d.; $n = 3$ biological replicates; *P* values were calculated by one-way ANOVA with multiple comparisons. (I) Steady-state levels of LAPTM4A and LC3B-II in WT, *LYSET* KO, and *GNPTAB* KO HEK293T cells treated with or without 500 μM AGPS inhibitor for 48 h. (J) Quantification of (I), Data were presented as mean ± s.d.; $n = 3$ biological replicates; *P* values were calculated by one-way ANOVA with multiple comparisons. (K) DQ-BSA analysis in *LYSET* KO and *GNPTAB* KO HEK293T cells treated with or without 500 μM AGPS inhibitor for 48 h. (L) Quantification of (K), Data were presented as mean ± s.d.; $n = 3$ biological replicates; *P* values were calculated by one-way ANOVA with multiple comparisons. Source data are available online for this figure.

lysosomal luminal enzymes remained largely unchanged, indicating that DKO did not restore the lysosomal trafficking of luminal enzymes. Western blot analysis further validated these findings, showing fewer accumulated substrates in DKO lysosomes while luminal enzyme levels were similarly reduced in both *LYSET* KO and DKO lysosomes (Fig. 5G).

In addition to proteomic analysis, we performed whole-cell and lysosomal lipidomics on these cell lines. As expected, whole-cell levels of plasmenyl-phosphatidylcholine (PC-P) and plasmenyl-phosphatidylethanolamine (PE-P) were markedly reduced in cells bearing *FAR1* KO (Fig. EV6A,B; Dataset EV4). Lysosomes isolated from *LYSET/FAR1* DKO cells exhibited vastly reduced levels of PC-P (~85% reduction) and PE-P (~ 90% reduction) compared to WT levels (Fig. EV6C,D; Dataset EV3). Importantly, levels of cholesteryl esters (CE), sphingomyelin (SM), triacylglycerols (TAG), phosphatidylcholine (PC), and phosphatidylinositol (PI) were markedly elevated in *LYSET* KO cells, likely due to deficiencies in their respective lysosomal hydrolases. These elevated lipid levels were significantly reduced in lysosomes from *LYSET/FAR1* DKO cells compared to *LYSET* KO cells, consistent with their partially restored degradative capacity (Fig. EV6E–I; Dataset EV3). Interestingly, the reduction in CE levels observed in *LYSET/FAR1* DKO cells contrasts with a previous report showing that peroxisomal disorders, including X-linked adrenoleuko-dystrophy (caused by ABCD1 mutations), infantile Refsum disease (caused by PEX1 mutations), and Zellweger syndrome (caused by PEX26 mutations), lead to marked cholesterol accumulation (Chu et al, 2015). This suggests that mutations in different peroxisomal genes may have opposing effects on lysosomal cholesterol levels. Meanwhile, levels of phosphatidylserine (PS), phosphatidylethanola-mine (PE), and phosphatidylglycerol (PG) remained relatively unchanged (Fig. EV6J–L; Dataset EV3).

Together, these results indicate that inhibiting ether lipid synthesis enhances the clearance of accumulated proteins and lipids in M6P-deficient cells, thereby improving lysosomal function.

## Excess ether lipids compromise lysosomal activity and drive lysosome accumulation

To rigorously test the regulatory role of ether lipids in lysosomal homeostasis, we supplemented *LYSET/FAR1* DKO cells with 1-O-hexadecylglycerol (HDG), a precursor of ether lipids. HDG is phosphorylated to 1-O-hexadecyl-sn-glycerol-3-phosphate and thereby bypasses the early peroxisomal steps of ether lipid synthesis (Fig. 6A) (Phuyal et al, 2015; Styger et al, 2002). As shown in

Fig. 6B,C, HDG treatment rescued LysoTracker signals in *LYSET/FAR1* DKO cells in a dose-dependent manner, mimicking the effects of *FAR1* genetic complementation.

Importantly, treating wild-type cells with excess HDG markedly increased lysosome numbers, as indicated by LysoTracker staining across multiple cell lines (HEK293T, U2OS, and HeLa; Fig. 6D–F) and by LAMP2 immunostaining (Fig. 6G). Moreover, high doses of HDG result in the accumulation of degradation substrates and a slower clearance rate (Fig. 6H–I).

These results confirm that ether lipids play a critical regulatory role in controlling lysosome number and function. Elevated ether lipid levels lead to lysosomal dysfunction and accumulation, whereas low or absent ether lipids can reduce lysosome burden and improve degradative capacity.

## Inhibiting ether lipid synthesis enhances lysosomal exocytosis and facilitates substrate clearance

Our lysosomal proteomics analysis above indicated that *FAR1* KO did not rescue levels of lysosomal enzymes in the *LYSET* KO background, suggesting that inhibiting ether lipid synthesis does not act through amelioration of M6P modification (Fig. 5E–G). To confirm this, we purified lysosomes and probed for M6P using a single-chain antibody scFv M6P (Zhang et al, 2022). As expected, WT and *FAR1* KO cells exhibited normal M6P modification, while both *LYSET* KO and *LYSET/FAR1* DKO cells showed a complete loss of M6P modification (Fig. EV7A), indicating that *FAR1* deletion does not restore this pathway.

We next considered that *FAR1* KO may enhance lysosomal exocytosis, thereby facilitating the secretion of toxic, undigested material from the cell. Lysosomal exocytosis is a process mediated by a trans-SNARE complex (VAMP7–Syntaxin4–SNAP23) (Samie et al, 2013; Samie and Xu, 2014) in which lysosomes fuse with the plasma membrane, releasing their contents into the extracellular space. Upon lysosomal calcium release through the TRPML1 channel, Synaptotagmin VII (SytVII) senses the elevated Ca²⁺ levels, triggering lysosome docking and fusion with the plasma membrane (Fig. 7A) (Reddy et al, 2001; Samie and Xu, 2014).

To test this hyper-exocytosis hypothesis, we first examined the secretion of lysosomal intraluminal vesicles (ILVs) to the extra-cellular space. These secreted vesicles are considered as extra-cellular vesicles (EVs). WT, *LYSET* KO, *LYSET/FAR1* DKO, and *FAR1* KO cells were cultured in EV-free medium for 24 h, after which conditioned medium was collected and EVs isolated via

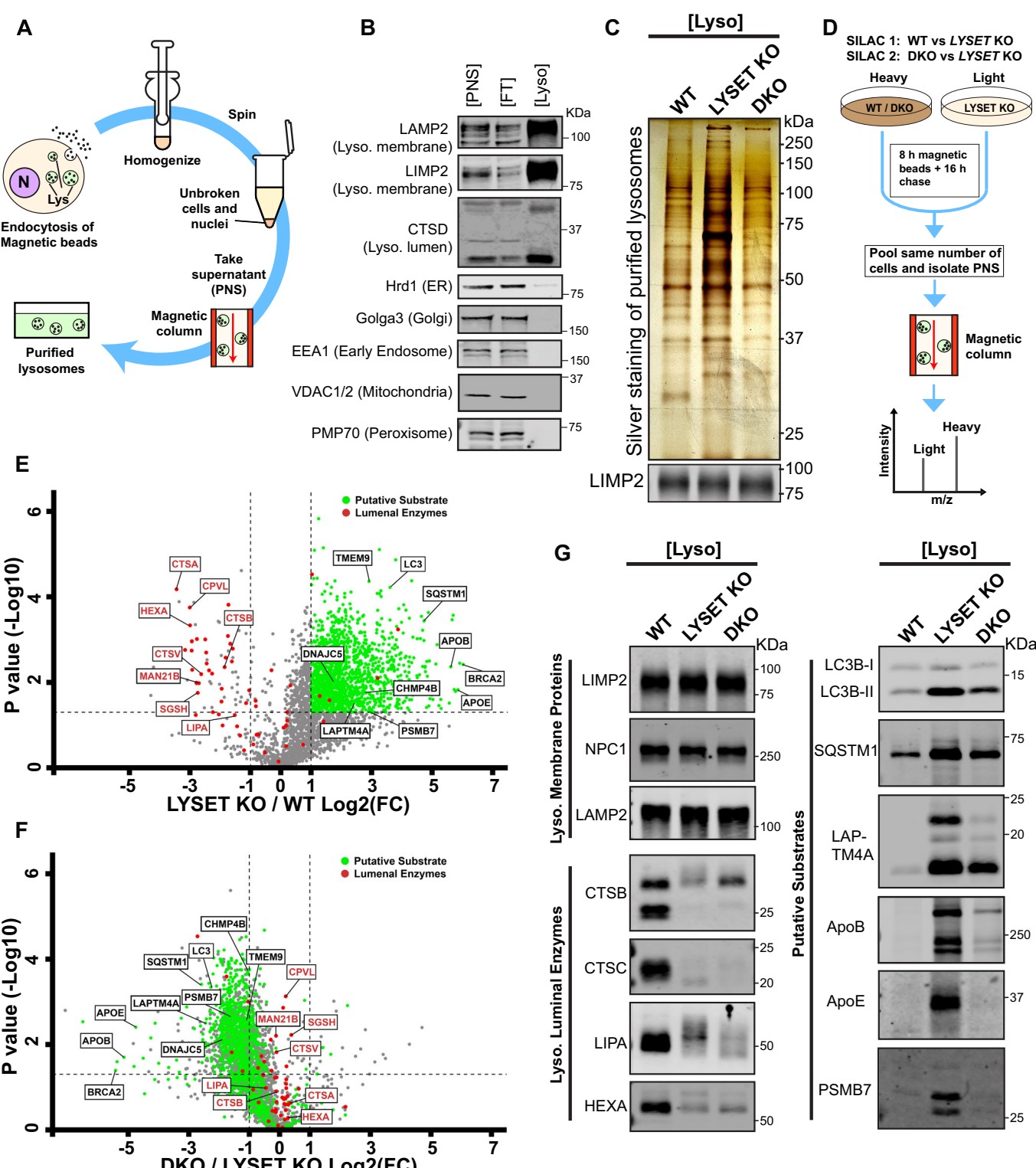

ultracentrifugation (Jackson et al, 2017). In the EV fraction, classic exosome markers (CD63, CD81, CD9, ALIX, and TSG101) were strongly increased in both *LYSET/FAR1* DKO and *FAR1* KO cells (Fig. 7B). Notably, the fact that *FAR1* knockout alone was sufficient to enhance EV secretion suggests that lysosomal dysfunction (e.g.,

*LYSET* KO) is not required for this effect. The EV fractions from *LYSET/FAR1* DKO cells also exhibited elevated levels of intraluminal vesicle markers LAPTM4A and RNF152 (Henn et al, 2025; Zhang et al, 2021), consistent with enhanced lysosomal exocytosis (Fig. 7C–F).

**Figure 5.  Lysosomal protein accumulation is reduced in *LYSET/FAR1* DKO cells.**

(A) Schematic illustration of magnetic bead-based lysosome purification. (B) Western blots showing different organelle markers in the post-nuclear supernatant (PNS), flow-through (FT), and lysosomal fractions (Lyso). (C) Silver staining of lysosomal fractions from WT, *LYSET* KO, and *LYSET/FAR1* DKO HEK293T cells. Samples were normalized to the same amount of LIMPII levels. (D) Schematic illustration of SILAC-based lysosomal proteomic analysis. Comparisons were made for WT vs. *LYSET* KO and *LYSET/FAR1* DKO vs. *LYSET* KO groups. (E) Volcano plot depicting different protein levels in purified lysosomes from *LYSET* KO and WT. Dashed lines indicate a fold change of Log$_2$(*LYSET* KO/WT) = ± 1.0 and significance of $P = 0.05$. Lysosomal enzymes are highlighted in red. Proteins with Log$_2$(*LYSET* KO/WT) > 1.0 and $p < 0.05$ are considered "putative substrates" degraded by lysosomes and are highlighted in green. $P$ values were calculated by a two-tailed unpaired t-test. (F) Volcano plot depicting different protein levels in purified lysosomes from *LYSET/FAR1* DKO and *LYSET* KO. Dashed lines indicate a fold change of Log$_2$(DKO/*LYSET* KO) = ± 1.0 and significance of $P = 0.05$. Lysosomal enzymes are highlighted in red. The "putative substrates" defined in (E) are also represented here, highlighted in green. $P$ values were calculated by a two-tailed unpaired t-test. (G) Western blot analysis of several lysosomal membrane proteins, luminal enzymes, and "putative substrates" from purified lysosomes in WT, *LYSET* KO, and *LYSET/FAR1* DKO HEK293T cells. Source data are available online for this figure.

Next, we measured extracellular levels of the mature lysosomal enzyme CTSD, which is initially synthesized as a large pre-pro-enzyme and undergoes proteolytic processing into its mature form (mCTSD) upon delivery to the lysosome (Mijanovic et al, 2021). As shown in Fig. 7G,H, the secretion of mCTSD was significantly increased following *FAR1* knockout, again suggesting increased lysosomal exocytosis.

Treating the cells with BAPTA-AM, a calcium chelator that impairs exocytosis (Medina et al, 2011), led to an accumulation of full-length LAPTM4A and LC3B-II in *LYSET/FAR1* DKO cells, revealing reduced clearance of lysosomal content (Fig. EV7B–D). Treatment with ML-SI3, a TRPML1 inhibitor that blocks lysosomal calcium release (Zhang et al, 2016), significantly reduced extra-cellular vesicle secretion (Fig. 7I) and increased LAPTM4A and LC3B-II levels in *LYSET/FAR1* DKO cells (Fig. 7J–L). These findings support the role of TRPML1 as the lysosomal calcium release channel essential for lysosomal exocytosis (Samie et al, 2013; Samie and Xu, 2014). Finally, we deleted key SNARE components involved in lysosomal exocytosis, including VAMP7 and STX4 (Rao et al, 2004; Samie and Xu, 2014), in the *LYSET/FAR1* DKO background, which also led to the accumulation of LAPTM4A and LC3B-II (Fig. EV7E–G), further confirming the role of lysosomal exocytosis in content clearance.

Together, these results suggest that inhibiting ether lipid synthesis strongly enhances lysosomal exocytosis, promoting the secretion of accumulated substrates and improving lysosomal clearance.

## Discussion

Our study uncovers a previously uncharacterized role for peroxisome-derived ether phospholipids in regulating lysosomal function. Our genome-wide screen in Mucolipidosis V (*LYSET* KO) cells identified multiple genes involved in ether lipid synthesis, as well as 11 peroxins that are likely responsible for transporting these enzymes, as suppressors of disease-associated lysosomal accumulation. Disruption of ether lipid biosynthesis, through genetic or pharmacological inhibition of key enzymes such as FAR1 and AGPS, alleviated lysosomal defects and restored lysosomal clearance capacity.

Ether lipids are known for their unique biophysical properties, including membrane stabilization and resistance to oxidative damage (Braverman and Moser, 2012; Dean and Lodhi, 2018). While these properties are critical for cellular integrity, their potential to alter lysosomal membrane dynamics or intracellular

signaling pathways remains underexplored. We speculate that disrupting ether lipid synthesis promotes lysosomal exocytosis by increasing the number or efficiency of lysosome-plasma membrane fusion events, thereby facilitating the clearance of accumulated lysosomal contents. Interestingly, ether lipid deficiency has also been shown to confer resistance to ferroptosis in cancer cells (Zou et al, 2020), likely by modulating CD44-dependent iron endocytosis (Mansell et al, 2026), suggesting a possible broader link between ether lipid levels and endomembrane trafficking. Further studies are needed to elucidate the specific biophysical or signaling properties of ether lipids that modulate lysosomal exocytosis and other membrane fusion/fission events. Our data demonstrates that the improvement in lysosomal function following inhibition of ether lipid synthesis stems from enhanced lysosomal exocytosis, as evidenced by increased secretion of lysosomal content through extracellular vesicles and the involvement of key exocytosis regulators such as TRPML1, VAMP7, and Syntaxin4. Lysosomal exocytosis is now recognized as a potential compensatory mechanism that can counteract lysosomal dysfunction in LSDs. Previous studies have demonstrated that enhancing lysosomal exocytosis improves lysosomal clearance and ameliorates disease phenotypes in models of LSDs (Atakpa et al, 2019; Medina et al, 2011). Our findings expand this paradigm by linking ether lipid deficiency to increased lysosomal exocytosis and thereby providing a new means to modulate this pathway. Importantly, we demonstrated that this effect occurs independently of the M6P pathway, suggesting a novel mechanism by which ether lipids regulate lysosomal function that could potentially be harnessed to sidestep the M6P pathway defects found in severe LSDs.

Our data further suggest that disrupting ether lipid synthesis enhances lysosomal degradative capacity in M6P-deficient cells. Lysosomal enzymes are synthesized as inactive proenzymes and are proteolytically processed into their mature forms upon delivery to lysosomes. In *LYSET* KO cells, SILAC analysis showed that most lysosomal enzymes were depleted from lysosomes; however, 14 enzymes were present at levels comparable to WT (Fig. 5E), indicating the existence of an alternative, M6P-independent trafficking pathway. Our unpublished data further suggest that these enzymes predominantly exist in their proenzyme forms.

Although our M6P modification analysis and SILAC data indicate that lysosomal enzyme trafficking is not restored in *LYSET/FAR1* DKO cells, we observe a marked shift from proenzyme to processed mature forms for several enzymes, including cathepsin D and acid α-glucosidase. This restoration of enzyme processing may underlie the enhanced lysosomal degradative capacity observed.

                                                               

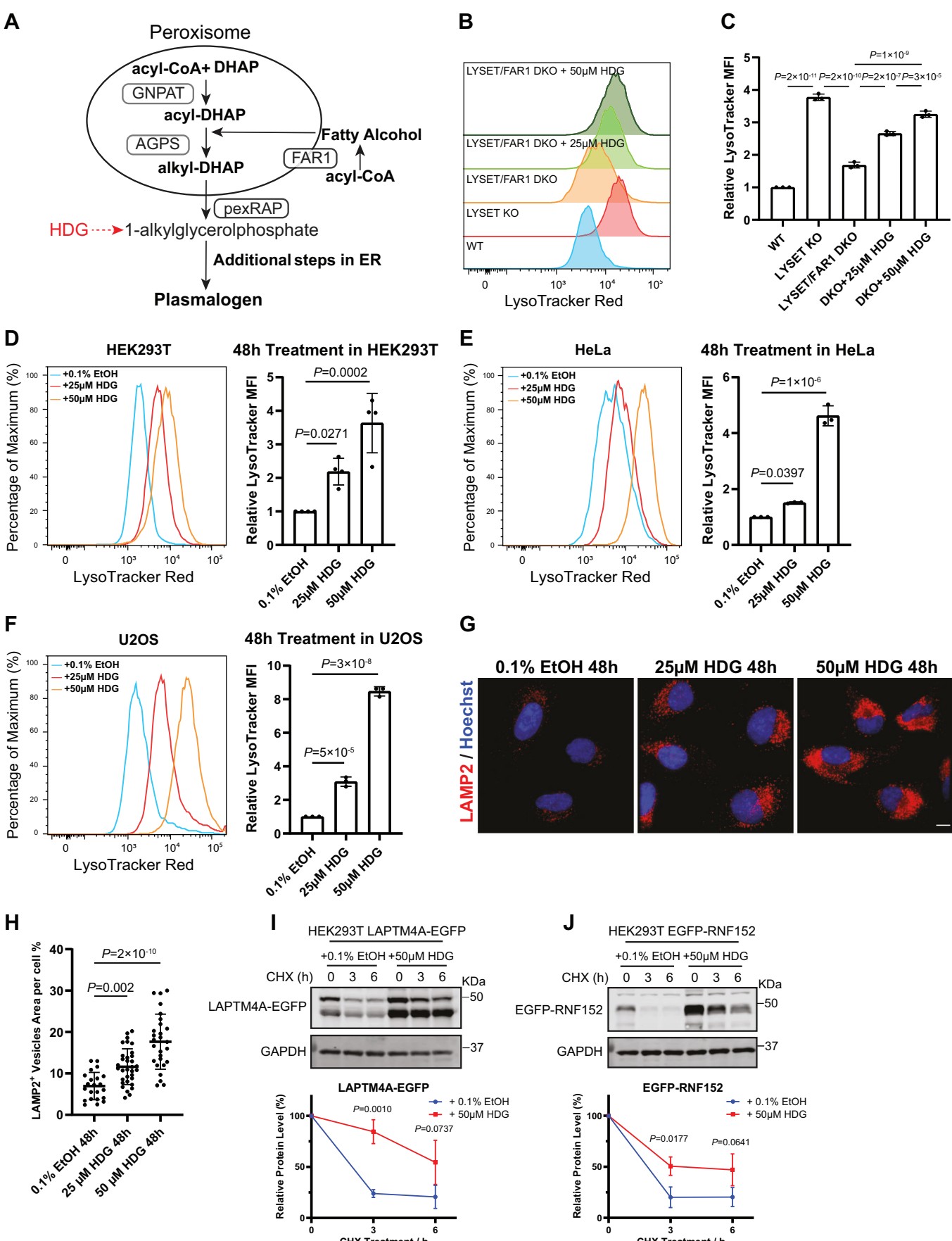

**Figure 6.   Excess ether lipids impair lysosomal function and promote lysosome accumulation.**

(A) Schematic model illustrating how HDG bypasses peroxisomal synthesis and is converted into ether lipids/plasmalogens. (B) LysoTracker Red intensities in WT, *LYSET* KO, *LYSET/FAR1* DKO, and DKO cells treated with 25 μM or 50 μM HDG for 48 h. (C) Normalized MFI of (B). Data were presented as mean ± s.d.; $n = 3$ biological replicates; *P* values were calculated by one-way ANOVA with multiple comparisons. (D–F) LysoTracker Red intensities and corresponding normalized MFI quantification in HEK293T (D), HeLa (E), and U2OS (F) cells treated with 25 μM or 50 μM HDG for 48 h. Data were presented as mean ± s.d.; $n = 4$ biological replicates; *P* values were calculated by one-way ANOVA with multiple comparisons. (G) LAMP2 immunostaining of HeLa cells treated with 0.1% EtOH (v/v), or 25 μM or 50 μM HDG for 48 h. Scale bar: 10 μm. (H) Quantification of (G). Data were presented as mean ± s.d.; $n = 27$ cells from three biological replicates; *P* values were calculated by one-way ANOVA with multiple comparisons. (I, J) Western blot analysis and quantification of HEK293T cells stably expressing LAPTM4A-EGFP (H) or EGFP-RNF152 (I). Cells were pretreated with 0.1% EtOH or 50 μM HDG for 48 h, followed by treatment with 100 μg/mL CHX for the indicated times. Data were presented as mean ± s.d.; $n = 3$ biological replicates; *P* values were calculated by multiple unpaired t-test. Source data are available online for this figure.

These findings provide a strong rationale for targeting ether lipid biosynthesis in lysosomal storage disorders (LSDs), particularly mucolipidosis. Small-molecule inhibitors of ether lipid synthesis enzymes, such as AGPS inhibitors, effectively restore lysosomal function in *LYSET* KO and *GNPTAB* KO cells. Notably, this effect is conserved across multiple cell types, underscoring the generality of this mechanism and its therapeutic potential. However, ether lipid deficiency is associated with severe developmental and neurological disorders (Braverman and Moser, 2012), suggesting that long-term inhibition of this pathway may have unintended adverse effects. Thus, more nuanced strategies, such as transient or tissue-specific modulation of ether lipid synthesis, may be required to minimize toxicity while preserving therapeutic benefit. Nonetheless, the ability to pharmacologically modulate lysosomal exocytosis opens new avenues for treating LSDs, where conventional approaches such as enzyme replacement therapy face significant limitations.

At the same time, our findings could have important implications for ether lipid deficiency disorders such as RCDP. Although lysosomal abnormalities have not been systematically examined in RCDP patient cells, our data suggest that further investigation is warranted to determine whether altered lysosomal dynamics could play a role in these disorders. For example, it is possible that aberrant lysosomal exocytosis could contribute to the bone developmental defects observed in RCDP, as excessive lysosomal exocytosis from osteoclasts may impair normal bone formation. Should this turn out to be the case, then modulating lysosomal exocytosis could represent a novel therapeutic avenue for RCDP and related ether lipid deficiency disorders.

## Methods

### Reagents and tools table

| Reagent/Resource | Reference or Source | Identifier |
| --- | --- | --- |
| **Cell lines** | | |
| HEK293T | ATCC | Cat# CRL-3216 |
| U2OS | Jay Xiaojun Tan Lab at the Ageing Institute, University of Pittsburgh | |
| Hela-TFEB-GFP | Haoxing Xu Lab at the University of Michigan | |
| Immortalized MEF (iMEF) | Diane Fingar Lab at the University of Michigan | |
| HEK293T LYSET KO | (Zhang et al, 2021) | |

| Reagent/Resource | Reference or Source | Identifier |
| --- | --- | --- |
| HEK293T GNPTAB KO | (Zhang et al, 2022) | |
| HEK293T TFEB/TFE3 DKO | This study | |
| HEK293T LYSET/FAR1 DKO | This study | |
| HEK293T GNPTAB/FAR1 DKO | This study | |
| HEK293T LYSET/AGPS DKO | This study | |
| HEK293T FAR1 KO | This study | |
| U2OS FAR1 KO | This study | |
| HEK293T EGFP-GNPAT | This study | |
| HEK293T AGPS-EGFP | This study | |
| iMEF mLYSET KO | This study | |
| **Antibodies** | | |
| Rabbit anti-TFEB | Cell Signaling Technology | Cat# 4240 |
| Rabbit anti-TFE3 | Millipore-Sigma | Cat# HPA023881 |
| Rabbit anti-LYSET | Millipore-Sigma | Cat# HPA-48559 |
| Rabbit anti-S6K | Cell Signaling Technology | Cat# 9202 |
| Rabbit anti-pS6K(Thr389) | Cell Signaling Technology | Cat# 9234 |
| Rabbit anti-4EBP1 | Cell Signaling Technology | Cat# 9644 |
| Rabbit anti-p4EBP1(Thr37/46) | Cell Signaling Technology | Cat# 2855 |
| Rabbit anti-FAR1 | Millipore-Sigma | Cat# HPA017322 |
| Rabbit anti-PEX19 | Abcam | Cat# ab137072 |
| Rabbit anti-GFP | Torrey Pines Biolabs | Cat# TP401 |
| Rabbit anti-PEX5 | Proteintech | Cat# 12545-1-AP |
| Rabbit anti-AGPS | Proteintech | Cat# 21011-1-AP |
| Rabbit anti-LAPTM4A | Millipore-Sigma | Cat# HPA068554-1 |
| Rabbit anti-LC3 | Proteintech | Cat# 14600-1-AP |

| Reagent/Resource | Reference or Source | Identifier |
| --- | --- | --- |
| Rabbit anti-LAMP2 | Developmental Studies Hybridoma Bank | Cat# H4B4 |
| Rabbit anti-LIMP2 | Novus Biologicals | Cat# NB400-129 |
| Rabbit anti-Hrd1 | Proteintech | Cat# 134723-1-AP |
| Rabbit anti-Golga3 | Proteintech | Cat# 21193-1-AP |
| Mouse anti-EEA1 | Santa Cruz Biotechnology | Cat# sc-137130 |
| Rabbit anti-VDAC1/2 | Proteintech | Cat# 10866-1-AP |
| Mouse anti-PMP70 | Abcam | Cat# ab3421 |
| Rabbit anti-CTSD | Proteintech | Cat# 21327-1-AP |
| Rabbit anti-NPC1 | Proteintech | Cat# 13926-1-AP |
| Goat anti-CTSB | R&D Systems | Cat# AF953 |
| Mouse anti-CTSC | Santa Cruz Biotechnology | Cat# sc-74590 |
| Mouse anti-LIPA | Santa Cruz Biotechnology | Cat# sc-58374 |
| Mouse anti-HEXA | Santa Cruz Biotechnology | Cat# sc-376735 |
| Rabbit anti-p62 | Proteintech | Cat# 18420-1-AP |
| Rabbit anti-ApoB | Proteintech | Cat# 20578-1-AP |
| Rabbit anti-ApoE | Abcam | Cat# ab183597 |
| Rabbit anti-PSMB7 | Cell Signaling Technology | Cat# 13207 |
| Mouse anti-CD63 | Developmental Studies Hybridoma Bank | Cat# H5C6 |
| Mouse anti-ALIX | Santa Cruz Biotechnology | Cat# sc-53540 |
| Mouse anti-TSG101 | Santa Cruz Biotechnology | Cat# sc-7964 |
| Mouse anti-CD81 | Invitrogen | Cat# 10630D |
| Mouse anti-CD9 | Millipore-Sigma | Cat# CBL162 |
| Rabbit anti-VAMP7 | Proteintech | Cat# 22268-1-AP |
| Rabbit anti-STX4 | Proteintech | Cat# 14988-1-AP |
| Mouse anti-GAPDH | Proteintech | Cat# 60004-1-Ig |
| Rabbit anti-GAPDH | Proteintech | Cat# 10494-1-AP |
| Mouse anti-Beta Actin | Proteintech | Cat# 66009-1-AP |
| Goat anti-mouse IRDye 680LT | LI-COR | Cat# 926-68020 |
| Goat anti-mouse IRDye 800CW | LI-COR | Cat# 926-32210 |
| Goat anti-rabbit IRDye 680LT | LI-COR | Cat# 926-68021 |
| Goat anti-rabbit IRDye 800CW | LI-COR | Cat# 926-32211 |
| Donkey anti-goat IRDye 800CW | LI-COR | Cat# 926-32214 |
| Anti-Rabbit HRP | Jackson Immunoresearch | Cat# 111-035-003 |
| Single-chain antibody against M6P (scFv M6P) | This study. It was generated as described previously (Zhang et al, 2022). | |
| **Recombinant DNA** | | |
| pSpCas9(BB)-2A-Puro (PX459) | Addgene | Cat #48139 |
| Lenti-multi-CRISPR | Addgene | Cat #85402 |
| CRISPR-sgTFEB-sgTFE3 | This study | |
| Lenti-CRISPR-sgLYSET | (Zhang et al, 2022) | |
| Lenti-CRISPR-sgFAR1 | This study | |
| Lenti-CRISPR-sgAGPS | This study | |
| Lenti-CRISPR-sgPEX19 | This study | |
| Lenti-CRISPR-sgPEX5 | This study | |
| Lenti-CRISPR-sgPEX7 | This study | |
| Phage2-3xFLAG-FAR1 | This study | |
| Phage2-AGPS | This study | |
| Phage2-EGFP-GNPAT | This study | |
| Phage2-AGPS-EGFP | This study | |
| Phage2-CTSD-FLAG | (Zhang et al, 2022) | |
| Phage2-LAPTM4A-EGFP | (Zhang et al, 2021) | |
| Pahge2-EGFP-RNF152 | (Zhang et al, 2021) | |
| pLJM1-FIRE-pHLy | Addgene | Cat #170775 |
| Lenti-CRISPR-sgmLYSET | This study | |
| Lenti-CRISPR-sgmFAR1 | This study | |
| Lenti-CRISPR-sgVAMP7 | This study | |
| Lenti-CRISPR-sgSTX4 | This study | |
| psPAX2 | Addgene | Cat #12260 |
| pMD2.G | Addgene | Cat #12259 |
| AG949 | University of Geneva | Cat# ABCD_AG949 |
| **Oligonucleotides and other sequence-based reagents** | | |
| PCR, qPCR primers | Custom order from Integrated DNA Technologies (IDT) | See methods |
| **Chemicals, Enzymes and other reagents** | | |
| Torin1 | Tocris Bioscience | Cat# 4247 |
| 1-O-Hexadecyl-sn-glycero (HDG) | Cayman Chemical | Cat# 25723 |
| ZINC-69435460 (AGPS inhibitor, AGPSi) | Enamine | Cat# Z1030248250 |

| Reagent/Resource | Reference or Source | Identifier |
|---|---|---|
| Cycloheximide (CHX) | Millipore-Sigma | Cat# 239763-M |
| BAPTA-AM | Cayman Chemical | Cat# 15551 |
| ML-SI3 | Gift from Haoxing Xu Lab | |
| LysoTracker DND-99 | Invitrogen | Cat# L7528 |
| DQ-BSA | Invitrogen | Cat# D12050 |
| **Software** | | |
| Fiji | https://fiji.sc/ | |
| Image Studio 6.0 | https://www.licorbio.com/image-studio | |
| FlowJo | https://www.flowjo.com/ | |
| **Other Reagents** | | |
| 10KDa cutoff Amicon Centrifugal filters | Sigma-Aldrich | Cat# UFC901024 |
| SILAC Kits | Thermo Scientific | Cat# A33972 |

## Mammalian cell culture

HEK293T (CRL-3216) cells were obtained from ATCC. U2OS, Hela-TFEB-GFP stable cell line and the immortalized MEF cell line were generously provided by Drs. Jay Xiaojun Tan, Haoxing Xu, and Diane Fingar, respectively. Cells were cultured in Dulbecco's Modified Eagle Medium (DMEM, 11965-092, Invitrogen, Grand Island, NY, USA) supplemented with 10% fetal bovine serum (FBS), (FB12999102, Fisher Scientific, Pittsburgh, PA, USA) and 1% penicillin-streptomycin (15070-063, Invitrogen, Grand Island, NY, USA). All cell lines were cultured at 37 °C in a humidified atmosphere with 5% $CO_2$ and tested negative for mycoplasma contamination. The following reagents were used at the indicated concentrations: 200 nM Torin1 (Cat. No. 4247, Tocris Bioscience, Minneapolis, MN, USA), 25 μM or 50 μM 1-O-Hexadecyl-sn-glycero (HDG) (No. 25723, Cayman Chemical, Ann Arbor, MI, USA), 50–500 μM ZINC-69435460 (AGPS inhibitor, AGPSi) (Z1030248250, Enamine, Kyiv, Ukraine), 100 μg/mL cycloheximide (CHX) (239763-M, Millipore-Sigma, Burlington, MA, USA), 10 μM BAPTA-AM (No. 15551, Cayman Chemical, Ann Arbor, MI, USA), 10 μM ML-SI3 (a gift from Dr. Haoxing Xu). For EBSS starvation assays, cells were cultured in 6-well plates in complete DMEM until reaching 70–80% confluence. Cells were then washed once with Earle's Balanced Salt Solution (EBSS; 24010-043, Thermo Fisher Scientific, Paisley, UK) and incubated in EBSS for the indicated times.

## Overexpression stable cell line generation

Stable cell lines were generated by lentivirus infection, as described previously (Zhang et al, 2021; Zhang et al, 2022). In brief, lentivirus was generated in HEK293T using the transfer plasmid, psPAX2 (Addgene, #12260) and pMD2.G (Addgene, #12259) at a 4:3:1 ratio. 72 h after transfection, the virus-containing supernatant was collected and applied through a 0.45 μm filter. Target cells were seeded in either 3.5 cm or 6 cm dishes and infected with the infectious media (DMEM containing 10% FBS, 10 μg/mL polybrene

(Millipore-Sigma, TR-1003-G), MOI between 0.3 to 0.5). Puromycin (Gibco, A11138-03) selection was used at 1 μg/mL. At least 10 days of selection was applied before subsequent analysis.

## Generation of CRISPR-Cas9 KO cell lines

Knockout cell lines were generated as described in Ran et al (2013). In brief, sgRNA guides were ligated into pSpCas9(BB)-2A-Puro (Addgene, #48139) or Lenti-multi-CRISPR (Addgene, #85402) plasmids. For single colony isolation, cells were transfected with CRISPR-Cas9 knockout plasmids. 24 h after transfection, cells were treated with 1 μg/mL puromycin for 72 h. Single cells were isolated into 96-well plates using limited dilution to a final concentration of 0.5 cells per well. The knockout colonies were screened and verified by Western blot or sequencing. For polyclonal knockout cell lines, cells were transduced with CRISPR-Cas9 sgRNA lentivirus. 48 h after transduction, cells were treated with 1 μg/mL puromycin for 10 days. The following sgRNAs were used in this study:

TFEB: 5'-CCTCCGGATGTAATCCACAG-3'
TFE3: 5'-TGTGTACAGTAGTCAAGGCG-3'
LYSET: 5'-ATGAACTTCCGTCAGCGGAT-3'
FAR1: 5'-AGCACTAATCCTTTCCACTG-3'
AGPS: 5'-CAATTTGACAGCTCATGTAG-3'
VAMP7: 5'-TTCTGAATGAGATAAAGAAG-3'
STX4: 5'-TGGTGCACCCGGGCACGGCA-3'
mLYSET: 5'-CCTTTATTCTTGTACGAGAG-3'
mFAR1: 5'-ATCTTACAGTAGCCGCACAG-3'
PEDS1: 5'-GAGGCGTCTTGTTGCCGCTG-3'

## Genome-wide CRISPR-Cas9 knockout screens

Human Brunello CRISPR knockout pooled library and virus were purchased from Addgene (73178-LV). The FACS-based CRISPR-Cas9 knockout screen was performed according to Joung et al and Lenk et al (Joung et al, 2019; Lenk et al, 2019). HEK293T *LYSET* KO cells stably expressing Cas9 were cultured in twelve 15 cm dishes at ~50% confluency and transduced with pooled lentivirus at MOI = 0.3. After 24 h, transduced cells were treated with 1 μg/mL puromycin for 7 days.

Cells were stained with LysoTracker Red and subjected to three rounds of sorting with the FACSAria III cell sorter (BD Biosciences, Franklin Lakes, NJ, USA). After each round of soring the bottom 5% of cells were collected containing the cells with low lysotracker intensity. For round 1, about $4 \times 10^8$ cells were stained and $7 \times 10^6$ cells were collected and allowed to expand for 10 days. For round 2, about $2 \times 10^8$ cells were stained and $4.0 \times 10^6$ cells were collected and allowed to expand for 10 days. Finally, about $2 \times 10^8$ cells were subjected to a third-round sorting, and $3 \times 10^6$ cells were collected.

Genomic DNA of presorted and sorted cells was purified using the Gentra Purogene kit (Qiagen, Germantown, MD, USA) according to manufacturer instructions, and quantified with the QuantiFluor dsDNA kit on a Quantus fluorometer (Promega, Madison, WI, USA). For sgRNA sequencing, the integrated sgRNAs were enriched by PCR amplification, using 16 replicate reactions per sample, with 1 μg gDNA template per reaction. Forward primers included a 10–12mer randomized sequence to mitigate impacts from low sequence diversity in the flanking constant adaptors:

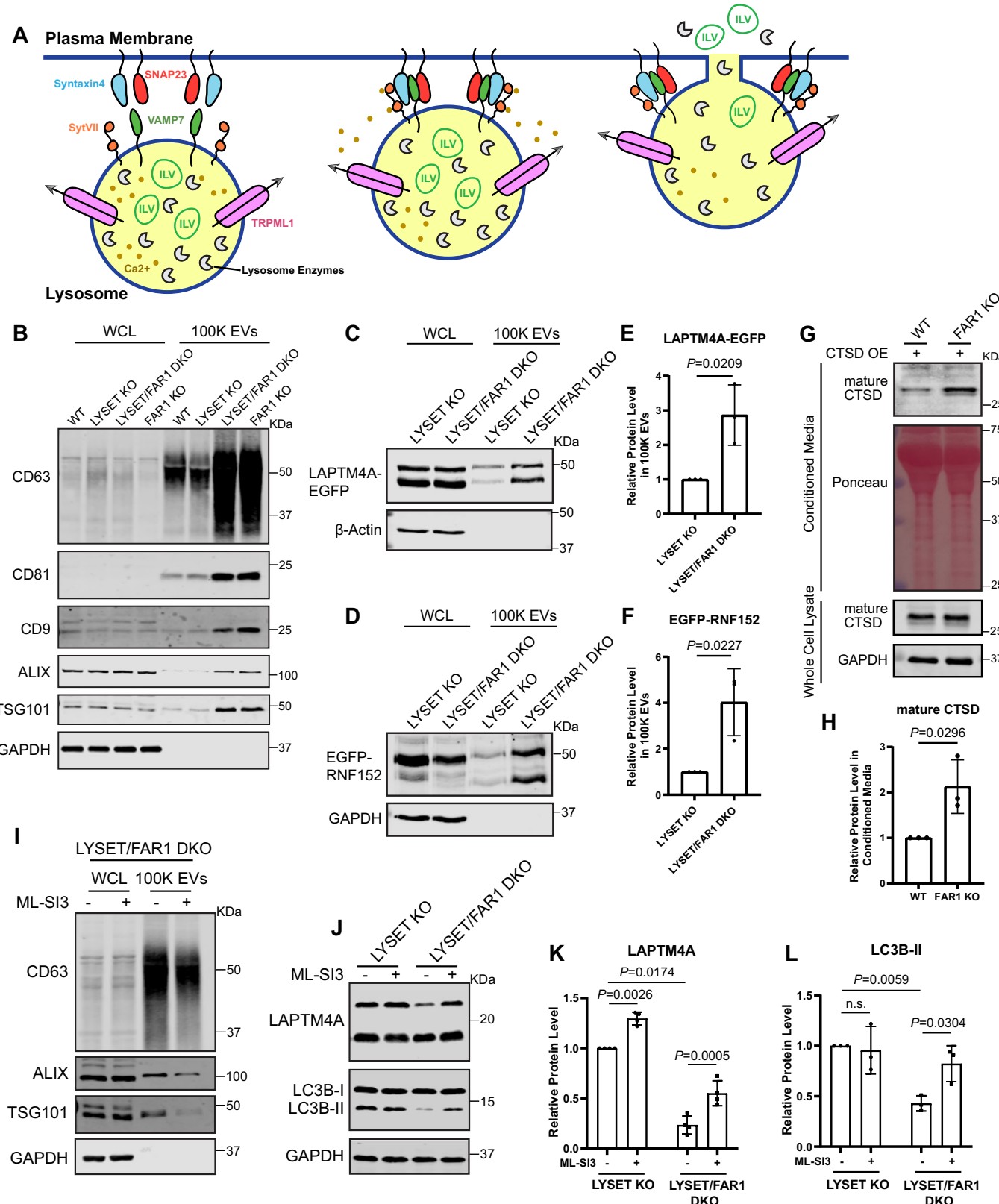

**Figure 7.  Blocking ether lipid synthesis stimulates lysosomal exocytosis.**

(**A**) Schematic illustrating lysosomal exocytosis and the associated machinery. During this process, intraluminal vesicles (ILVs) and luminal enzymes are released into the extracellular space. (**B**) Western blot analysis of exosome markers in whole-cell lysates (WCL) and extracellular vesicle fractions enriched by ultracentrifugation at $100,000 \times g$ for 16 h (100 K EVs) from 24-h conditioned media of WT, *LYSET* KO, *LYSET/FAR1* DKO, and *FAR1* KO HEK293T cells. The loaded material corresponds to $0.2 \times 10^6$ cells for WCL and $20 \times 10^6$ cells for 100 K EVs. These experiments were repeated three times with similar results. (**C, D**) *LYSET* KO or *LYSET/FAR1* DKO HEK293T cells stably expressing LAPTM4A-EGFP or EGFP-RNF152 were generated. Protein levels of LAPTM4A-EGFP (**C**) and EGFP-RNF152 (**D**) in WCL and 100 K EV fractions were analyzed by Western blots. (**E, F**) Quantification of panels (**C**) and (**D**), respectively. Data were presented as mean ± s.d.; $n = 3$ biological replicates; $P$ values were calculated by a two-tailed unpaired t-test. (**G**) Whole-cell lysates (WCL) and conditioned media from WT and *FAR1* KO HEK293T cells were analyzed for the secretion of mature cathepsin D (CTSD). Both cells were stably expressing CTSD. (**H**) Quantification of mature CTSD levels in (**G**). Data were presented as mean ± s.d.; $n = 3$ biological replicates; $P$ values were calculated by a two-tailed unpaired t-test. (**I**) Western blot analysis of the indicated proteins in WCL and 100 K extracellular vesicles from *LYSET/FAR1* DKO HEK293T cells treated with or without 10 μM of the TRPML1 inhibitor ML-SI3 for 48 h. (**J**) Western blot analysis of LAPTM4A and LC3B-II in *LYSET* KO and *LYSET/FAR1* DKO cells treated with or without 10 μM ML-SI3 for 48 h. (**K, L**) Quantification of full-length LAPTM4A and LC3B-II levels in (**J**). Data were presented as mean ± s.d.; $n = 4$ biological replicates; $P$ values were calculated by a two-tailed paired t-test. Source data are available online for this figure.

Fwd_trustub_p5_LG:  ACACTCTTTCCCTACACGACGCTCTTCCGATCT-[N10/N11/N12]-TCTTGTGGAAAGGACGAAACACCG

Rev_trustub_p7_LG:  GTGACTGGAGTTCAGACGTGTGCTCTTCCGATCT TCTACTATTCTTTCCCCTGCACTGT

Per-sample enrichment PCR replicates were pooled, and standard Illumina p5/p7 adaptor sequences and unique dual index barcodes were added for a second round of PCR. The libraries were pooled and sequenced on a NovaSeq 6000 instrument using 150-bp paired-end reads.

Flanking adaptor sequences were removed with cutadapt (Martin, 2011), and read counts for each sgRNA were identified with the BWA (Li and Durbin, 2009). Per-sgRNA read counts were processed under a beta-binomial model using CB2 to estimate (Jeong et al, 2019), for each gene, the fold-change between initial and final sorted frequencies and the Benjamini-Hochberg FDR value.

## Sample preparation and western blotting

Cells were harvested in ice-cold 1x PBS, centrifuged at $2700 \times g$ for 1 min, and lysed in lysis buffer (20 mM Tris, pH 8.0, 150 mM NaCl, 1% Triton X-100) containing 1x protease inhibitor cocktail (B14001, Selleckchem, Houston, TX, USA) and 1x phosphatase inhibitor cocktail 3 (P0044, Sigma-Aldrich St. Louis, MO, United States) (for phosphorylation blot analysis) at 4 °C for 20 min. The lysates were centrifuged at $20,000 \times g$ for 15 min at 4 °C, and the supernatants were collected. Protein concentrations were measured using Bradford reagent (#5000205, Bio-Rad, Hercules, CA, USA) following the manufacturer's instructions. Samples were normalized to equal protein concentrations and mixed with 2x urea sample buffer (150 mM Tris, pH 6.8, 6 M urea, 6% SDS, 40% glycerol, 100 mM DTT, 0.1% Bromophenol blue). For CD63 immunoblotting, samples were prepared with 2x urea sample buffer without DTT. All samples were heated at 65 °C for 10 min.

A total of 20 μg of protein was loaded onto SDS-PAGE gels, separated, and transferred to nitrocellulose membranes. For extracellular vesicles (EVs) analysis, whole-cell lysates from 0.2 million cells and EVs isolated from 20 million cells were loaded. Membranes were blocked in 5% milk or 5% BSA (for phosphorylation blot analysis) in TBST (20 mM Tris, pH 7.5, 150 mM NaCl, 0.1% Tween-20), incubated with primary antibodies overnight at 4 °C, washed five times with TBST, and subsequently incubated with secondary antibodies. Membranes were washed five times again in TBST and visualized using the Odyssey CLx imaging

system (LI-COR, Lincoln, NE, USA) or developed with CL-XPosure film (Thermo Fisher Scientific, Waltham, MA, USA). Immunoblots were quantified using Image Studio 6.0 software (LI-COR) or ImageJ for films.

## Immunostaining, microscopy, and imaging processing

Cells were cultured on 1.5 circular glass coverslips, washed with ice-cold 1x PBS, and fixed in cold 100% methanol for 8 min at −20 °C. Samples were blocked in 3% BSA (in 1x PBS) for 1 h at room temperature, followed by incubation with primary antibodies overnight at 4 °C and secondary antibodies for 1 h at room temperature. Nuclei were stained with 2 μM Hoechst (62249, Thermo Scientific, Waltham, MA, USA) for 6 min. Coverslips were mounted in Fluoromount-G (SouthernBiotech, Birmingham, AL, USA) and allowed to cure overnight before imaging.

The following antibodies were used for immunofluorescence (IF): mouse anti-LAMP2 (Developmental Studies Hybridoma Bank, H4B4, 1:100), rabbit anti-FAR1 (Millipore-Sigma, HPA017322, 1:100), rabbit anti-PMP70 (Abcam, ab3421, 1:100), mouse anti-PMP70 (Abcam, ab211533, 1:100). Secondary antibodies included TRITC goat anti-mouse (Jackson ImmunoResearch, 115-025-003, 1:200) and TRITC goat anti-rabbit (Jackson ImmunoResearch, 111-025-003, 1:200).

Microscopy was performed with a DeltaVision system (GE Healthcare Life Sciences, Marlborough, MA, USA) or a Dragonfly Confocal Microscope System (Andor Technology, Belfast, UK). The DeltaVision microscope was equipped with a scientific CMOS camera and an Olympus UPLXAP0 60X objective. The following filter sets were used: FITC (excitation, 475/28; emission, 525/48), TRITC (excitation 542/27; emission 594/45), and DAPI (excitation 390/18; emission 435/48). Image acquisition and deconvolution were performed with the SOFTWORX program. For spinning disk confocal microscopy, the following filter sets were used: FITC (excitation, 475/28; emission, 525/48), TRITC (excitation 542/27; emission 594/45), and DAPI (excitation 390/18; emission 435/48). Image acquisition and deconvolution were performed with the Fusion program. Images were further cropped or adjusted using ImageJ (National Institutes of Health).

For the LAMP2$^+$ vesicle area percentage quantification, a total of more than 30 cells from three different sets of experiments were included. Images were processed and analyzed by ImageJ (NIH). Briefly, the boundary of each individual cell was outlined as regions of interest (ROIs) in the DIC channel. The LAMP2 channel was

then separated, and the intensity was adjusted to the same scale. The individual ROI was measured for the area (cell area) and area limit to threshold (LAMP2$^+$ vesicles area). The LAMP2$^+$ vesicle area percentage was calculated by dividing the LAMP2$^+$ vesicle area by cell area.

Colocalization was calculated using Pearson's correlation coefficient (r) with JACoP from the BIOP toolbox in FIJI (ImageJ). For each sample, about 10 regions of interest (ROI) ($50 \times 50 \, \mu m$) were selected across the images. The Pearson correlation coefficient was calculated within each ROI, and the average r value across all selected regions was used for further statistical analysis.

## LysoTracker staining, DQ-BSA assay, and flow cytometry analysis

Cells were incubated with 50 nM LysoTracker DND-99 (L7528, Invitrogen, Eugene, OR, USA) for 30 min or 5 µg/mL DQ-BSA (D12050, Invitrogen, Eugene, OR, USA) for 6 h. After incubation, cells were washed with 1x PBS and trypsinized to ensure complete dissociation from the culture dishes. The detached cells were neutralized using DMEM supplemented with 10% FBS and centrifuged at $200 \times g$ for 3 min. The cell pellets were resuspended in ice-cold 1x PBS and analyzed using a Ze5 flow cytometer (Bio-Rad, Hercules, CA, USA). Data was processed and analyzed using FlowJo software.

## Lysosome purification using DexoMAG40 beads

Lysosome purification was performed based on the protocol by Hancock-Cerutti et al with modifications(Hancock-Cerutti et al, 2022). Briefly, HEK293T cells were seeded in 15 cm dishes and grown to ~50% confluency. The culture medium was replaced with fresh medium supplemented with 10 mM HEPES and 10% DexoMAG40 (Liquids Research Limited, Bangor, UK) for an 8-h pulse. Following this, the medium was exchanged with fresh medium for a 16-h chase. Cells were washed twice with ice-cold 1x PBS, scraped into 5 mL PBS on ice, and centrifuged at $200 \times g$ for 10 min at 4 °C. The cell pellet was resuspended in 3 mL homogenization buffer (HB) (5 mM Tris, 250 mM Sucrose, 1 mM EGTA, pH 7.4) supplemented with a protease inhibitor cocktail. The suspension was homogenized on ice with 30 strokes and passed through a 22 G needle 8 times. The lysate was centrifuged at $800 \times g$ for 10 min at 4 °C. The supernatant was loaded onto an HB-equilibrated magnetic LS column mounted on a MACS® MultiStand (Miltenyi Biotec, Gaithersburg, MD, USA). After washing with 20 mL HB, the column was removed and eluted with two 1 mL HB aliquots using a plunger. The eluate was centrifuged at $20,000 \times g$ for 1 h at 4 °C. The pellet was collected for further analysis.

## Silver staining

Silver staining was performed as previously described, with minor modifications (Yang et al, 2018). SDS-PAGE gels were rinsed with water and fixed overnight in fixation solution (50% methanol, 12% acetic acid, 0.05% formaldehyde). After fixation, gels were washed three times with 50% ethanol for 20 min, incubated in 0.2 g/L sodium thiosulfate ($Na_2S_2O_3$; Cat#72049, Sigma, St. Louis, MO, USA) for 1 min, and stained with 4 g/L silver nitrate ($AgNO_3$; Cat#209139, Sigma, St. Louis, MO, USA) for 20 min. Gels were then developed in a solution containing 60 g/L sodium carbonate, 4 mg/L sodium thiosulfate, and 0.05% formaldehyde prior to imaging.

## Extracellular vesicles (EVs) isolation

EVs were isolated following previously described protocols (Jackson et al, 2017; Thery et al, 2006; Wang et al, 2021), with slight modifications. Briefly, HEK293T cells were seeded in two 15 cm dishes or four 10 cm dishes and allowed to reach ~50% confluence the next day. The cells were then incubated for 24 h in EV-free culture medium. EV-free culture medium was prepared by centrifuging complete medium at $110,000 \times g$ for 20 h at 4 °C, followed by sterilization through a 0.2 µm filter.

Conditioned medium was collected and subjected to sequential centrifugation: $500 \times g$ for 10 min at 4 °C to remove cells, followed by $4000 \times g$ for 20 min at 4 °C to eliminate cell debris and apoptotic bodies. The supernatant was filtered through a 0.2 µm filter to remove large EVs. The filtered medium was ultracentrifuged at $110,000 \times g$ for 90 min at 4 °C to pellet small EVs. After carefully removing the supernatant, the pellet was washed with PBS and subjected to a second round of ultracentrifugation at $110,000 \times g$ for 90 min at 4 °C. The final EV pellet was resuspended in the lysis buffer (20 mM Tris, pH 8.0, 150 mM NaCl, 1% Triton X-100) containing 1x protease inhibitor cocktail for WB analysis.

## Secretion assay

HEK293T WT and *FAR1* KO cells stably expressing CTSD were cultured in 10 cm dish to reach 70–80% confluence. Cells were washed with serum-free DMEM twice and incubated with 10 mL serum free media for 20 h. The conditioned media were collected and transferred to a 50 mL falcon tube. The media was centrifuged at $1000 \times g$ for 20 min at 4 °C to remove the cell debris. The supernatant was concentrated to ~150 µl using 10KDa cutoff Amicon Centrifugal filters (UFC901024, Sigma-Aldrich, St. Louis, MO, USA). Samples were mixed with 4x LDS sample buffer (NP0007, Invitrogen, Carlsbad, CA, USA) and heated at 65 °C for 8 min. Loading volume for SDS-PAGE was normalized based on equal same cell numbers.

## Stable isotope labeling by amino acids in cell culture (SILAC) and mass spectrometry analysis of lysosomal proteins

SILAC labeling was performed following the manufacturer's instructions (A33972, Thermo Scientific, Waltham, MA, USA). Briefly, cells were cultured for 10 days in SILAC DMEM medium supplemented with 10% dialyzed FBS, 1% Penicillin-Streptomycin, and isotopically labeled amino acids. Light labeling was achieved with 100 mg/L L-Arginine-HCl and 100 mg/L L-Lysine-2HCl, and heavy labeling was achieved with 100 mg/L $^{13}C_6{}^{15}N_4$ L-Arginine-HCl and 100 mg/L $^{13}C_6{}^{15}N_2$ L-Lysine-2HCl.

For lysosomal proteome analysis, equal numbers ( ~ 40 million) of WT HEK293T (heavy-labeled) and HEK293T *LYSET* KO (light-labeled) cells, or HEK293T *LYSET/FAR1* DKO (heavy-labeled) and HEK293T *LYSET* KO (light-labeled) cells, were combined prior to lysosomal purification. Approximately 25 µg of purified lysosomes per sample were resolved by SDS-PAGE and stained with SyproRuby gel stain (S12000, Invitrogen, Carlsbad, CA, USA). Each sample was prepared in triplicate. Entire gel lanes were excised and submitted to the Taplin Mass Spectrometry Facility at Harvard Medical School for mass spectrometry analysis.

 

Excised gel lanes were cut into approximately 1 mm³ pieces. The samples were reduced with 1 mM DTT for 30 min at 60 °C, then alkylated with 5 mM iodoacetamide for 15 min in the dark at room temperature. Gel pieces were then subjected to a modified in-gel trypsin digestion procedure (Shevchenko et al, 1996). Gel pieces were washed and dehydrated with acetonitrile for 10 min, followed by the removal of acetonitrile. Gel pieces were then completely dried in a SpeedVac. Rehydration of the gel pieces was performed with 50 mM ammonium bicarbonate solution containing 12.5 ng/µl modified sequencing-grade trypsin (Promega, Madison, WI) at 4 °C. Samples were then placed in a 37 °C room overnight. Peptides were later extracted by removing the ammonium bicarbonate solution, followed by one wash with a solution containing 50% acetonitrile and 1% formic acid. The extracts were then dried in a speed-vac ( ~ 1 h). The samples were stored at 4 °C until analysis. On the day of analysis, the samples were reconstituted in 5–10 µl of HPLC solvent A (2.5% acetonitrile, 0.1% formic acid). A nano-scale reverse-phase HPLC capillary column was created by packing 2.6 µm C18 spherical silica beads into a fused silica capillary (100 µm inner diameter × ~ 30 cm length) with a flame-drawn glass tip (Peng and Gygi, 2001). After equilibrating the column, each sample was loaded via a Famos auto sampler (LC Packings, San Francisco, CA) onto the column. A gradient was formed, and peptides were eluted with increasing concentrations of solvent B (97.5% acetonitrile, 0.1% formic acid). As each peptide was eluted, it was subjected to electrospray ionization and then entered an LTQ Orbitrap Velos Pro ion-trap mass spectrometer (Thermo Fisher Scientific, San Jose, CA). Eluting peptides were detected, isolated, and fragmented to produce a tandem mass spectrum of specific fragment ions for each peptide. Peptide sequences (and hence protein identity) were determined by matching protein or translated nucleotide databases with the acquired fragmentation pattern by the software program, Sequest (ThermoFinnigan, San Jose, CA) (Eng et al, 1994). The differential modification of 8.0142 and 10.0083 mass units for lysine and arginine, respectively, were included in the database searches to find SILAC-labeled peptides. All databases include a reversed version of all the sequences, and the data was filtered to a one percent or lower peptide false discovery rate. The mass spectrometry data have been deposited to the ProteomeXchange Consortium via the PRIDE (Perez-Riverol et al, 2025) partner repository.

## Lipidomics

All solvents used were HPLC or LC/MS grade and purchased from Sigma-Aldrich (St. Louis, MO, USA). Splash® Lipidomix® standards were obtained from Avanti (Alabaster, AL, USA). Lipid extractions were performed using a modified methyl tert-butyl ether (mTBE) method in 16 × 100 mm glass tubes with PTFE-lined caps (Fisher Scientific, Pittsburgh, PA, USA) to minimize contamination (Matyash et al, 2008). Glass Pasteur pipettes, automated eVol syringes (Trajan, Australia), and solvent-resistant pipette tips (Mettler-Toledo, Columbus, OH, USA) were used to further reduce contamination risks.

For each extraction, lysosomes purified from ~2 × 10⁸ cells were transferred to glass tubes. Lipids were extracted with 1 mL water, 1 mL methanol, and 2 mL mTBE, followed by vortexing and centrifugation at 2671 × g for 5 min. The organic phase was collected, spiked with 20 µL diluted Splash® Lipidomix® standards, dried under nitrogen, and resuspended in 400 µL hexane.

Lipid profiling and ether lipid analysis were performed by LC-MS/MS using a SCIEX QTRAP 6500+ coupled to a Shimadzu LC-30AD HPLC system. Samples were separated on a 150 × 2.1 mm Supelco Ascentis silica column at a flow rate of 0.3 mL/min, with a gradient involving solvents A (hexane), B (mTBE), C (isopropanol-water, 90:10), and D (acetonitrile-water, 95:5 with 10 mM ammonium acetate). The oven temperature was maintained at 25 °C. Data acquisition was done in positive and negative ionization modes using multiple reaction monitoring (MRM), and results were analyzed using MultiQuant software (SCIEX). Lipid species were normalized to their corresponding internal standards. Lysosomal lipid species results were further normalized to their corresponding LIMP2 protein levels for further statistical analysis.

## RNA isolation and RT-qPCR

RNA was extracted from approximately 1.5 million cells using TRIzol (15-596-026, Invitrogen, Carlsbad, CA, USA) and the PureLink RNA Mini Kit (12183025, Invitrogen, Carlsbad, CA, USA), following the manufacturer's instructions. A total of 5 µg of RNA was reverse-transcribed to cDNA in a 20 µl reaction using the High-Capacity cDNA Reverse Transcription Kit (Applied Biosystems, Foster City, CA, USA). Quantitative real-time PCR was performed on a StepOne™ Plus qPCR system (Applied Biosystems). The 10 µl reaction mix consisted of 1 µl cDNA, 1 µl gene-specific primer mix (10 µM), 5 µl SYBR™ Green PCR Master Mix (Applied Biosystems), and 2 µl water. PCR cycling conditions were as follows: 50 °C for 2 min, 95 °C for 10 min, followed by 40 cycles of 95 °C for 15 s and 60 °C for 1 min. Target gene expression levels were normalized to the endogenous reference gene β-actin, and data were calculated using the $2^{-\triangle\triangle Ct}$ method. The qPCR primers used in this study are listed below:

ATP6v0d1:
Forward: 5'-GAGAAGATGGTGGTGGAGTTC-3'
Reverse: 5'-GGATCACGTTGTCGATCATGTA-3'
Mcoln1:
Forward: 5'-TCTTCCAGCACGGAGACAAC-3'
Reverse: 5'-GCCACATGAACCCCACAAAC-3'
Mcoln3:
Forward: 5'-ATGCTCGTGTGGCTTGGAGTCA-3'
Reverse: 5'-CCATCCACAGAAGCAGTAACCTA-3'
CLCN7:
Forward: 5'-CACAGTTGCCTTCGTGCTGATC-3'
Reverse: 5'-TGGAGTTGTACTCGCCATCTGC-3'
HEXA:
Forward: 5'-GGAGGTCATTGAATACGCACGG-3'
Reverse: 5'-GGATTCACTGGTCCAAAGGTGC-3'
HEXB:
Forward: 5'-GATCCATTGTCTGGCAGGAGGT-3'
Reverse: 5'-GGAAGCCAGATGCTGTGACTCT-3'
NEU1:
Forward: 5'-AAGGCTGAGAACGACTTCGG-3'
Reverse: 5'-CCTCAGCAAAGGCGAGAAGA-3'
CTSA:
Forward: 5'-GCTTCGTGAAGGAGTTCTCCCA-3'
Reverse: 5'-CTGTGGTCATCAGTATGGCTGC-3'

 

CTSD:
Forward: 5'-GGACTACACGCTCAAGGTG-3'
Reverse: 5'-GTTGTCACGGTCAAACACAG-3'
GLA:
Forward: 5'-CAATGGATTGGCAAGGACGC-3'
Reverse: 5'-TCTTTGGGGAGCCATCCAAC-3'
NPC2:
Forward: 5'-GGAGTGGCAACTTCAGGATGAC-3'
Reverse: 5'-CTGGAGGTGCTGTCAAGAGTCT-3'
ACTB:
Forward: 5'-TCCCTGGAGAAGAGCTACGA-3'
Reverse: 5'-AGCACTGTGTTGGCGTACAG-3'
PEX7:
Forward: 5'-CTTCAGCCTCAGGTGATCAG-3'
Reverse: 5'-CTCATTGTATTTACACCAGTCAC-3'

## Flow cytometry-based lysosomal pH measurement using FIRE-pHly

pH measurement was performed following the protocol by Chin et al, with modifications (Chin et al, 2021). Briefly, cells stably expressing FIRE-pHLy (Addgene, #170775) were harvested, pelleted, and resuspended in 1 mL pH calibration buffer (5 mM NaCl, 115 mM KCl, 1.3 mM MgSO$_4$·7H$_2$O, 25 mM MES, 10 µM Nigericin (J61349, Thermo Scientific, Waltham, MA, USA), 1x Monensin (00-4505-51, Invitrogen, Carlsbad, CA, USA) with pH values at 3.5, 4.0, 4.5, 5.0, 5.5, 6.0, 6.5, or 7.0), or in 1 mL PBS. Cells were incubated for approximately 20 min before analysis using a flow cytometer. Fluorescence intensities of mTFP and mCherry were measured using the 509/24 and 615/24 filters, respectively, and mean fluorescence intensities (MFI) were calculated using FlowJo software. The MFI ratio (mTFP/mCherry) was determined for each sample, and standard curves for each cell line were generated by plotting MFI ratios against the corresponding pH values. Lysosomal pH values were derived by referencing the standard curves.

## Data availability

The proteomics data have been deposited to the ProteomeXchange Consortium via the PRIDE partner repository with the dataset identifier PXD074652 and https://doi.org/10.6019/PXD074652.

The source data of this paper are collected in the following database record: biostudies:S-SCDT-10_1038-S44318-026-00791-3.

## Peer review information

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

## Acknowledgements

We thank members of the Li laboratory for their insightful discussions and technical support. We are also grateful to Drs. Qi Geng, Jay Tan, Ken Inoki, Haoxing Xu, Diane Fingar, Yihong Ye, and Kristen Verhey for generously sharing reagents. Special thanks to the Taplin Mass Spectrometry Facility for conducting the SILAC analysis. This research was supported by the Protein Folding and Diseases Initiative, a MICHR Pathway Pilot grant, and NIH grants R01HD109346 and R35GM153356 to M Li, as well as R35GM153286 to J Kitzman and UL1TR003163, P01HL160487, and 1P30DK127984-01A1 to J MacDonald.

## Author contributions

**Liang Chen**: Conceptualization; Resources; Data curation; Formal analysis; Validation; Investigation; Visualization; Methodology; Writing—original draft; Writing—review and editing. **Danielle Henn**: Data curation; Formal analysis; Validation; Investigation; Visualization; Writing—original draft; Writing—review and editing. **Zhongzheng Dong**: Data curation; Formal analysis; Validation; Investigation. **Jiaxuan Liang**: Data curation; Investigation. **Aleksander Wielenga**: Data curation; Formal analysis. **Goncalo Vale**: Data curation; Formal analysis; Investigation; Methodology. **Bala Burugula**: Investigation. **Junyi Zou**: Investigation. **Yamuna Krishnan**: Resources; Supervision. **Jeffrey G McDonald**: Resources; Supervision; Funding acquisition. **Jacob Kitzman**: Resources; Software; Supervision; Funding acquisition; Methodology; Writing—review and editing. **Ming Li**: Conceptualization; Resources; Data curation; Formal analysis; Supervision; Funding acquisition; Investigation; Visualization; Methodology; Project administration; Writing—review and editing.

Source data underlying figure panels in this paper may have individual authorship assigned. Where available, figure panel/source data authorship is listed in the following database record: biostudies:S-SCDT-10_1038-S44318-026-00791-3.

## Disclosure and competing interests statement

The University of Michigan has filed a provisional patent application based on the results of this study, with ML and LC listed as inventors. The authors declare no other competing interests.

# Expanded View Figures

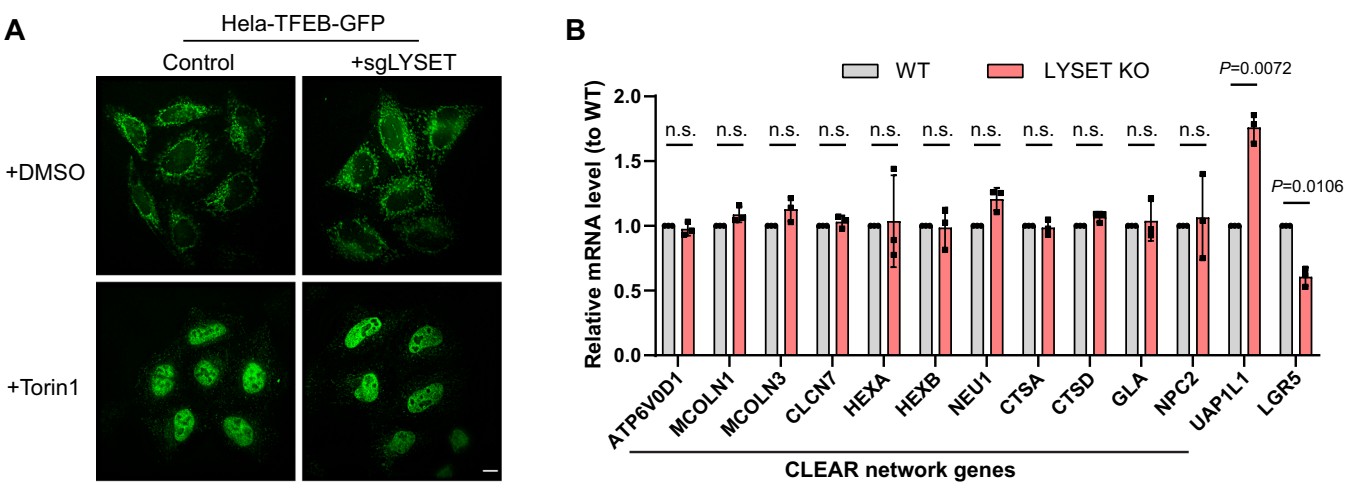

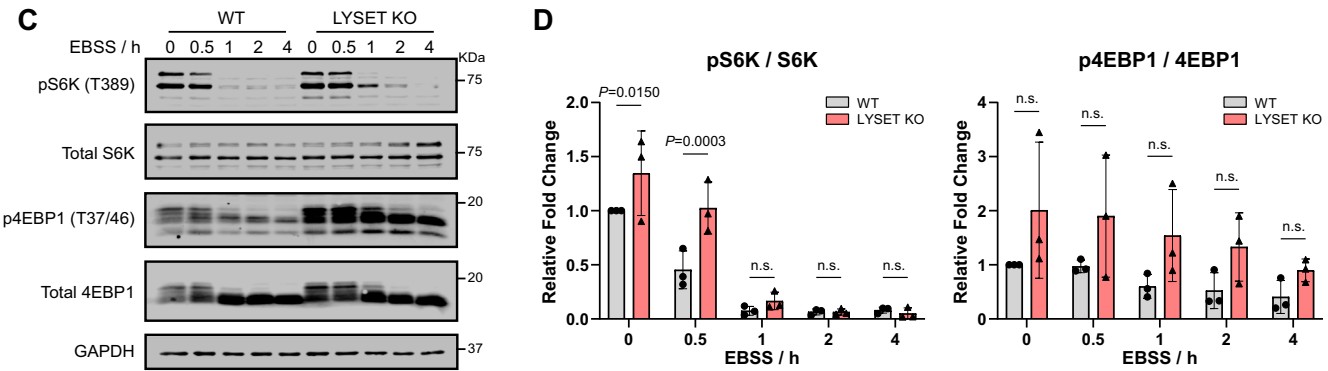

**Figure EV1. Additional evidence for mTORC1- and TFEB/TFE3-independent lysosome accumulation in *LYSET* KO cells. Related to Figure 1.**

(A) Fluorescence images of WT and *LYSET* KO HeLa cells stably expressing TFEB-GFP. Cells were treated with either dimethyl sulfoxide (DMSO) or 200 nM Torin1 (an mTORC1 inhibitor) for 1 hr. Scale bar: 10 µm. (B) Comparison of CLEAR network gene transcription levels in *LYSET* KO and WT HEK293T cells. Data were presented as mean ± s.d.; $n = 3$ biological replicates. *P* values were calculated using multiple unpaired t-tests with Welch's correction. (C) Western blot analysis of S6K and 4EBP1 phosphorylation in WT and *LYSET* KO HEK293T cells under EBSS treatment. The experiments were repeated three times with similar results. (D) Quantification of (C). Data were presented as mean ± s.d.; $n = 3$ biological replicates; *P* values were calculated by two-way ANOVA with multiple comparisons. Source data are available online for this figure.

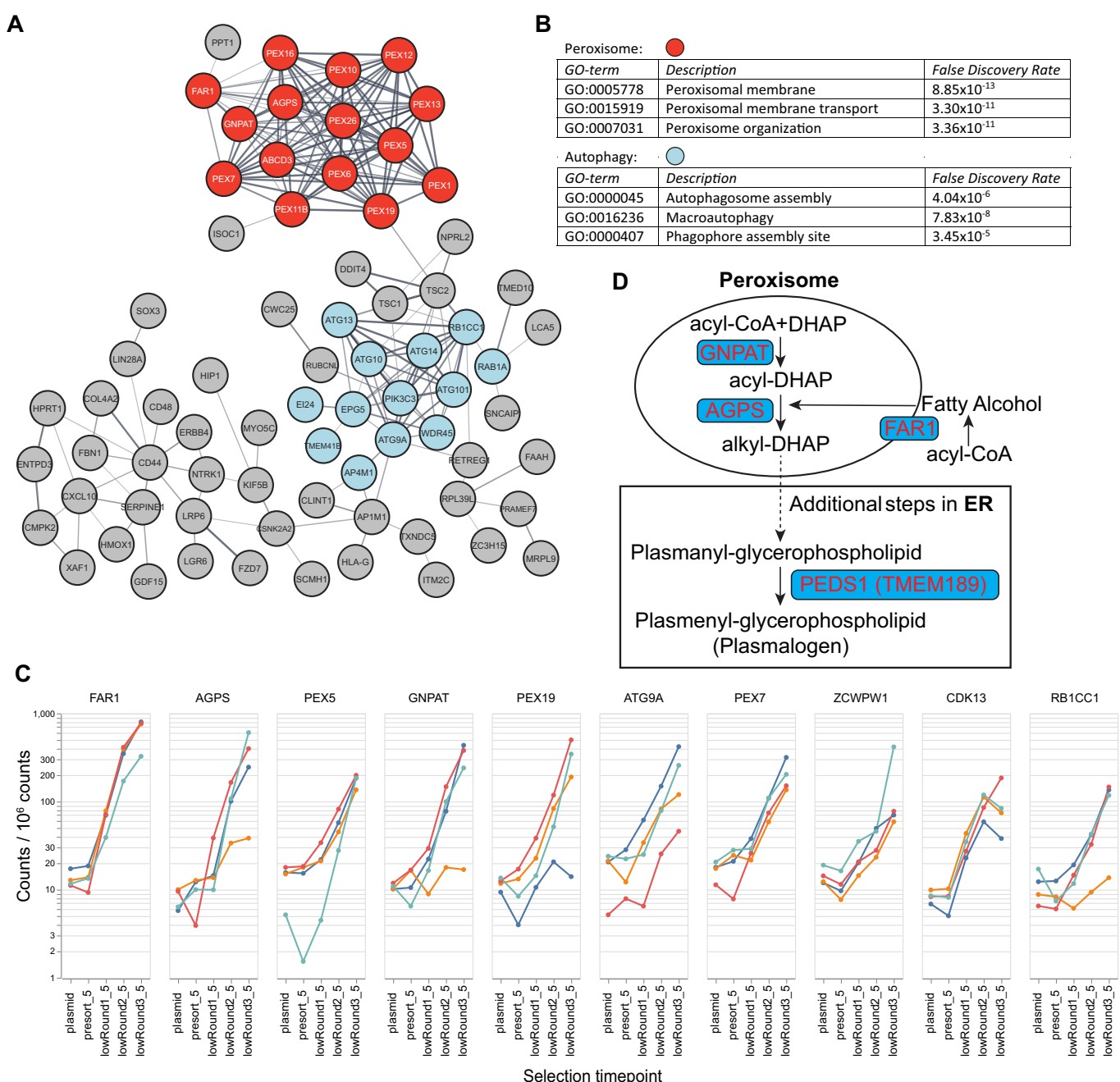

**Figure EV2.  Further analysis of CRISPR-Cas9 KO screen results. Related to Figure 2.**

(A) STRING network analysis of enriched genes (log2(fold change) > 1) from the screen. Peroxisome genes are highlighted in red, and autophagy genes in blue. (B) GO terms, descriptions, and FDR values of enriched pathways identified by STRING analysis in (A). (C) Consistent enrichment across individual sgRNAs. For each top-hit gene, per-sgRNA frequencies (counts per million) are plotted across selection time points. "Plasmid" denotes the starting plasmid library, and "presort" denotes the transduced cell population prior to sorting. Each color represents an independent guide targeting a given gene. (D) The peroxisomal steps of the plasmalogen synthesis pathway. The final ER step, in which PEDS1 converts plasmanyl to plasmenyl phospholipids, is also highlighted.

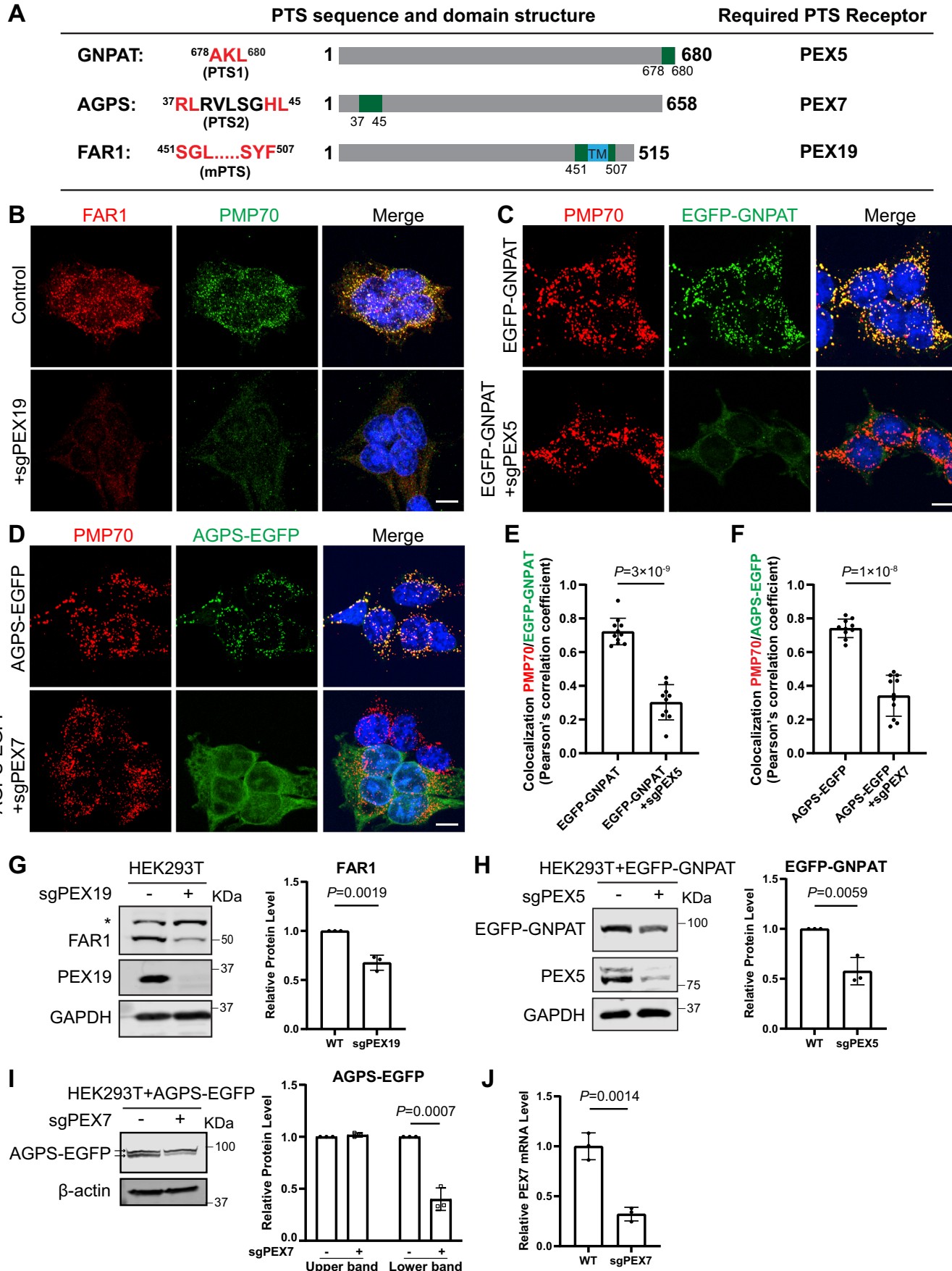

**Figure EV3.   Peroxisomal targeting of GNPAT, AGPS, and FAR1 depends on PEX5, PEX7, and PEX19, respectively. Related to Figure 2.**

(A) Schematic representations of GNPAT, AGPS, and FAR1 proteins. Peroxisomal targeting sequences and their positions within each protein are highlighted in red and green. The corresponding peroxisomal-targeting receptors are also indicated. TM: transmembrane domain. (B) Immunofluorescence images of endogenous FAR1 and PMP70 in WT and *PEX19* KO HEK293T cells. Scale bar: 10 μm. (C) Localization of EGFP-GNPAT and endogenous PMP70 in WT and *PEX5* KO HEK293T cells. Scale bar: 10 μm. (D) Localization of AGPS-EGFP and endogenous PMP70 in WT and *PEX7* KO HEK293T cells. Scale bar: 10 μm. (E, F) Pearson's coefficient analysis of co-localization between EGFP-GNPAT and PMP70 in (C), and between AGPS-EGFP and PMP70 in (D). Data were presented as mean ± s.d.; $n = 10$ ROIs from three biological replicates. *P* values were calculated by a two-tailed unpaired t-test. (G) Western blot analysis for endogenous FAR1 in WT and *PEX19* KO HEK293T cells, with quantification. * Indicates a non-specific band. Data were presented as mean ± s.d.; $n = 3$ biological replicates. *P* values were calculated by a two-tailed unpaired t-test. (H) Western blot analysis of EGFP-GNPAT in WT and *PEX5* KO HEK293T cells, probed with a GFP antibody. Data were presented as mean ± s.d., $n = 3$ biological replicates. *P* values were calculated using a two-tailed unpaired t-test. (I) Western blot analysis of AGPS-EGFP in WT and *PEX7* KO HEK293T cells, probed with a GFP antibody. The upper and lower bands of AGPS-EGFP are indicated by arrows. Data were presented as mean ± s.d., $n = 3$ biological replicates. *P* values were calculated using a two-tailed unpaired t-test. (J) RT-qPCR analysis confirms the reduction of *PEX7* mRNA in *PEX7* KO cells. Data were presented as mean ± s.d.; $n = 3$ biological replicates; *P* values were calculated by a two-tailed unpaired t-test. For (B–D), these experiments were repeated three times with similar results. Source data are available online for this figure.

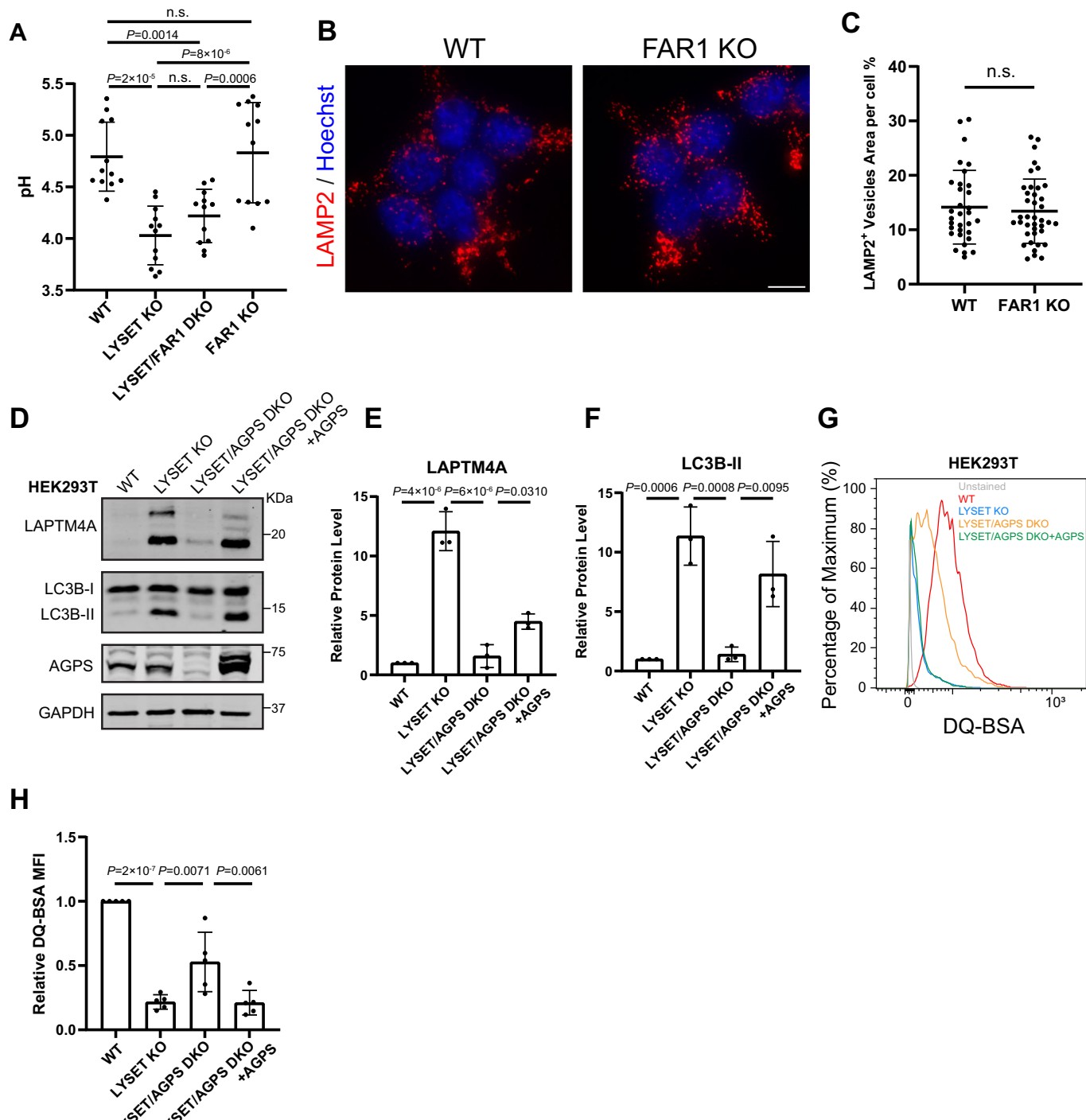

**Figure EV4. Effects of ether lipid deficiency on lysosomal pH, number, and protein degradation. Related to Figure 3.**

(A) Lysosomal pH in WT, *LYSET* KO, *LYSET/FAR1* DKO, and *FAR1* KO HEK293T cells. Cells are stably transfected with the FIRE-pHly (a ratiometric pH biosensor), and the signal intensities of mTFP and mCherry were measured by flow cytometry. Lysosomal pH was calculated from the ratio of mTFP to mCherry signals. Data were presented as mean ± s.d.; $n = 12$ from three biological replicates; $P$ values were calculated by one-way ANOVA with multiple comparisons. (B, C) LAMP2 immunostaining (B) and quantification (C) in WT and *FAR1* KO HEK293T cells. Data were presented as mean ± s.d.; $n = 32$ cells from three biological replicates; $P$ values were calculated by two-tailed unpaired t-test. Scale bar: 10 μm. (D) Steady-state levels of LAPTM4A and LC3B-II in WT, *LYSET* KO, *LYSET/AGPS* DKO, and *LYSET/AGPS* DKO cells complemented with AGPS overexpression. (E, F) Quantification of full-length LAPTM4A and LC3B-II levels in (D). Data were presented as mean ± s.d.; $n = 3$ biological replicates; $P$ values were calculated by one-way ANOVA with multiple comparisons. (G) DQ-BSA assay in WT, *LYSET* KO, *LYSET/AGPS* DKO cells, and *LYSET/AGPS* DKO cells complemented with AGPS overexpression. (H) Normalized MFI in (G). Data were presented as mean ± s.d.; $n = 5$ biological replicates; $P$ values were calculated by one-way ANOVA with multiple comparisons. Source data are available online for this figure.

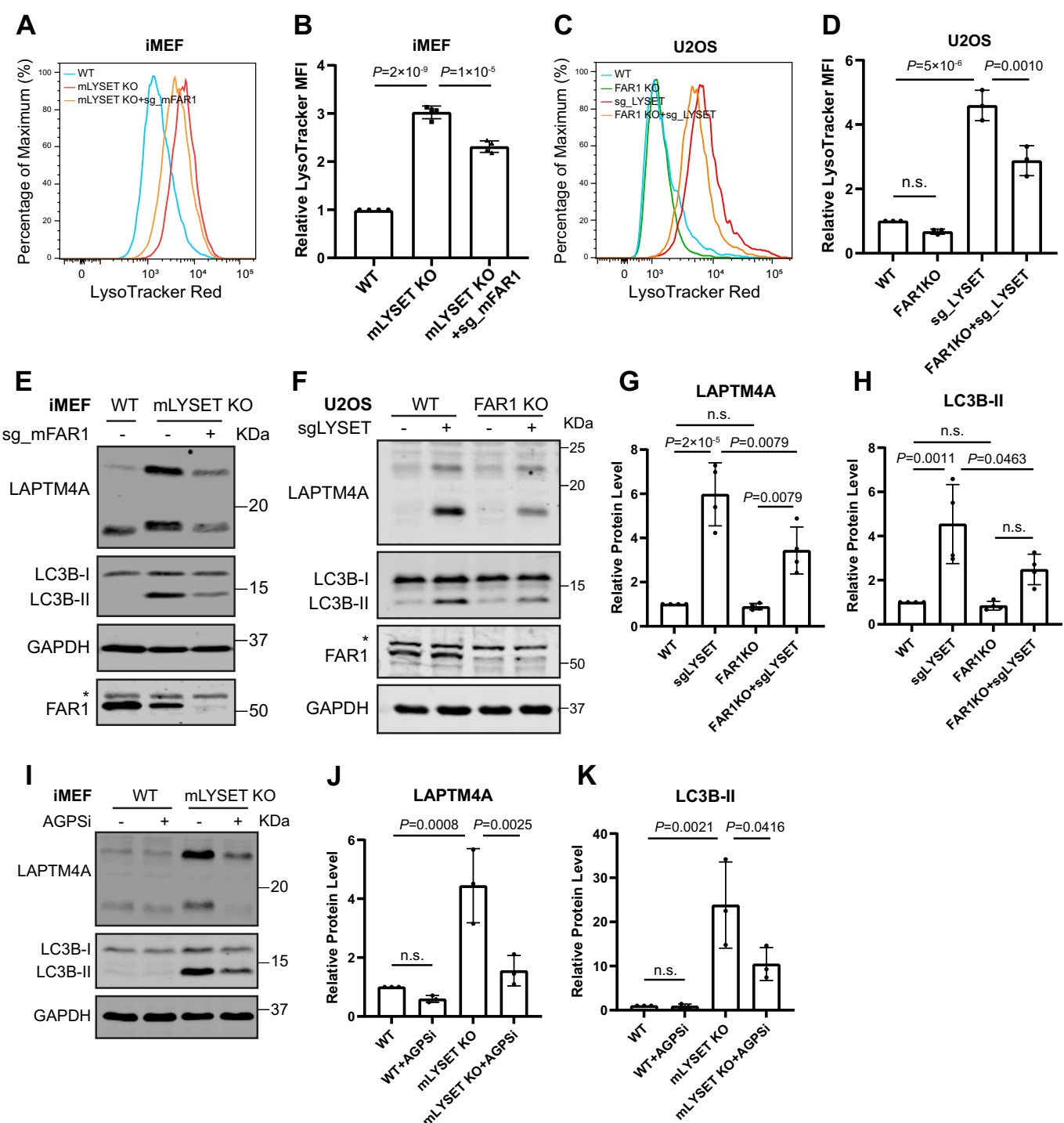

**Figure EV5. Conserved effects of the ether lipid synthesis pathway across species and cell lines. Related to Figure 3.**

(A) LysoTracker Red intensity in WT, *LYSET* KO, and *LYSET/FAR1* DKO iMEF cells. (B) Normalized MFI in (A). Data were presented as mean ± s.d.; $n = 4$ biological replicates; *P* values were calculated by one-way ANOVA with multiple comparisons. (C) LysoTracker Red intensity in WT, *FAR1* KO, *LYSET* KO, and *LYSET/FAR1* DKO U2OS cells. (D) Normalized MFI in (C). Data were presented as mean ± s.d.; $n = 3$ biological replicates; *P* values were calculated by one-way ANOVA with multiple comparisons. (E) Steady-state levels of LAPTM4A and LC3B-II in WT, *LYSET* KO, and *LYSET* KO iMEF cells transfected with FAR1 guide RNA. * Indicates a non-specific band. The experiments were repeated three times with similar results. (F) Steady-state levels of LAPTM4A and LC3B-II in WT and *FAR1* KO U2OS cells transfected with or without LYSET guide RNA. * Indicates a non-specific band. (G, H) Quantification of total LAPTM4A (G) and LC3B-II (H) levels in (F). Data were presented as mean ± s.d.; $n = 4$ biological replicates; *P* values were calculated by one-way ANOVA with multiple comparisons. (I) Steady-state levels of LAPTM4A and LC3B-II in WT, *LYSET* KO iMEF cells treated with or without 250 μM AGPS inhibitor for 48 h. (J, K) Quantification of total LAPTM4A (J) and LC3B-II (K) levels in (I), Data were presented as mean ± s.d.; $n = 3$ biological replicates; *P* values were calculated by one-way ANOVA with multiple comparisons. Source data are available online for this figure.

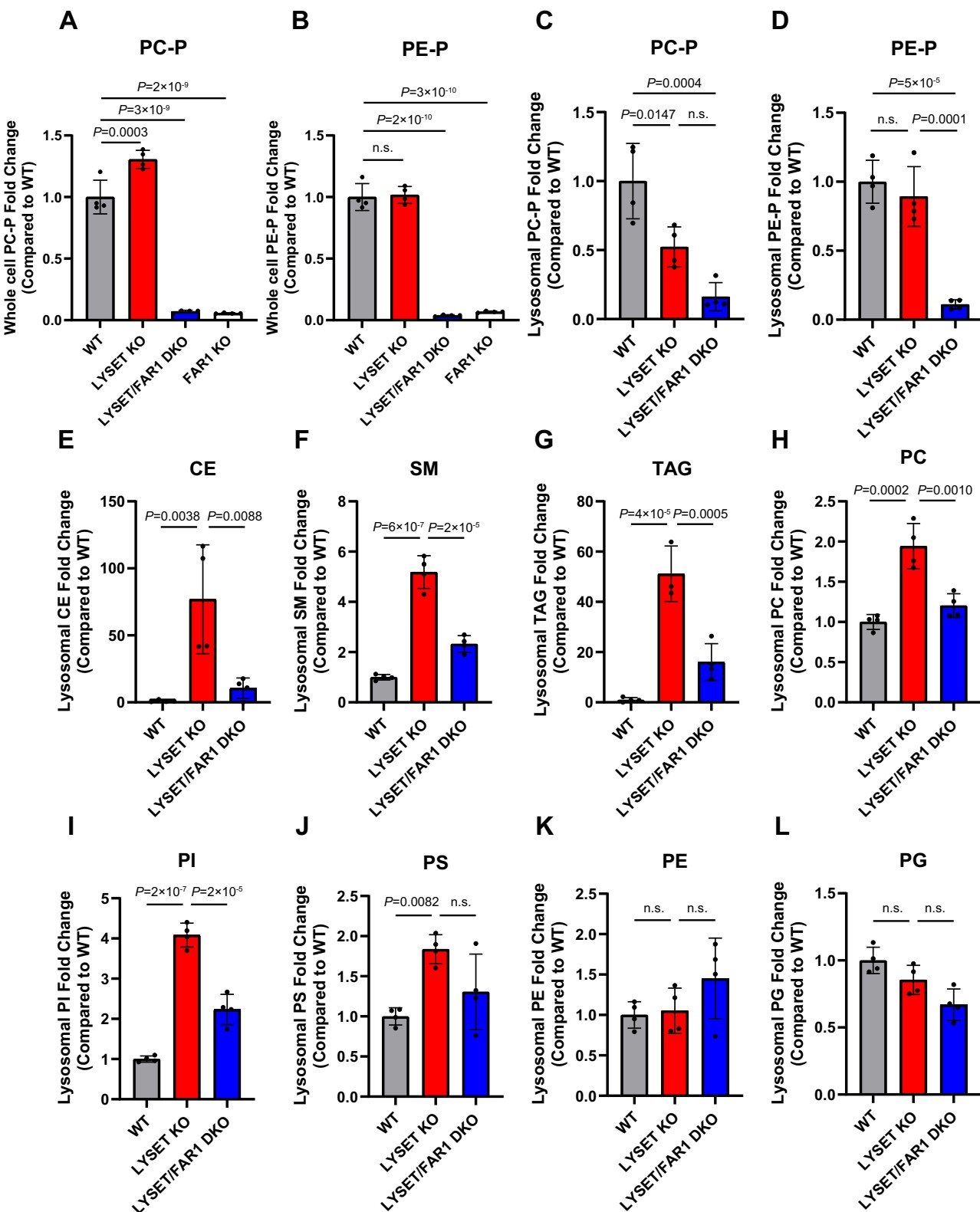

◀ **Figure EV6. Lysosomal lipid accumulation is reduced in *LYSET/FAR1* DKO cells. Related to Figure 5.**

(A, B) Whole-cell plasmenyl-phosphatidylcholine (PC-P) and plasmenyl- phosphatidylethanolamine (PE-P) levels in WT, *LYSET* KO, *LYSET/FAR1* DKO, and *FAR1* KO HEK293T cells. Fold changes were normalized to WT levels. Data were presented as mean ± s.d.; $n = 4$ biological replicates; $P$ values were calculated by one-way ANOVA with multiple comparisons. (C–L) Relative levels of plasmenyl-phosphatidylcholine (PC-P), plasmenyl-phosphatidylethanolamine (PE-P), cholesteryl esters (CE), sphingomyelin (SM), triacylglycerols (TAG), phosphatidylcholine (PC), phosphatidylinositol (PI), phosphatidylserine (PS), phosphatidylethanolamine (PE), and phosphatidylglycerol (PG) in purified lysosomes from WT, *LYSET* KO, and *LYSET/FAR1* DKO HEK293T cells. Fold changes were normalized to WT levels. Data were presented as mean ± s.d.; $n = 4$ biological replicates; $P$ values were calculated by one-way ANOVA with multiple comparisons. Source data are available online for this figure.

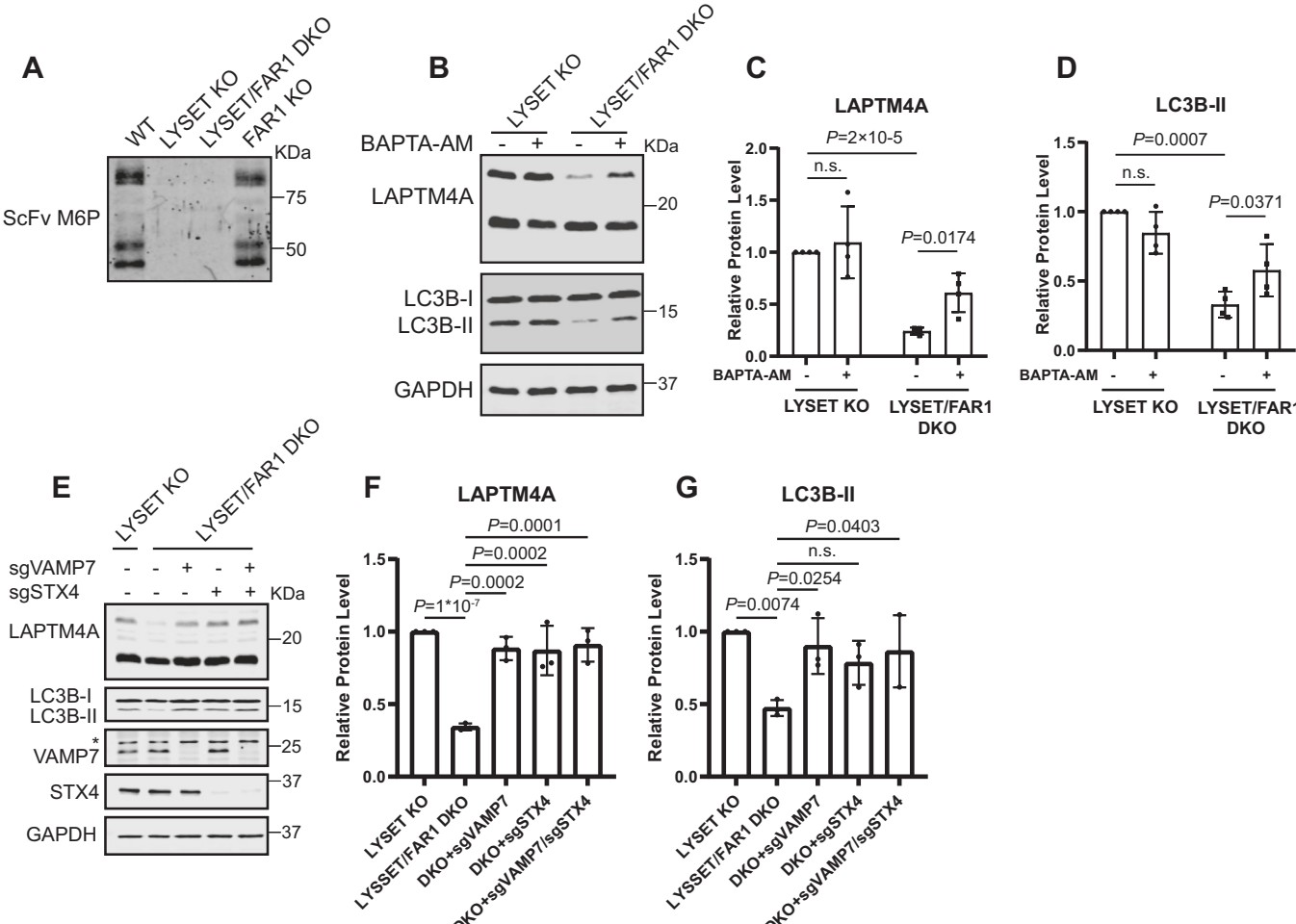

**Figure EV7. Inhibition of lysosomal exocytosis restores LAPTM4A and LC3B-II levels in *LYSET/FAR1* DKO cells. Related to Figure 7.**

(A) Western blot analysis of the M6P modifications in purified lysosomes from the indicated cell lines, detected by a single-chain antibody against M6P (scFv M6P). (B) Western blot analysis of LAPTM4A and LC3B-II in *LYSET* KO and *LYSET/FAR1* DKO HEK293T cells treated with or without 10 µM BAPTA-AM for 48 h. (C, D) Normalized protein levels of full-length LAPTM4A and LC3B-II in (B). Data were presented as mean ± s.d.; n = 4 biological replicates; P values were calculated by two-tailed paired t-test. (E) Western blot analysis of LAPTM4A and LC3B-II in *LYSET* KO, *LYSET/FAR1* DKO, *LYSET/FAR1/VAMP7* TKO, *LYSET/FAR1/STX4* TKO, and *LYSET/FAR1/VAMP7/ STX4* QKO HEK293T cells. * Indicates a non-specific band. (F, G) Normalized protein levels of full-length LAPTM4A and LC3B-II in (E), Data were presented as mean ± s.d.; n = 3 biological replicates; P values were calculated by one-way ANOVA with multiple comparisons. Source data are available online for this figure.

