## [Peer Review File · The EMBO Journal]

Peroxisome-derived ether lipids regulate lysosomal exocytosis

Liang Chen, Danielle Henn, Zhongzheng Dong, Jiaxuan Liang, Aleksander Wielenga, Goncalo Vale, Bala Burugula, Junyi Zou, Yamuna Krishnan, Jeffrey McDonald, Jacob Kitzman, and Ming Li

Corresponding author(s): Ming Li (mlium@umich.edu)

Review Timeline:

Submission Date:	18th Nov 25
Editorial Decision:	15th Jan 26
Revision Received:	26th Mar 26
Accepted:	15th Apr 26

Editor: William Teale

Transaction Report:

This manuscript was transferred to The EMBO JOURNAL following peer review at another journal.

Reviewer #1:

The study by Liang Cheng shows a new connection between peroxisomes and lysosomes executed by ether-bound glycerolipids. The findings in the manuscript are novel and represent a significant advance in the field but the focus and background knowledge lies clearly on the side of lysosomes, while accuracy in the field of peroxisomes and ether lipids is missing. There is a clear need to render the paper introduction and discussion more general and add specific literature, also a more detailed analysis of ether lipid metabolism is needed.

Major points:

- Plasmalogens are a subclass of ether lipids and since the discovery of PEDS1 well distinguishable from their alkyl-counterparts. It is questionable why the authors claim that the crosstalk between lysosomes and peroxisomes is due to plasmalogens even though PEDS1 is never mentioned in the paper, much less investigated. In order to deepen the findings and make them valuable for the lipid community, authors must include data on PEDS1 where accessible to show the potential impact of plasmalogen synthesis and clearly distinguish them from alkyl ether lipids.

We thank the reviewer for raising this important point and for the constructive suggestions. To address this point, we generated two independent HEK293T lines carrying double knockouts of LYSET and PEDS1 (TMEM189), which would lack plasmalogens but carry higher levels of the other main class of ether lipids (plasmalogen lipids). In both LYSET/TMEM189 DKO cell lines, we did not observe a reduction in LysoTracker Red intensity (new Figure 3M-N), in contrast to the drastic reduction observed in LYSET/FAR1 DKO, which lacks all ether lipids. Based on these results, we concluded that the observed lysosomal phenotypes are primarily driven by ether lipids broadly, rather than plasmalogens specifically. Accordingly, we have revised the conclusions throughout the manuscript to reflect this distinction.

This finding also explains why PEDS1 (TMEM189), unlike FAR1, AGPS, and GNPAT, was not identified in our screen. We believe this clarification strengthens our study and provides important insight for the lipid biology community. We are grateful to the reviewer for prompting this critical analysis.

- Already the first sentence of the abstract does not acknowledge the literature in the field. It was clearly shown in 2022 that there exists a link between peroxisomes and lysosomes, namely through cholesteryl ester (PMID: 25860611 which is further highlighted in PMID: 25860602).

Thank you for pointing this out. We have rewritten the abstract to ensure that it does not misrepresent prior work in this area, and we are now citing the paper the reviewer mentioned in the lipidomics section of our Results (lines 308-311) to help put our work into context.

This continues in the introduction, that does not introduce the reader sufficiently to the field. For example, some paragraphs of the results would be much better suited in the introduction as important information is contained there (first two paragraphs of the results and lines 102-107, also add references in this part). Furthermore, there is a list of 28 papers when searching for plasmalogen AND lysosome in Pubmed which show among other findings that lyso- and plasmalogen-phosphatidylethanolamine were increased in Gaucher and Parkinson's (PMID: 36142296), in Sandhoff disease (SD), another lysosomal disease decreased concentrations in lipids typical of the myelin sheath, galactosylceramides and plasmalogen-phosphatidylethanolamines were found, suggesting that reduced synthesis of myelin lipids is an early event in the development of disease pathology (PMID: 33396723), and when the GBA gene encoding for the lysosomal enzyme glucocerebrosidase was mutated, plasmalogen phosphatidylethanolamine (PEp) were decreased (PMID: 28890071). None of these references are mentioned in the manuscript.

We thank the reviewer for this helpful suggestion. In the revised manuscript, we have added references to prior studies linking plasmalogens with lysosomal function and lysosomal storage disorders, including Gaucher disease², Sandhoff disease³, and Parkinson's disease⁴. We have also moved some of the information from the first Results subsection, as the reviewer suggested.

- In the results section it is not always clear what cells were used and why (HEK versus HeLa), which methods were employed (e.g. for FAR knockout in line 189), why certain inhibitors were used (e.g. Torin-1) and which markers for example are specific for which organelle (line 257). This lack of important details continues also in the figure legends, which are really only a list of titles, with no n numbers, experimental details given and give the reader no possibility to understand and rate the experimental evidence.

Thank you for pointing all these out:

- 1) Cell lines are now clearly specified in their respective figure legends.
- 2) Torin-1 is described in the legend for Figure S1.
- 3) Organelle markers in Figure 5B are now directly labeled.
- 4) n numbers, the statistical analysis method, and exact p-values are either labeled on the figures or described in the legends.
- 5) We've also added more experimental details in the figure legends—additional information is available in the Materials & Methods section.

- CrispR cell lines need validation: it is standard to use more than one sgRNA to exclude off target effects, genomic DNA sequencing results should give details about the exact indels present in the individual clones.

For most of our CRISPR knockout cell lines, gene disruption was validated at the protein level by western blot analysis. To rule out off-target effects, we also performed complementation assays for the LYSET/FAR1 DKO, GNPTAB/FAR1 DKO, and LYSET/AGPS DKO cells (see new Figure 3A–L and Figure S5L–H).

For targets lacking reliable antibodies, we used alternative validation methods. For example, PEX7 polyclonal knockout cell lines were validated by RT-qPCR (Figure S3E) and by measuring the reduction in its transport substrate, AGPS (Figure S3D). In the case of *LYSET/TMEM189* DKO lines (#1 and #2), validation was done by sequencing the sgRNA target regions (see Supporting Figure 1 below).

Supporting Figure 1: Sequencing results of the sgRNA target regions in two independent LYSET/TMEM189 DKO HEK293T cell lines.

- The discussion must contain a broader outlook on what these finding might mean for RCDP patients and whether lysosomal function changes have been found in the patients. The authors should discuss what the current idea is about what will happen to the lysosomal cargo after the proposed exocytosis. Work on ether lipids and their impact on membrane processes like endocytosis is again not acknowledged (PMID: 38562716).

We thank the reviewer for these valuable suggestions. In response, we have added the last paragraph to the Discussion section addressing the potential implications of our findings for explaining the bone phenotypes observed in RCDP patients (lines 429-437). In addition, we have cited several papers on the known roles of ether lipids (lines 397-400), which suggest intriguing mechanistic hypotheses for their effect on lysosomal exocytosis that require further investigation.

Minor points:

- Many errors in the use of capital and small initial letters, spelling mistakes (e.g. Gauche line 87)

We have thoroughly edited the text to address these issues.

- Unexplained abbreviations (e.g. LSD in the abstract, HDG etc)

Thank you for pointing this out. We have now spelled out abbreviations throughout the manuscript.

- GNPTAB-deficiency represents ML type II and III alpha/beta and ref is missing.

We have added references.

- lysosomal upregulation (line 98), do you mean an increased number of lysosomes?

Yes, sorry for the confusion. We have replaced the term “lysosomal upregulation” with “increase in lysosome numbers” and revised the wording accordingly throughout the manuscript.

- line 111: TKO is never used in figures, use consistent nomenclature of cell lines.

TKO stands for “triple knock out”; it is described in the main text. We also specified the genotypes in the legends of Figures 1F and 1I.

- Antibody against M6P was prepared by the authors (statement in antibody validation in the reporting summary states ALL are commercially available).

We apologize for the confusion. The scFv M6P antibody is commercially available from the Geneva Antibody Facility at the University of Geneva (Cat#: AG949, <https://abcd-antibodies.com/products/anti-m6p-antibody>). In our experience, they also distribute the plasmid encoding this antibody along with a purification protocol.

- Why did the authors use an eGFP-fusion for GNPAT but not for AGPS, if the anti-AGPS antibody was not usable in immunohistochemistry?

We have now generated a stable AGPS-EGFP cell line. As shown in Figure S3H, knocking out PEX7 abolished its colocalization with PMP70.

- Table S2 has to be formatted in a way that cells are completely readable and searchable.

As submitted, Table S2 is a standard Excel file with four clearly organized sheets and is both readable and searchable on our end. It is possible that the table may not have displayed correctly in the reviewer’s file package they received from the journal. We are happy to coordinate with journal staff to make any necessary formatting adjustments to ensure that the table is readable and searchable upon publication.

- Table S3 only contains TAG not all elevated lipids (lines 279-283).

The updated Table S3 contains 10 pages, each listing different lipids identified in our analysis. The pages are organized by lipid class as follows: CE (page 1), TAG (page 2), SM (page 3), PC (page 4), PS (page 5), PG (page 6), PI (page 7), PE (page 8), PC-P (page 9), and PE-P (page 10).

- For all figures/legends: Quantification of Western blots and immunostainings should be given, used methods and antibody/reagents briefly stated in the legends. n numbers, must

be given for each experiment, as well as statistics and in case of cell pictures/blots etc it must be clear that one representative image of (how many?) is shown.

- Fig. 1: panel I, which DKO clone is shown?

- Fig. 2: What's shown in red and what in green, what in blue?

- Fig. S3: red, blue, green in panel A not explained

We have addressed the reviewer's comments as follows:

- 1) We have added the n numbers and statistical information to the figure legends for all relevant experiments.
- 2) In Figure 1I, the specific DKO clone number has now been indicated.
- 3) In the legend for Figure 2E, we have added the following clarification: "Peroxisomal proteins are highlighted in red (enzymes involved in ether lipid synthesis) or green (peroxins)."
- 4) For Figure S3, we have included the following explanation in the legend: "Peroxisomal targeting signal sequences and locations within each protein are highlighted in red and green. Associated receptors for their peroxisomal targeting are listed respectively. TM: transmembrane domain."

-Methods: It should be mentioned in the Western blot section which blocking buffer was used for the individual antibodies. From line 740 on the writing style is suddenly changed and methods sound like instructions. Companies must have city and country. Where was nigericin purchased?

We have made the following modifications regarding the blocking buffer and nigericin:

- 1) "Membranes were blocked in 5% milk or 5% BSA (for phosphorylation blot analysis) in TBST (20 mM Tris, pH 7.5, 150 mM NaCl, 0.1% Tween-20)."
- 2) Nigericin was used at a final concentration of 10 μ M (Cat# J61349, Thermo Scientific, Waltham, MA, USA).

Additionally, we have provided more detailed information for chemicals in the Methods section, including the city and country of origin for all companies, as requested.

- Lipidomics: It is stated that plasmalogens were analyzed; however, no measurements were found in the documents. How exactly were plasmalogens and ether lipids measured, and where are the data?

We have added plasmalogen quantification data from both whole-cell and lysosomal fractions in the new Figure S6. Detailed information on individual plasmalogen species is provided in the updated Supplementary Tables 3 and 4.

Comment on statistics:

Accuracy of error bars and probability values could not be judged as n numbers are missing.

We have included the n numbers in the figure legends and directly labeled the p-values on the figures.

Reviewer #2:

In this manuscript by Chen et al. the authors identify a potential link between lysosomes and peroxisomes while trying to determine the molecular pathway responsible for lysosomal expansion in cells with dysfunctional lysosomes. Using a CRISPR/Cas9 genome-wide screen in LYSET-deficient cells, the researchers identified plasmalogens as critical regulators of lysosomal function. They discovered that disrupting plasmalogen biosynthesis restored lysosomal clearance of undigested materials by enhancing lysosomal exocytosis, independent of the traditional mannose-6-phosphate pathway. These findings provide potential interesting insights into mechanisms that regulate lysosomal exocytosis and suggest that targeting plasmalogen metabolism could be a promising therapeutic strategy for lysosomal storage disorders. While this study describes an interesting and previously unexplored relationship between plasmalogen synthesis and LSD, it is crucially lacking quantitative analysis of plasmalogens both at whole cell and lysosome level. The authors repeatedly implicate “plasmalogen deficiency”, including in the title of the manuscript, but fail to provide any analysis of their levels in FAR1 KO cells nor WT controls. Furthermore, the authors fail to show, or even speculate on the mechanistic role of plasmalogens. The current title massively overstates the impact of the study, as ablating enzymes normally localized to the peroxisome and having an effect in the lysosome does not confer “crosstalk”. Thus, while we believe that the finding is interesting, we think that the role of plasmalogen as a lipid in regulating lysosome biology is completely lacking. One hypothesis that can explain the beneficial effects of targeting plasmalogen synthesis enzymes is substrate reduction, especially if these lipids accumulate in the lysosome in disease state, which is contrary to a direct crosstalk hypothesis. Following are some of our comments on the study.

1) “Crosstalk” stated in the title is never shown by the authors. In addition, the authors should show “plasmalogen deficiency” as currently not shown in the manuscript.

We thank the reviewer for bringing this point to our attention. In response, we have revised the title to: “Peroxisome-derived ether lipids regulate lysosomal homeostasis through exocytosis control.”

2) The authors claim throughout the paper that the effect is mediated by “plasmalogen deficiency”. However, there is no validation that plasmalogen levels are actually increasing or decreasing in LYSET KO and how it responds in DKO. The authors show that enzymes involved in synthesis are implicated, but fail to show any quantification of plasmalogen levels. As the manuscript currently stands, all mention of “plasmalogen deficiency” should be removed unless it is shown that levels are actually changed. Figure S7 shows that the authors have access to LC/MS, so the lack of data showing changes to plasmalogen levels in FAR1 KO cells is puzzling. Without this data, the manuscript seems incomplete.

Thank you for raising this important point. We have added lipidomics for ether lipid quantification from both lysosomal and whole-cell levels to the new Figure S6. As shown, knocking out *FAR1* abolishes ether lipid levels. Detailed information on ether lipid species is provided in the updated Supplementary Tables 3 and 4.

3) A major limitation of the FACS-based CRISPR screen employed is that there are no LYSET-WT control lines to compare final sgRNA frequencies to. This makes it impossible to know whether plasmalogen-related biosynthesis is essential under only lysosomal stress or if it is also important under steady state. The authors should KO FAR1 in cells to see if the decrease in LAMP immunoreactivity is an effect only under lysosomal stress, or if plasmalogen synthesis machinery induces lysosomal exocytosis at baseline.

Our original CRISPR screen was specifically designed to identify suppressor mutants in the *LYSET* KO background; therefore, we did not perform the screen in WT cells. We have now revised the text when introducing the screen setup to make this point more clear.

In response to the reviewer's question, we examined LAMP2 staining in *FAR1* KO cells (new Figure S4B–C) and found no significant difference in LAMP2 signal between WT and *FAR1* KO cells. This supports the idea that the reduction in lysosome number occurs only in the context of lysosomal storage disease backgrounds.

4) Please show FAR1 colocalization to PMP70 in fig S3D.

Done – please check the updated Figure S3F. In WT cells, colocalization of *FAR1* and *PMP70* was observed. In *PEX19* knockout cells, peroxisomal compartments were lost, and both *PMP70* and *FAR1* appeared diffuse in the cytoplasm. A similar observation for *PMP70* localization in the absence of *PEX19* was also reported by Korotkova et al ⁵.

5) In figure 3M, in order to substantiate the claim that HDG supplementation restores lysotracker signal in a “dose dependent manner”, please supply statistical comparison between 25uM and 50uM conditions.

We have expanded the HDG analysis into a new Figure 6. The original Figure 3M is now presented as Figure 6B, with the corresponding quantification and statistical analysis shown in Figure 6C.

6) The authors claim that “knocking out plasmalogen synthesis genes reduced lysosome numbers to near WT levels in M6P-deficient cells”. However, this is only demonstrated in one cell line, and only a small reduction is seen in MEFs. Either clarify that this effect is only in HEK293Ts, demonstrate this in other human cell lines, or remove the statement. Further, some in vivo validation that targeting plasmalogen biosynthesis (perhaps with the AGPS small molecule inhibitor) can alleviate LSD phenotypes would be useful.

We thank the reviewer for raising these points. We also tested the effect of knocking out *FAR1* on lysosome reduction in U2OS cells. As shown in Figure S5C-D, the LysoTracker intensity was also significantly reduced in *FAR1/LYSET* DKO compared to *LYSET* KO U2OS cells. The effect strength might vary depending on cell types, and we have added the text “A similar reduction in lysosome number following *FAR1* disruption was also observed in *LYSET* KO iMEFs and U2OS cells” (lines 216-218)

We wish we could include in vivo assays, such as studies in LSD mouse models. However, generating the necessary mouse lines will take years, and we are not there yet. It already took us five years of work to bring this manuscript to completion. Validation of this idea in vivo is certainly the next step we plan to pursue, but that will form the basis of a new manuscript in a few years’ time.

7) The authors showed that lysosomal upregulation in LYSET deficient cells is independent of mTOR and TFEB, however it remains possible that disruption of plasmalogen synthesis somehow alters these pathways. Please test whether mTOR or TFEB is altered in LYSET/FAR1 DKO cells.

As requested, we assessed the phosphorylation status of two mTORC1 substrates, S6K and 4EBP1, in WT, *LYSET* KO, and *LYSET/FAR1* DKO HEK293T cells (Supporting Figure 2A), as a proxy for mTOR pathway activity. Both substrates exhibited phosphorylation levels comparable to those in WT cells. Additionally, TFEB localization remained cytosolic across all three cell lines (Supporting Figure 2B). Consistent with these observations, our RNA-seq analysis revealed no significant changes in the transcription of CLEAR network genes (Supporting Figure 2C). Altogether, these results support our initial conclusions that this effect is not acting through mTORC1 or TFEB.

Supporting Figure 2: mTORC1 and TFEB activity in LYSET/FAR1 DKO cells. **(A)** Western blot analysis of mTORC1 substrates. **(B)** TFEB-GFP localization in WT, LYSET KO, and LYSET/FAR1 DKO HEK293T cells. Scale bar: 10 μ m. **(C)** Fold-change and p-values of selected CLEAR network gene transcription levels in LYSET/FAR1 DKO versus LYSET KO HEK293T cells

8) More should be shown to suggest that increase in EVs is due to lysosomal exocytosis. In LYSET WT cells, it would be beneficial to show that FAR1 KO increases lysosomal enzymes in the conditioned media. In addition, it would be nice to show if in LYSET/FAR1 DKO cells, there is an increase in lysosomal membrane proteins at the plasma membrane (LAMP1/2). Seeing if these changes are then reversed by blocking TRPML1 function would be convincing.

We thank the reviewer for the helpful suggestions. In response, we have now performed additional experiments that further support our lysosomal exocytosis hypothesis: (1) detection of mature Cathepsin D (mCTSD) secretion in conditioned media, and (2) increased secretion of lysosomal intraluminal vesicle (ILV) cargos, including LAPTM4A-EGFP and EGFP-RNF152, in the 100,000 \times g extracellular vesicle pellet (updated Figure 7C–H). These observed increases in secreted mature lysosomal enzymes and

digested cargo in the extracellular media or extracellular vesicles provide strong evidence supporting enhanced lysosomal exocytosis in FAR1 KO cells.

Reviewer #3 (Remarks to the Author):

Summary of Key Results:

The authors set out to determine factors that can reduce the number of lysosomes in cells with mutations that cause lysosomal storage disorders. Through a CRISPR KO screen, they identify genes enriched in peroxisome formation and ether lipid biosynthesis that reduce lysosome staining. The authors go on to show that impairing plasmalogen synthesis increases lysosome exocytosis, suggesting that ether lipids repress lysosome exocytosis in cells with mutations causing lysosomal storage disorders. This finding suggests novel approaches to the treatment of lysosomal storage diseases. The authors could extend the impact of this paper by further exploring the effect of ether lipids on lysosome exocytosis in WT cells. I found this paper to be clear and well written, and very interesting and broadly impactful for scientists interested in lysosomes, peroxisomes, lipids, and unconventional secretion.

Thank you so much for your kind words.

Claims:

1) Disrupting plasmalogen synthesis reduces lysosomal accumulation of undigested material and increases clearance. This claim is well supported by the data.

Thank you!

2) Clearance is through increased exocytosis. This claim is well supported by the data, though the authors also suggest it is through increased digestive capacity, which is less clearly supported.

Thank you for pointing this out. While Figures 4C–D show that LAPT4A and LC3B-II have faster rates of disappearance, this could suggest either enhanced lysosomal digestion or a faster rate of exocytosis used to clear accumulated substrates. However, the DQ-BSA analysis in Figures 4E–F and 4K–L supports improved digestive capacity. Since DQ-BSA must be endocytosed and cleaved by lysosomal enzymes to generate fluorescence, the observed increase in DQ-BSA signal, despite a potentially increased rate of exocytosis, indicates that more active digestion is occurring. This enhancement is best explained by an actual improvement in lysosomal enzymatic function rather than exocytosis alone.

3) This works with genetic or pharmacological inhibition of plasmalogen synthesis. This claim is well supported.

Thank you!

4) *Unexpected link between peroxisome derived lipids and lysosomes. I agree that this link is unexpected.*

Thank you!

Originality and Significance:

This paper is the first to show a connection of plasmalogen synthesis to lysosome exocytosis, and thus it is an important finding for understanding treatment options and disease progressions in LSDs. However, it is not the first to show that lipids can regulate lysosome exocytosis as high levels of cholesterol can alter the SNARE VAMP7's localization (Fraldi et al 2010) and sphingomyelin has been shown to inhibit TRPML1 channel activity and TRPML1 mediated lysosomal calcium release (Shen et al Nature Comm 2012). Since the authors show that plasmalogen deficiencies reduce the accumulation of cholesteryl esters and sphingomyelin, they could take it a step further and test whether the cholesterol and sphingomyelin change is sufficient for the observed changes in lysosomal exocytosis.

We thank the reviewer for suggesting these papers. The study by Fraldi et al. concluded that free cholesterol accumulation on endolysosomal membranes can inhibit lysosomal fusion with other membranes. However, in our experiments, free cholesterol levels remained nearly unchanged among WT, LYSET KO, and LYSET/FAR1 DKO cells. In contrast, cholesteryl esters (CE) showed dramatic changes (see our response to Reviewer #3 and Supporting Figure 7). These observations suggest that free cholesterol is unlikely to contribute significantly to the lysosomal phenotypes observed in our cell lines.

In addition, Shen et al. demonstrated that sphingomyelin (SM) accumulation within lysosomes can inhibit TRPML1-mediated Ca^{2+} release. In our analysis, we also observed SM accumulation in LYSET KO cells compared to wild type, and a reduction of SM levels in LYSET/FAR1 double knockout (DKO) cells relative to LYSET KO (new Figure S6F). However, we believe that changes in SM alone are insufficient to explain the observed alterations in lysosomal exocytosis. If SM were the sole determinant, the exocytosis level in LYSET/FAR1 DKO cells would be expected to return to wild-type levels. Instead, as shown in new Figure 7B, exocytosis in the DKO cells is significantly higher than in wild type, suggesting that ether lipids exert additional regulatory effects on lysosomal exocytosis beyond their influence on SM composition.

Major Comments:

1) *The authors need to be more precise with their language when discussing lysosome biogenesis. For example:*

Line 33, Line 124 – it is inaccurate to say that plasmalogens regulate “lysosomal

biogenesis”when they regulate exocytosis.

Lines 85 and 98-“lysosomal upregulation”and“increased lysosomal biogenesis”implies transcriptional mechanisms that increase synthesis. The data discussed shows increased abundance of lysosomes, which could be through biogenesis, but also could be through other mechanisms, as the authors show. The authors should be precise with their description of increased lysosome levels.

Line 186-187-“Plasmalogen as a Negative Regulator of Lysosome Biogenesis”

We thank the reviewer for the helpful comments regarding the use of the term “lysosome biogenesis.” To avoid confusion, we have revised the wording throughout the manuscript. The title has been updated to “Peroxisome-derived ether lipids regulate lysosomal homeostasis through exocytosis control.” Additionally, we have replaced “lysosome biogenesis” with more precise terms such as “lysosome numbers” or “lysosome quantity” where appropriate.

2) The authors discuss two mechanisms by which plasmalogen deficiency reduces the observed accumulation of lysosome substrates: increased lysosomal digestive capacity and increased lysosome exocytosis. It seems contradictory that plasmalogen deficiency could increase lysosomal digestive capacity without restoring lumenal enzyme levels, and therefore the authors should address whether the observed increased digestive capacity is due to an increase in exocytosis.

Thank you very much for raising these excellent questions. Indeed, if the M6P pathway is not restored by deleting FAR1 or AGPS, how can lysosomal digestion capacity be improved? This was a question that puzzled us for some time, until we saw the data presented in Figure 5E, which shows SILAC-based proteomic analysis comparing purified lysosomes from LYSET KO and WT cells.

As you can see, while most luminal enzymes are reduced in LYSET KO lysosomes, we found that the protein levels of 14 enzymes remained largely unchanged, and 5 enzymes were even increased compared to WT (Figure 5E, the red dots in the middle or the right section, and Supporting Figure 3 below). This suggests that a subset of lysosomal enzymes can still be delivered via M6P-independent pathways. For example, cathepsin D (CTSD) and acid alpha-glucosidase (GAA) remain abundant in purified lysosomes from LYSET KO cells (Supporting Figure 4), although their processing into mature forms is severely impaired.

Furthermore, in LYSET/FAR1 DKO cells, we observed significantly improved maturation of both pro-CTSD and pro-GAA, supporting our hypothesis that lysosomal digestion capacity is indeed improved in the DKO context.

Why does this improved processing occur in LYSET/FAR1 DKO cells? We propose that the accumulation of undigested materials in LYSET KO lysosomes may inhibit residual enzyme activity, blocking further processing and activation of M6P-independent enzymes. In LYSET/FAR1 DKO cells, increased lysosomal exocytosis helps eliminate toxic accumulations, thereby relieving inhibition on residual enzyme function. This allows the remaining enzymes (e.g., CTSD and GAA) to be processed more efficiently, which in turn leads to further improvements in lysosomal digestion capacity.

Since the increase in lysosome digestive capacity is shown primarily through the DQ-BSA assay (Figure S5), the authors should both clearly explain how this assay works, show the colocalization of DQ-BSA signal with LAMP1, and show the effect of lysosome exocytosis on this assay.

Thank you for the suggestion. We have added additional explanations to the text. DQ-BSA is a derivative of BSA conjugated with a self-quenched dye. Upon delivery to the lysosome, it is cleaved by lysosomal proteases, resulting in fluorescence dequenching that can be detected as increased fluorescence intensity. The colocalization between DQ-BSA and LysoTracker is shown in Supporting Figure 5. Since DQ-BSA is a well-established probe in the lysosome field, we have chosen not to include this validation data in the main figures.

Is it possible that exocytosis is responsible for the signal changes in the DQ-BSA assay?

Exocytosis can only reduce the DQ-BSA signal, not increase it. Prior to reaching the lysosome, DQ-BSA fluorescence is self-quenched and undetectable. Once delivered to the lysosome, it can be exocytosed either before or after activation; however, in both cases, lysosomal exocytosis would decrease the detectable DQ-BSA fluorescence. However, in *LYSET/FAR1* DKO cells, DQ-BSA fluorescence was increased compared to *LYSET* single KO cells. The appropriate interpretation is that, despite the potentially negative impact of increased lysosomal exocytosis, the observed rise in DQ-BSA signal reflects improved lysosomal digestive capacity.

Does TRPML1 inhibition or VAMP7 knockdown change DQ-BSA levels?

This is a great question, but a challenging one to interpret experimentally. We would hypothesize two opposing effects of TRPML1 inhibition on DQ-BSA intensity: (1) impaired lysosomal degradation, which may reduce DQ-BSA signal by limiting substrate digestion; and (2) decreased lysosomal exocytosis, which may increase DQ-BSA signal by retaining the fluorescent degradation products within lysosomes. The balance between these two effects likely varies among different genetic backgrounds, making DQ-BSA data challenging to interpret in this context. Notably, both of these effects of TRPML1 inhibition are expected to enhance LAPTM4A and LC3B-II protein levels, which we now show in the new Figure 7J–L.

To investigate this further, we treated WT, LYSET KO, and LYSET/FAR1 DKO cells with 10 μ M ML-SI3 (a TRPML1 antagonist) for 48 hours—the same treatment used in the new Figure 7J–L—and examined DQ-BSA intensity (Supporting Figure 6). We observed a significant increase in DQ-BSA signal in WT cells but not in LYSET KO or LYSET/FAR1 DKO cells in response to TRPML1 inhibition. This indicates to us that TRPML1 inhibition can increase lysosomal degradation capacity in the WT background, but unfortunately remains inconclusive as to its effects in the LYSET KO and LYSET/FAR1 DKO setting.

Supporting Figure 6: DQ-BSA analysis of WT, LYSET KO, and LYSET/FAR1 DKO cells after ML-SI3 treatment.

Is the uptake of DQ-BSA similar in LYSET KO vs LYSET/FAR1 DKO cells?

This is an excellent question. Unfortunately, it's tricky to address directly: DQ-BSA fluorescence only occurs after proteolytic cleavage by lysosomal enzymes, which removes the dye quenching effect. Therefore, we cannot quantify the total amount of DQ-BSA taken up by the cell due to the lack of fluorescence.

However, DQ-BSA enters cells through endocytosis. According to studies by Thai et al.⁶ and Mansell et al.,⁷ ether lipid deficiency reduces the rate of endocytosis, suggesting

that less DQ-BSA may be internalized by LYSET/FAR1 DKO cells. Despite this possibility, LYSET/FAR1 DKO cells display higher DQ-BSA fluorescence signals than LYSET KO cells. Together with our finding that these cells exhibit increased lysosomal exocytosis, this suggests that the DKO cells have regained at least partial proteolytic activity within their lysosomes.

Likewise, are the faster “degradation kinetics” of LAPTM4A and LC3B-II explained by increased exocytosis of these substrates?

The faster reduction of accumulated LAPTM4A and LC3B-II likely reflects a combined effect of improved lysosomal digestion and enhanced exocytosis. We have revised the wording in the text to reflect this more accurately (lines 240-242).

The addback of HDG is a compelling experiment to show that the observed effects on lysosome accumulation are indeed acting through plasmalogens. The authors should take this a step further and directly test if the addback of plasmalogen HDG suppresses lysosome exocytosis or improves lysosome digestive capacity.

Thank you for these excellent suggestions. We have expanded our HDG analysis into a new Figure 6. This figure demonstrates the following:

- 1) Supplementation with HDG rescues LysoTracker signals in LYSET/FAR1 DKO cells (Figure 6B-C).
- 2) HDG treatment in three different WT cell lines, HeLa, HEK293T, and U2OS, further increases LysoTracker signals (Figure 6D-F), which is associated with impaired lysosomal digestion (Figure 6H-I).

Together, these data suggest that ether lipids negatively regulate lysosomal function. A reduction in ether lipid levels can improve lysosomal storage conditions, likely by promoting lysosomal exocytosis. In contrast, excessive ether lipids impair lysosomal digestion and trigger an increase of lysosome numbers.

Does the disruption of plasmalogen synthesis increase lysosome numbers in WT cells or is this phenomenon limited to defective lysosomes? Does excess plasmalogen effect lysosome exocytosis in WT cells? This could be assessed by adding HDG to WT cells or possibly through overexpression of FAR1, as in Figure 3A. I think the effect of plasmalogen levels on lysosome exocytosis in WT cells must be addressed to determine the extent of the relevance of this finding.

We thank the reviewer for raising these insightful questions. Knocking out FAR1 does not reduce lysosome numbers in HEK293T cells (new Figure S4B-C). In contrast, feeding HeLa cells with HDG increases lysosome numbers (new Figure 6G). As shown in the new Figure 7B, FAR1 knockout enhances lysosomal exocytosis in both WT and LYSET KO backgrounds.

The study from Phuyal et al⁸ showed that feeding PC-3 cells with excessive HDG enhances lysosome exocytosis. Together, these findings suggest that ether lipid levels and lysosomal exocytosis may follow an inverted bell-shaped relationship. That is, both low ether-lipid levels (as in FAR1 knockout) and excessive ether-lipid levels (as in HDG-fed cells) can stimulate lysosomal exocytosis. This relationship appears to hold true in both the WT and LYSET KO backgrounds, and in different cell types.

The authors currently speculate that the change in cholesteryl esters between LYSET KO and DKO is due to a change in degradation. Plasmalogen levels have been linked to cholesterol synthesis by SQLE (Honsho et al JBC 2015). Can the authors discuss the possibility of other regulatory responses to the plasmalogen deficit?

Thank you for bringing this relevant study to our attention. Intrigued by the findings in Honsho et al. (JBC, 2015), we measured whole-cell cholesterol levels in WT, LYSET KO, and LYSET/FAR1 DKO cells using a commercial cholesterol quantification kit (CS0005, Sigma Aldrich). This assay allows us to measure free cholesterol (FC), cholesteryl esters (CE), and total cholesterol.

As shown in supporting figure 7, in LYSET KO cells, CE levels increased dramatically, whereas FC levels remained largely unchanged, resulting in a significant increase in total cholesterol. This CE accumulation is likely due to the loss of active lysosomal enzymes, as lysosomes are the major site for CE hydrolysis. Consistent with this hypothesis, lipidomics analysis revealed a dramatic increase in CE levels in purified lysosomes (Figure S6).

In contrast, LYSET/FAR1 DKO cells showed a strong reduction in CE levels, while FC levels remained unchanged, leading to the reduction of total cholesterol. The reduction in CE is likely due to enhanced lysosomal exocytosis, as lipidomics data indicated ~90% lower CE content in purified lysosomes from DKO cells (Figure S6).

We were unable to directly assess SQLE expression due to the lack of a reliable antibody. Honsho et al. reported that reduced plasmalogen levels increase SQLE expression, potentially elevating cholesterol synthesis. However, our results differ from this prediction: compared to LYSET KO cells, the LYSET/FAR1 DKO cells exhibited reduced CE and total cholesterol levels, rather than an increase. It should be noted that we did not

compare WT to *FAR1* KO cells, which might exhibit elevated cholesterol levels following *FAR1* knockout alone.

The abstract explicitly says that plasmalogen deficiency facilitates “secretion of accumulated substrates” – but what is shown is an increase in extracellular vesicles that could be explained by mechanisms other than lysosome exocytosis: increased vesicle formation, or increased budding of EVs from the plasma membrane. The conclusion that the effects are through lysosome exocytosis would be better supported if the authors could show that the lysosomal substrates that they have been following throughout the paper, such as LAPT4A, have increased levels in the media, or that lysosomal membrane proteins such as LAMP1 are now on the plasma membrane as a result of increased lysosome fusion with the plasma membrane.

These are all excellent suggestions, thank you! To further support our enhanced lysosomal exocytosis hypothesis, we first measured the levels of mature CTSD in conditioned media. Because CTSD must first reach the lysosome and be processed into its mature form, detecting mature CTSD in conditioned media specifically reflects lysosomal exocytosis activity. We overexpressed exogenous CTSD in WT and *FAR1* KO cells (but not in the *LYSET* KO background, due to defective lysosomal enzyme processing). We observed higher levels of mature CTSD in the conditioned media of *FAR1* KO cells (new Fig. 6G, H).

Similarly, we examined two other lysosomal degradation cargos, LAPT4A-EGFP and EGFP-RNF152, which are internalized from the lysosomal membrane into intraluminal vesicles (ILVs) for degradation⁹. Lysosome-derived ILVs can also be recovered from conditioned media via 100,000 × g ultracentrifugation. We detected about 3-4 fold higher levels of LAPT4A-EGFP and EGFP-RNF152 in this fraction from *LYSET/FAR1* DKO cells, consistent with increased lysosomal exocytosis compared to *LYSET* KO.

Minor comments:

Abstract: The authors could add a claim that the accumulation of lysosomes is not driven by TFEB.

We have added this claim as suggested. Thank you!

Line 100 – Please explain what DQ-BSA measures.

We have added sentences explaining the assay (lines 101-105).

Lines 102-103 – Please provide a citation for the prevailing hypothesis that TFEB/TFE3 drives lysosome biogenesis. The authors observation that the double KO of TFEB/TFE3 does not significantly reduce lysosome numbers appears to challenge this hypothesis and warrants further discussion.

We have added references.

Line 106 – citation needed for “CLEAR” network.

We have added references.

Line 119 – EBSS treatment is not described in the methods.

We have added the EBSS treatment in the Materials and Methods (lines 763-766).

Line 393 – In Figure 1D there are two populations present in the LYSET and GNPTAB KO. It would thus be more appropriate to report the median fluorescence intensity rather than the mean. Related: Is the double population an indication of incomplete KO? I cannot determine from the results, methods, or figure legend whether these KO cell lines are monoclonal or polyclonal. Please clarify in figure legends and provide verification (as in Figure 1F).

For Figure 1D, we tested both the mean and median methods for the DQ-BSA intensity analysis and obtained similar results. The cell lines used are monoclonal, generated and validated by sequencing in our previous study⁹. In this paper, monoclonal KOs are labeled as “XXX KO” and polyclonal KOs as “+sg_XXX” (XXX indicates the targeted gene). While we are confident in the single colonies we isolated, we observed a minor residual population in both GNPTAB KO and LYSET KO lines for reasons that remain unclear in the DQ-BSA assay. Western blot validation of the knockouts is provided in the supporting figure below (Supporting Figure 8).

Supporting Figure 8: Validation of LYSET and GNPTAB knockout cells by western blot.

Figure S1B – This figure needs a positive control, such as LYSET mRNA transcripts. The n.s. line is confusing, as it appears to show that none of the transcript levels change significantly relative to each other, but the relevant comparison is for each transcript between WT levels and LYSET KO levels. Perhaps this should be shown with raw values for each of these instead of with the normalization.

Supporting Figure 9 (also new Figure S1B): Transcription levels of indicated CLEAR network genes in *LYSET* KO to WT HEK293T cells.

As suggested, we added a WT control column for each gene to improve data presentation in Figure S1B (also Supporting Figure 9). We also included positive (UAP1L1) and negative (LGR5) controls, which were previously reported to be upregulated and downregulated, respectively, in *LYSET* KO cells⁹. Please note that UAP1L1 and LGR5 are not CLEAR network genes.

Line 144-145 – The authors need to be more transparent about the full outcomes of the screen, as this screen will be a resource to the community and other scientists may be interested in how enriched their gene of interest was. The fold change and significance for all genes assayed in the screen should be provided in a supplemental table. It would also be interesting to see Figure 2E with genes known to influencing lysosome biogenesis/numbers and cholesterol synthesis highlighted, such as TFEB and the CLEAR network.

We thank the reviewer for these suggestions. We have added a new sheet, “All genes,” to Supplementary Table 1, providing the full screening results. We examined TFEB, TFE3, and SQLE in our screen list; however, they were not enriched, suggesting that knockout of these genes does not reduce LysoTracker intensity in *LYSET* KO cells.

Figure 2C – please provide quantification for LAMP2 staining. It is important to know if sorting by LysoTracker screening correlated quantitatively with LAMP2 immunofluorescence.

Our quantification method requires bright-field or DIC images to calculate cell area. Unfortunately, we did not acquire bright-field images for this experiment, as the reduction in LAMP2 signal was already apparent between the “LYSET KO” and “after 3rd round sorting” samples. More importantly, we dedicated the entire Figure 3 to validating the screening results, where we quantified LAMP2 signals and confirmed their consistency with LysoTracker staining.

Figure S3A - The position of the Pex5, Pex7, and Pex19 labels makes it unclear what gene is drawn out.

We have added the header “Required PTS Receptor” to improve clarity. Thank you!

Line 175-176 – Please provide a reference for the determination of the peroxisome targeting signals by “sequence analysis”, especially for Far1, since mPTSs are difficult to predict from sequence, and stronger evidence would be from Honsho et al.

We have added references for the peroxisome targeting signals (now line 196). Thank you!

Figure 3 H, J, L, N and S4C – please add more information about replicates to figure legend. It would be helpful if S4B and C were presented similarly to Fig. 3G and H.

We have added the requested information to the figure legends for Figures 3H, J, L, and N. As for Figure S4B,C (now Figure S5A,B), because the reduction in LysoTracker intensity in iMEF cells is less pronounced, though still significant, compared to HEK293T cells, we chose to present the flow cytometry data in an overlaid format for better comparison.

Figure 3 - How do lysosome numbers compare between an AGPS KO and an AGPS/LYSET double knockout? Or a FAR1 KO and a FAR1/LYSET double knockout? Do lysosomes still increase upon loss of LYSET, suggesting that LYSET induces lysosome accumulation through a mechanism separate from a loss of exocytosis?

Because knocking out either FAR1 or AGPS abolishes ether lipid synthesis, we focused on FAR1 to address these questions. As shown in Supporting Figure 10, there is no significant difference in lysosome numbers among WT, FAR1 KO, and LYSET/FAR1 DKO cells, indicating two main conclusions: 1) that FAR1 KO alone does not significantly alter WT lysosome numbers, and 2) that the effect of FAR1 KO in the LYSET KO background is to restore lysosome number to near-WT levels.

Supporting Figure 10: Lysosome number comparison among WT, FAR1 KO, and LYSET/FAR1 DKO cells. Data represent mean \pm s.d.; $n=32$ cells, from three biological replicates.

Figure S5A – which band corresponds to LAPT M4A as quantified in B? What is the other band in the top blot? Please address this in the LAPT M4A blots throughout the paper, as the two bands behave quite differently and muddy the interpretation.

We used the upper LAPT M4A band for quantification in Figure S5A (now Figure 4B) and have added a description of the LAPT M4A quantification in the figure legends. Based on the molecular weight, the upper band corresponds to the full-length LAPT M4A, while the lower band represents a cleavage product of LAPT M4A^{9,10}. Our western blot data show that both the full-length and cleaved forms of LAPT M4A are reduced upon FAR1 or AGPS knockout, as well as after treatment with an AGPS inhibitor.

Figure 4B – It would be helpful to label the expected subcellular localization for the shown proteins (ie -LAMP2 in lysosome, Hrd in ER). Given the paper discusses peroxisome/lysosome crosstalk, it would be nice to show a peroxisomal protein here to rule out an direct contact/association of lysosomes and peroxisomes.

Thank you. We added organelle information and included PMP70 in the new Figure 5B as suggested.

Figure 4G – the levels of specific luminal substrates have varying sensitivity to plasmalogen synthesis – ie ApoE is fully recovered while SQSTM1 is not. Can the authors speculate as to why this might be the case? Are some lysosomes more prone to exocytosis than others?

The protein level differences observed by Western blot are consistent with the changes detected by SILAC mass spectrometry. The variability in reduction levels among

different substrates may be due to differences in antibody sensitivity, the rates at which substrates are delivered to lysosomes (e.g., autophagic versus endocytic cargoes), or variations in their protein expression levels.

Figure 5 - A cartoon for “EV formation and lysosome exocytosis” would be clarifying.

We have added a cartoon model as suggested (Figure 7A). Thank you!

Line 316 – Figure 5I – I believe this reference is to Figure 5F.

We corrected this.

Figure S7 – Why aren't plasmalogens shown in this figure? Do plasmalogen levels increase in LYSET KO cells?

We have added the lysosomal and whole-cell plasmalogen data in the updated Figure S6. Detailed plasmalogen information is also provided in the revised Supplementary Tables 3 and 4.

Reviewer #4 (Remarks to the Author):

Uncovering Peroxisome-Lysosome Crosstalk: Plasmalogen Deficiency Restores Lysosomal Function by Stimulating Exocytosis

This study by Chen et al elegantly uncovers a role for plasmalogens, a class of glycerophospholipids that are synthesised in peroxisomes, in late endosome/lysosome exocytosis. The data indicate that reducing plasmalogen synthesis increases exocytosis to clear lysosomal expansion/storage in cellular models of mucopolipidosis. The findings are novel, the approach logical and thorough and the data convincing and clearly presented. The study robustly demonstrates that loss of activity of plasminogen biosynthesis enzymes clears lysosomal accumulation and improves degradative capacity and exocytosis dependent on TRPML1-mediated Ca²⁺ release is proposed as a mechanism.

Thank you so much for the positive comments.

The authors have manipulated peroxisomal proteins to disrupt plasmalogen biosynthesis, a process that is completed in the ER and importantly, have shown that effects of disrupting plasmalogen biosynthesis can be reversed by the addition of plasmalogen precursor HDG, which bypasses the peroxisomal steps. However, the study is not without shortcomings, especially the lack of any investigation into how plasmalogens might influence TRPML1 activity.

How do the authors propose plasmalogens inhibit TRPML1? Is this a direct interaction or an indirect effect?

We do not suggest that plasmalogens (or ether lipids) can inhibit TRPML1, although this hypothesis could explain many of our observations. We only showed that ether lipid deficiency can increase lysosome exocytosis, and TRPML1 and SNARE proteins are important for the exocytosis process. In our revised Discussion, we speculate that ether lipids may impact membrane fusion/fission processes more broadly, given their previously described links to endocytosis and this new finding of their involvement in exocytosis regulation. However, these mechanistic questions are beyond the scope of the current study, which serves to establish an important new link between peroxisome-derived ether lipids and lysosomal exocytosis that can ameliorate disease-associated lysosomal accumulation and dysfunction.

Some of this may be beyond the scope of the study, but should at least be considered/discussed:

How do plasmalogens reach the lysosome? Are ER:lysosome contacts involved or do newly synthesized plasmalogens undergo vesicular traffic to the lysosome?

We thank the reviewer for these insightful comments. We believe that both vesicular trafficking and ER–lysosome contact sites contribute to the delivery of ether lipids to lysosomes. A recent study by Hancock-Cerutti et al.¹¹ showed that ether lipid levels were reduced in VPS13C KO lysosomes, but not at the whole-cell level, suggesting that lipid transfer can occur through contact site–mediated mechanisms. Furthermore, our data show that knocking out members of the ORP family or ATG2A/B reduces LysoTracker intensity in LYSET KO cells (see Supporting Figure 11 below), supporting the idea that ER-lysosome contacts play an important role in lipid transfer to lysosomes.

Since VAPs have been implicated in ER contact sites with both peroxisomes (likely necessary for plasmalogen biosynthesis) and lysosomes, does VAP depletion similarly reduce the lysosomal phenotypes in LYSET KO cells?

Thank you for this excellent suggestion. Because of the significant redundancy between VAPA and VAPB, we attempted to generate a VAPA/VAPB double knockout but were unsuccessful, likely due to the lethality of the homozygous knockout. Previous studies have reported that members of the oxysterol-binding protein (OSBP)-related protein (ORP) family participate in ER-lysosome contact sites, and that ATG2 mediates lipid transfer between ER and lysosomes¹². Based on this, we examined ORP quadruple knockout (ORP9, ORP10, ORP11, OSBP; ORP-QKO) and ATG2A/B DKO cells. As shown in Supporting Figure 11, both ORP-QKO and ATG2 DKO reduced LysoTracker intensity in LYSET KO cells, whereas PI4K2A KO had no significant effect. These results indicate that blocking ER-lysosome lipid transfer can partially alleviate the lysosomal accumulation phenotype observed in LYSET KO cells, highlighting a role for ER-lysosome contact sites in regulating lysosome number. However, whether this effect arises from impaired transfer of ether lipids, ester lipids, or both remains to be determined.

Are ER contacts with peroxisomes and/or lysosomes altered in LYSET-KO or on inhibition of plasmalogen biosynthesis or after addition of HDG?

We labeled the ER, lysosomes, and peroxisomes and analyzed their co-localization in WT, LYSET KO, and LYSET/FAR1 DKO HEK293T cells (Supporting Figure 12). We did not observe any significant changes in ER–peroxisome contacts among these cell lines. However, ER–lysosome overlap was increased in LYSET KO cells compared to WT, likely reflecting the elevated lysosome number. In LYSET/FAR1 DKO cells, this overlap returned to WT levels.

Interestingly, HDG treatment did not significantly alter either ER–lysosome or ER–peroxisome contacts, even though lysosome numbers increased following treatment (Supporting Figure 13). The basis for this discrepancy remains unclear.

Supporting Figure 12: Immunofluorescence showing LAMP1-mCherry (lysosome), PMP70 (peroxisome), and EGFP-VAPA (ER) localizations in WT, LYSET KO, and LYSET/FAR1 DKO HEK293T cells. Scale bar: 10 μ m.

Supporting Figure 13: Immunofluorescence images showing the localization of LAMP1-mCherry (lysosomes), PMP70 (peroxisomes), and EGFP-VAPA (ER) in HEK293T cells after treatment with 50 μ M HDG for 24 hours. Scale bar, 10 μ m.

- *TRPML1* activity is strongly influenced by the phospholipid environment and interaction with PI(4,5)P₂ has been shown to inhibit *TRPML1*. Do plasmalogens similarly bind *TRPML1* or is the proposed effect on *TRPML1* activity indirect?

Thank you for the suggestion. Our lab does not work on electrophysiology. In the past, our department had a *TRPML1* expert, Haoxing Xu, but he returned to China about three years ago due to geopolitical tensions between the two countries, and we cannot collaborate with his current lab either. We also contacted another PI in Canada who trained with him, but their lab was recently flooded and will not be able to restart electrophysiology recordings for about a year. Finally, we feel that testing the mechanistic details of whether

and how ether lipids might affect TRPML1 channel activity is outside the scope of this study.

Is PI(4,5)P₂ distribution affected by plasmalogen biosynthesis?

We expressed eGFP-2xPH-PLCD1 in WT, LYSET KO, and LYSET/FAR1 DKO HEK293T cells to visualize PI(4,5)P₂ distribution. In all three cell lines, PI(4,5)P₂ was predominantly localized to the plasma membrane, with minimal differences observed between conditions (see supporting figure 14 below).

Supporting Figure 14: Maximum projection images showing lysosome (labeled by LysoTracker Red) and PI(4,5)P₂ (labeled by EGFP-2xPH-PLCD1) distribution in WT, LYSET KO, and LYSET/FAR1 DKO HEK293T cells. Scale bar: 10 μ m.

Minor points:

1. Line 124: "suggests an alternative, yet unidentified mechanism driving lysosome biogenesis in these conditions"... is it definitely increased biogenesis or perhaps the absence of increased TFEF activation suggests that reduced clearance/disruption of the lysosome lifecycle may contribute to the lysosomal expansion?

Thank you for pointing this out. We agree that the term “lysosome biogenesis” could be misleading, as we did not observe transcriptional upregulation of lysosomal genes (Figure S1B). In the revised manuscript, we have replaced this term with “increase in lysosome numbers” or “lysosome accumulation” to more accurately reflect our observations.

2. Figure S3E: No GNPAT-GFP signal at all is visible in PEX5-silenced cells - how can we be sure that these cells are expressing EGFP-GNPAT? Also, do you expect reduced protein when peroxisomal import is inhibited? Wouldn't expression be expected to be upregulated by the reduction in plasmalogen? Is the reduced protein prevented by inhibiting degradation? Do you see mislocalisation, eg does FAR-1 appear on lipid droplets in Pex19 silenced cells?

Please see the updated Figure S3. After knocking out PEX5, EGFP-GNPAT became diffusely distributed in the cytosol but did not disappear (Fig. S3H). Western blot analysis showed that EGFP-GNPAT migrated as two bands at ~100 kDa. The lower band was markedly reduced in the PEX5 KO cells, while the upper band was slightly increased. These results suggest that EGFP-GNPAT may undergo post-import cleavage after entering the peroxisome. Knocking out PEX5 abolished both import and cleavage, resulting in cytoplasmic accumulation of EGFP-GNPAT. Interestingly, similar results were observed for AGPS-EGFP. The precise cleavage sites and the enzymes responsible for processing GNPAT and AGPS will be determined in future studies.

Plasmalogen deficiency has been reported to induce FAR1 expression, while excessive plasmalogen promotes FAR1 degradation¹³. The mechanism of FAR1 degradation remains unclear. Our SILAC analysis suggests that LYSET knockout leads to FAR1 accumulation in purified lysosomes, a finding confirmed by western blot (see Supporting Figure 15), indicating that lysosomes at least contribute to FAR1 degradation.

Supporting Figure 15: Western blot analysis of FAR1 in purified lysosomes from WT and LYSET KO cells. Note the increased FAR1 protein level in LYSET KO lysosomes.

The turnover of FAR1 after PEX19 knockout is also an interesting question, but it is beyond the scope of this study, which focuses on the relationship between ether lipids synthesis and lysosomal function.

Supporting Figure 16: Effects of different fixation conditions on FAR1 immunostaining and lipid droplet visualization. Under 4% PFA fixation, only background signal was detected for FAR1 (A), whereas methanol fixation produced strong and specific FAR1 staining (B). In contrast, lipid droplets labeled with BODIPY 493/503 were clearly visualized under 4% PFA fixation (C) but not with methanol fixation (D). Treatment with 0.5 mM oleic acid (OA) was used as a positive control to induce lipid droplet formation. No lipid droplets were detected in untreated HEK293T cells, even under PFA fixation. Scale bar: 10 μ m.

Regarding whether FAR1 mislocalizes to lipid droplets after PEX19 knockout, we attempted to co-stain FAR1 and lipid droplets in PEX19-silenced cells. However, due to technical limitations, we were unable to identify a fixation condition compatible with both markers: 4% PFA fixation did not work for FAR1 immunostaining, while methanol fixation failed for lipid droplet staining with BODIPY 493/503. Moreover, we did not detect lipid droplets in HEK293T cells without oleic acid treatment (Supporting Figure 16). These results suggest that FAR1 is unlikely to localize to lipid droplets in HEK293T cells under normal culture conditions, as essentially no lipid droplets were detected under these conditions.

3. Fig 3: The focus is very much on the role of peroxisome proteins but since plasmalogen biosynthesis is completed in the ER, it would be interesting to see if depletion of ER-localised plasmalogen biosynthesis enzymes similarly reduces the lysosome expansion phenotype.

We thank the reviewer for raising this important question and for the thoughtful suggestions. A similar point was also noted by Reviewer #1.

To address this, we knocked out PEDS1 (TMEM189), an ER-localized enzyme responsible for converting plasmalogen lipids into plasmalogen lipids (plasmalogens), in LYSET KO cells. In the resulting LYSET/TMEM189 DKO cells, we did not observe a significant reduction in LysoTracker intensity (new Figure 5M-N). These results indicate that ether lipids, rather than the plasmalogen subclass, play the predominant role in regulating

lysosomal function in HEK293T cells. We have revised the manuscript to reflect this updated conclusion.

This finding also explains why PEDS1 (TMEM189) did not appear as a top hit in our genetic screen. We appreciate the reviewer's insightful suggestion, which helped strengthen our study.

4. Previous reports implicating plasmalogen biosynthesis in exocytosis should be cited (eg. PMID 25519911).

Per this reviewer's request, this paper has been cited.

5. If the mechanism of lysosome normalisation is activated TRPML1 when plasmalogen biosynthesis is reduced, then does TRPML1 activation with MLSA also normalise the lysosome phenotypes in M6P-deficient cells?

According to the study by Medina and colleagues¹⁴, overexpression of TFEB induces lysosomal exocytosis through upregulation of TRPML1, thereby promoting cellular clearance in lysosomal storage diseases. Other studies have shown that TRPML1 agonists can reduce substrate accumulation in NPC1 KO and CLN3 KO cells^{15,16}. Based on these findings, we hypothesized that ML-SA5 treatment might normalize lysosomal phenotypes in M6P-deficient cells.

However, when we treated WT, LYSET KO, and LYSET/FAR1 DKO cells with the TRPML1 agonist ML-SA5 (1 μ M, 6 h), we instead observed an increase in LAPT4A and LC3B-II protein levels (Supporting Figure 17). Previous reports indicate that TRPML1 agonists can elevate LC3-II levels by inhibiting autophagosome degradation and that TRPML1 activation can promote TFEB nuclear translocation and transcriptional upregulation of autophagy and lysosomal genes¹⁷. Therefore, we interpret the observed increase in LAPT4A and LC3B-II as a result of transcriptional upregulation and/or impaired degradation induced by TRPML1 activation.

Supporting Figure 17: Western blot analysis of LAPT4A and LC3B-II levels in WT, LYSET KO, and LYSET/FAR1 DKO cells following 6-hour treatment with 1 μ M ML-SA5.

6. *It would be nice to see if cytosolic calcium (eg Fura-2 measurements) is increased when plasmalogen synthesis is blocked (though this might be technically difficult).*

We thank the reviewer for this suggestion. Unfortunately, we do not have the setup to perform Fura-2 measurements.

References:

- 1 Charles, K. N. *et al.* Functional Peroxisomes Are Essential for Efficient Cholesterol Sensing and Synthesis. *Front Cell Dev Biol* **8**, 560266 (2020). <https://doi.org/10.3389/fcell.2020.560266>
- 2 Lopez de Frutos, L. *et al.* Serum Phospholipid Profile Changes in Gaucher Disease and Parkinson's Disease. *Int J Mol Sci* **23** (2022). <https://doi.org/10.3390/ijms231810387>
- 3 Lecommandeur, E. *et al.* Decrease in Myelin-Associated Lipids Precedes Neuronal Loss and Glial Activation in the CNS of the Sandhoff Mouse as Determined by Metabolomics. *Metabolites* **11** (2020). <https://doi.org/10.3390/metabo11010018>
- 4 Guedes, L. C. *et al.* Serum lipid alterations in GBA-associated Parkinson's disease. *Parkinsonism Relat Disord* **44**, 58-65 (2017). <https://doi.org/10.1016/j.parkreldis.2017.08.026>
- 5 Korotkova, D. *et al.* Fluorescent fatty acid conjugates for live cell imaging of peroxisomes. *Nat Commun* **15**, 4314 (2024). <https://doi.org/10.1038/s41467-024-48679-2>
- 6 Thai, T. P. *et al.* Impaired membrane traffic in defective ether lipid biosynthesis. *Hum Mol Genet* **10**, 127-136 (2001). <https://doi.org/10.1093/hmg/10.2.127>
- 7 Mansell, R. P. *et al.* Ether lipids influence cancer cell fate by modulating iron uptake. *bioRxiv* (2025). <https://doi.org/10.1101/2024.03.20.585922>
- 8 Phuyal, S. *et al.* The ether lipid precursor hexadecylglycerol stimulates the release and changes the composition of exosomes derived from PC-3 cells. *J Biol Chem* **290**, 4225-4237 (2015). <https://doi.org/10.1074/jbc.M114.593962>
- 9 Zhang, W. *et al.* GCAF(TM251) regulates lysosome biogenesis by activating the mannose-6-phosphate pathway. *Nat Commun* **13**, 5351 (2022). <https://doi.org/10.1038/s41467-022-33025-1>
- 10 Zhang, W. *et al.* A conserved ubiquitin- and ESCRT-dependent pathway internalizes human lysosomal membrane proteins for degradation. *PLoS Biol* **19**, e3001361 (2021). <https://doi.org/10.1371/journal.pbio.3001361>
- 11 Hancock-Cerutti, W. *et al.* ER-lysosome lipid transfer protein VPS13C/PARK23 prevents aberrant mtDNA-dependent STING signaling. *J Cell Biol* **221** (2022). <https://doi.org/10.1083/jcb.202106046>

- 12 Tan, J. X. & Finkel, T. A phosphoinositide signalling pathway mediates rapid lysosomal repair. *Nature* **609**, 815-821 (2022). <https://doi.org/10.1038/s41586-022-05164-4>
- 13 Honsho, M., Asaoku, S. & Fujiki, Y. Posttranslational regulation of fatty acyl-CoA reductase 1, Far1, controls ether glycerophospholipid synthesis. *J Biol Chem* **285**, 8537-8542 (2010). <https://doi.org/10.1074/jbc.M109.083311>
- 14 Medina, D. L. *et al.* Transcriptional activation of lysosomal exocytosis promotes cellular clearance. *Dev Cell* **21**, 421-430 (2011). <https://doi.org/10.1016/j.devcel.2011.07.016>
- 15 Shen, D. *et al.* Lipid storage disorders block lysosomal trafficking by inhibiting a TRP channel and lysosomal calcium release. *Nat Commun* **3**, 731 (2012). <https://doi.org/10.1038/ncomms1735>
- 16 Wunkhaus, D. *et al.* TRPML1 activation ameliorates lysosomal phenotypes in CLN3 deficient retinal pigment epithelial cells. *Sci Rep* **14**, 17469 (2024). <https://doi.org/10.1038/s41598-024-67479-8>
- 17 Qi, J. *et al.* MCOLN1/TRPML1 finely controls oncogenic autophagy in cancer by mediating zinc influx. *Autophagy* **17**, 4401-4422 (2021). <https://doi.org/10.1080/15548627.2021.1917132>

Dear Ming,

Thank you for submitting your manuscript for consideration by the EMBO Journal. It has now been seen by two of the four original referees whose comments are enclosed. As you will see, both referees are broadly in favour of publication, pending the incorporation of the changes that are listed in their reports.

I will write again soon with a list of editorial revisions that will also need to be made.

Given the referees' positive recommendations, I would like to invite you to submit a revised version of the manuscript, addressing the comments of all three reviewers. I should add that it is EMBO Journal policy to allow only a single round of revision, and acceptance of your manuscript will therefore depend on the completeness of your responses in this revised version.

Please bear in mind that your letter of response to the referees will form part of the Review Process File, and will therefore be available online to the community. For more details on our Transparent Editorial Process, please review our Editorial Policies page: <https://link.springer.com/partners/embo-press/editorial-policies>.

We generally allow three months as standard revision time. As a matter of policy, competing manuscripts published during this period will not negatively impact on our assessment of the conceptual advance presented by your study. However, we request that you contact the editor as soon as possible upon publication of any related work, to discuss how to proceed.

Thank you for the opportunity to consider your work for publication. I look forward to your revision.

Best wishes,

William

William Teale, PhD
Editor
The EMBO Journal
w.teale@embojournal.org

When submitting your revised manuscript, please carefully review the instructions below and include the following items:

- 1) a .docx formatted version of the manuscript text (including legends for main figures, EV figures and tables). Please make sure that the changes are highlighted to be clearly visible.
- 2) individual production quality figure files as .eps, .tif, .jpg (one file per figure).
- 3) a .docx formatted letter INCLUDING the reviewers' reports and your detailed point-by-point response to their comments. As part of the EMBO Press transparent editorial process, the point-by-point response is part of the Review Process File (RPF), which will be published alongside your paper.
- 4) a complete author checklist, which you can download from our author guidelines (<https://link.springer.com/journal/44318/submission-guidelines#cms-Revised-submissions>). Please insert information in the checklist that is also reflected in the manuscript. The completed author checklist will also be part of the RPF.
- 5) Please note that all corresponding authors are required to supply an ORCID ID for their name upon submission of a revised manuscript.
- 6) We require a 'Data Availability' section after the Materials and Methods. Before submitting your revision, primary datasets produced in this study need to be deposited in an appropriate public database, and the accession numbers and database listed under 'Data Availability'. Please remember to provide a reviewer password if the datasets are not yet public (see <https://link.springer.com/partners/embo-press/editorial-policies#Data%20deposition>). If no data deposition in external databases is needed for this paper, please then state in this section: This study includes no data deposited in external repositories. Note that the Data Availability Section is restricted to new primary data that are part of this study.

Note - All links should resolve to a page where the data can be accessed.

- 7) When assembling figures, please refer to our figure preparation guideline in order to ensure proper formatting and readability in print as well as on screen:
<https://link.springer.com/journal/44318/submission-guidelines#cms-Figure-and-data-presentation>

8) For data quantification: please specify the name of the statistical test used to generate error bars and P values, the number (n) of independent experiments (specify technical or biological replicates) underlying each data point and the test used to calculate p-values in each figure legend. The figure legends should contain a basic description of n, P and the test applied. Graphs must include a description of the bars and the error bars (s.d., s.e.m.).

9) We would also encourage you to include the source data for figure panels that show essential data. Numerical data can be provided as individual .xls or .csv files (including a tab describing the data). For 'blots' or microscopy, uncropped images should be submitted (using a zip archive or a single pdf per main figure if multiple images need to be supplied for one panel). Additional information on source data and instruction on how to label the files are available at <https://link.springer.com/journal/44318/submission-guidelines#cms-Source-data>

10) We replaced Supplementary Information with Expanded View (EV) Figures and Tables that are collapsible/expandable online (see examples in <https://www.embopress.org/doi/10.15252/emboj.201695874>). A maximum of 5 EV Figures can be typeset. EV Figures should be cited as 'Figure EV1, Figure EV2' etc. in the text and their respective legends should be included in the main text after the legends of regular figures.

- For the figures that you do NOT wish to display as Expanded View figures, they should be bundled together with their legends in a single PDF file called *Appendix*, which should start with a short Table of Content. Appendix figures should be referred to in the main text as: "Appendix Figure S1, Appendix Figure S2" etc. See detailed instructions regarding expanded view here: < <https://link.springer.com/journal/44318/submission-guidelines#cms-Expanded-View-data> >.

12) Our journal encourages inclusion of *data citations in the reference list* to directly cite datasets that were re-used and obtained from public databases. Data citations in the article text are distinct from normal bibliographical citations and should directly link to the database records from which the data can be accessed. In the main text, data citations are formatted as follows: "Data ref: Smith et al, 2001" or "Data ref: NCBI Sequence Read Archive PRJNA342805, 2017". In the Reference list, data citations must be labeled with "[DATASET]". A data reference must provide the database name, accession number/identifiers and a resolvable link to the landing page from which the data can be accessed at the end of the reference. Further instructions are available at <

Further instructions for preparing your revised manuscript:

Read our guidance for manuscript revisions and related editorial policies: <https://link.springer.com/journal/44318/submission-guidelines#cms-Revised-submissions>

<https://media.springernature.com/original/springer-cms/rest/v1/content/27825798/data/v1>

- a point-by-point response to the referees' comments, with a detailed description of the changes made (as a word file).
- a word file of the manuscript text.
- individual production quality figure files (one file per figure)
- a complete author checklist
- Expanded View files (replacing Supplementary Information)
- a Reagents and Tools Table as part of the Methods section

Please remember: Digital image enhancement is acceptable practice, as long as it accurately represents the original data and

conforms to community standards. If a figure has been subjected to significant electronic manipulation, this must be noted in the figure legend or in the 'Methods' section. The editors reserve the right to request original versions of figures and the original images that were used to assemble the figure.

We realize that it is difficult to revise to a specific deadline. In the interest of protecting the conceptual advance provided by the work, we recommend a revision within 3 months (15th Apr 2026). Please discuss the revision progress ahead of this time with the editor if you require more time to complete the revisions. Use the link below to submit your revision:

Referee #1:

The study by Liang Cheng reveals a novel connection between peroxisomes and lysosomes mediated by ether-bound glycerolipids. The manuscript's findings are novel and represent a significant advance in the field.

The revised version is much more balanced between the lysosomal and peroxisomal sections, and has clearly benefited from the revision process.

The introduction and discussion sections have been broadened in line with the reviewers' suggestions, and new experiments have been added to support the overall conclusions, which are now more justified.

Based on my previous comments, there are still some minor points to be addressed in this revised form:

1. Fluorescence microscopy pictures should be accompanied by a detailed analysis (as shown in Fig. 3B for 3A) for all pictures. Similarly, all western blots need densitometric evaluation.
2. Ref. 51 (Chu, Cell, 2015) is now cited but only in the results section. Given that this publication clearly demonstrates a lipid link between the two organelles in question, I believe it should be referenced in the introduction or discussion sections.
3. PEDS1 is the official gene symbol for the PEDS1 protein. (<https://www.ncbi.nlm.nih.gov/gene/387521>). TMEM189 can be mentioned as the previous gene symbol but the authors should then stick to the newest nomenclature.
4. RCDP consists of four, rather than five, subtypes. FAR1 leads to PFCRD (see the OMIM entry # 616154).
5. The Methods section (lines 933-941) still reads like a laboratory protocol.
6. Fig. 2F: correct to "phospholipids".

Referee #2:

Chen et al set out to identify genes that reduce lysosome accumulation in Mucopolipidosis V cell lines. They find that the loss of ether lipids (and therefore loss of peroxisome function) improves lysosome clearance, which they then show is through a combined mechanism of lysosomal exocytosis and improved digestion. The work is nicely done, and the implications for lysosomal storage disease and WT cells are interesting and clear. I expect future mechanistic work will reveal interesting new functions of ether lipids in lysosome/plasma membrane fusion. I find the study to be compelling, and impactful for scientists interested in lysosomes, peroxisomes, lipids, and unconventional secretion.

The authors have adequately addressed the reviewer concerns, and significantly improved the manuscript. I recommend it for publication, assuming the authors can address the one major concern that the mass spec data is inadequately documented. I have included minor suggestions that I would leave to the authors' discretion.

Major concern:

SILAC/MS data must be better documented in methods, excel spreadsheets, and raw data uploaded to a repository. It would have been nice to see the difference between WT and the DKO.

In the methods about this: "Gel bands were excised" - It doesn't come across in the figure that you excised specific bands for analysis. It would be informative to label dominant protein changes you identified on the gel in Fig. 5C (ie 30 kDa protein in WT that is not in LYSET KO or DKO? 300 kDa protein that is in KO and DKO but not in WT)

Methods is inadequate: How did the Mass Spec Facility collect data? What instrument, HPLC column, MS parameters? How were peptides/proteins identified? Where is the MS data deposited? The data reporting here is insufficient. The currently available excel data table should be labeled in the excel data sheet with Supp Table Number, and data should include protein data (protein name, accession number and molecular weight, number of assigned spectra and unique sequences used for

protein identification, percent sequence coverage, database search score) and peptide data (peptide sequence data with start and stop residue numbers, observed mass, mass error, scores/expect values for assignment, post-translational modifications, probability determination for modification sites).

Minor suggestions for improvement:

Line 39: "This mechanism" - authors say repeatedly that mechanism is beyond the scope. I would revise this sentence.

Line 111: "leading to the dephosphorylation..." could be "dephosphorylation and nuclear translocation" for clarity on why localization is relevant in Fig. S1A

Line 119 and Fig 1F - You discuss this cell line as a triple knockout, but the TKO nomenclature isn't explicitly used in the figure: +/- sgRNA for LYSET. It would help to be consistent.

Figure 2F - typo in "phosphoipids"

Figure S3B - What is the evidence that the starred band is non-specific?

Figure S3D - triangle/arrowheads don't point to bands on immunoblot.

Line 196 - I don't think it is accurate to say PEX5 knockdown reduced total levels of GNPAT from Fig. S2C. In general the immunoblots show evidence of altered post-translational modifications more so than total levels. Figures S2F-H show the change in levels for FAR1 and GNPAT (but not so much AGPS). It's not absolutely essential, but I would recommend moving S3F-J before S3B-E in the figure and discussion.

Figure S3F - Quantification of colocalization with FAR1/PMP70 could be shown for completeness

Fig S4 D-H is brought up after Fig S5.

Fig S4G - the X axis could be altered to make it easier to see the relevant data.

Line 259-261 - this speculation about pharmacological relevance should be in the discussion, not the results.

In fig 4 or S5, the authors could have looked at the degradation capacity of WT and FAR1 KO cells. While they showed no change in lysosome levels in FAR1 KO cells compared to WT, they did not directly look for changes in degradation kinetics.

Figure 5F- I would appreciate some discussion of the proteins that went up between DKO/LYSET KO. (Gray dots on right). It would have been nice to see the difference between WT and the DKO.

Line 291: Fig 5F should also cite a supplementary Table

Line 329: Lysosomal digestive function should be lysosomal clearance here, since it is not a DQ-BSA assay that directly reads out on digestion.

Line 352 - The connection between extracellular vesicles and lysosome hyper-exocytosis isn't immediately clear to the reader, who may read extracellular vesicles and think about ciliary ectosomes rather than ILV release with lysosome exocytosis. Suggest "To test this hyper-exocytosis hypothesis, we first examined the secretion of intraluminal vesicles to the extracellular space" to match with Fig 7A ILV notation.

Discussion -

Line 394: Suggest edits to this sentence to make the directionality of impact immediately clear to the reader: Disrupting ether lipids improves lysosomal exocytosis by increasing number or efficiency of lysosome-plasma membrane fusion events, thereby facilitating the clearance of" To put it another way, ether lipids inhibit lysosome-plasma membrane fusion.

I would like the authors to add to their discussion the comments about whether increased clearance is due to both increased lysosomal exocytosis and increased lysosomal digestive capacity. The increased digestion is counterintuitive given how much the authors point out that M6P targeting does not increase, so some of their comments about SILAC showing that 14 enzymes are unchanged in the discussion could be clarifying to the reader. What is degrading DQ-BSA in LYSET/FAR1 KO cells?

I am surprised the authors do not bring up the HDG experiment in the discussion. It is interesting and relevant that excess ether lipids can impair lysosome exocytosis in WT cells. - Fig 6D - F. Could diet increase ether lipid exposure?

Response to the reviewer's comments:

We sincerely thank the reviewers for their positive evaluation of our revised manuscript and appreciate their helpful and constructive comments. We have revised the manuscript accordingly, as detailed below.

Referee #1

The study by Liang Cheng reveals a novel connection between peroxisomes and lysosomes mediated by ether-bound glycerolipids. The manuscript's findings are novel and represent a significant advance in the field.

The revised version is much more balanced between the lysosomal and peroxisomal sections, and has clearly benefited from the revision process.

The introduction and discussion sections have been broadened in line with the reviewers' suggestions, and new experiments have been added to support the overall conclusions, which are now more justified.

Based on my previous comments, there are still some minor points to be addressed in this revised form:

1. Fluorescence microscopy pictures should be accompanied by a detailed analysis (as shown in Fig. 3B for 3A) for all pictures. Similarly, all western blots need densitometric evaluation.

Thank you for the suggestion. We have now included additional quantitative analyses of fluorescence images, including Fig. 1A, Fig. 1J, and Fig. 6H. Only Fig. EV1A and Fig. 2C lack quantification.

In Fig. EV1A, TFEB-GFP remains cytosolic in LYSET KO cells, which is clearly evident. This conclusion is further supported by RNA-seq data in Fig. EV2B, showing that transcription of CLEAR network genes is unchanged.

Similarly, Fig. 2C demonstrates increased lysosome abundance in LYSET KO cells, which is restored to near WT levels after the third round of cell sorting. This finding is independently supported by LysoTracker Red staining and flow cytometry data shown in Fig. 2B.

Regarding Western blots, densitometric quantification has been performed and is now included for Figs. EV1D, EV3H, EV5J, EV5K, 6I, and 6J. We did not include quantification for Fig. 1F, as the knockout cell lines were verified by sequencing and the protein differences are unambiguous. Fig. 5G was not quantified because the conclusions are independently supported by the quantitative SILAC data in Figs. 5E–F and EV Dataset 2. Lastly, Figures 7B and 7I do not include quantification; although these experiments were independently repeated at least three times with consistent trends, statistical significance was not achieved due to inherent biological variability. We are happy to provide the original raw data upon request.

2. Ref. 51 (Chu, Cell, 2015) is now cited but only in the results section. Given that this publication clearly demonstrates a lipid link between the two organelles in question, I believe it should be referenced in the introduction or discussion sections.

Thank you for the suggestion. We have included additional details in the Results section (lines 324–330). The apparent inconsistency is likely due to differences in the peroxisomal genes studied. Chu et al. did not identify ether lipid synthesis genes in their screen for cholesterol accumulation mutants but instead focused on ABCD1.

In contrast, our screen was designed to identify mutants that alleviate lysosomal accumulation, and accordingly, we identified components of the ether lipid synthesis pathway rather than ABCD1. We have added the following sentence: "This suggests that mutations in different peroxisomal genes may have opposing effects on lysosomal cholesterol levels."

3. PEDS1 is the official gene symbol for the PEDS1 protein. (<https://www.ncbi.nlm.nih.gov/gene/387521>). TMEM189 can be mentioned as the previous gene symbol but the authors should then stick to the newest nomenclature.

We have updated TMEM189 to PEDS1 throughout the text and figures. Thank you!

4. RCDP consists of four, rather than five, subtypes. FAR1 leads to PFCRD (see the OMIM entry # 616154).

We thank the reviewer for pointing this out. In the OMIM entry referenced, the "Nomenclature" section states: "Baroy et al. (2015) considered the disorder described by Buchert et al. (2014) to be a form of rhizomelic chondrodysplasia punctata, which they termed type 4 (RCDP4)."

We acknowledge that there is ongoing debate regarding the classification of RCDP4, which is caused by FAR1 mutations. A key distinction lies in the skeletal phenotype: RCDP4 lacks the characteristic long-bone abnormalities observed in other RCDP subtypes (Buchert et al, 2014). This difference may reflect the presence of FAR2 in the human genome.

Several experts in peroxisomal disorders and pediatric genetics, including Nancy E. Braverman (McGill University), Michael B. Bober (Nemours/Alfred I. duPont Hospital for Children), and Eirik Frengen (Oslo University Hospital), support classifying FAR1-related disease as RCDP4 (Baroy et al, 2015; Duker et al, 2017).

Given this precedent and our observation of consistent phenotypes between FAR1 KO and AGPS KO cell lines, we have maintained the use of 'RCDP4' to describe this clinical entity.

5. The Methods section (lines 933-941) still reads like a laboratory protocol.

We have revised the corresponding section on Silver Staining. Thank you!

6. Fig. 2F: correct to "phospholipids".

We have corrected this typo. Thank you!

Referee #3

Chen et al set out to identify genes that reduce lysosome accumulation in Mucopolysaccharidosis V cell lines. They find that the loss of ether lipids (and therefore loss of peroxisome function) improves lysosome clearance, which they then show is through a combined mechanism of lysosomal exocytosis and improved digestion. The work is nicely done, and the implications for lysosomal storage disease and WT cells are interesting and clear. I expect future mechanistic work will reveal interesting new functions of ether lipids in lysosome/plasma membrane fusion. I find the study to be compelling, and impactful for scientists interested in lysosomes, peroxisomes, lipids, and unconventional secretion.

The authors have adequately addressed the reviewer concerns, and significantly improved the

manuscript. I recommend it for publication, assuming the authors can address the one major concern that the mass spec data is inadequately documented. I have included minor suggestions that I would leave to the authors' discretion.

Major concern:

SILAC/MS data must be better documented in methods, excel spreadsheets, and raw data uploaded to a repository. It would have been nice to see the difference between WT and the DKO.

Thank you for the suggestion. We have expanded the description of sample preparation for MS/SILAC in the Methods section. In addition, all MS/SILAC data have been deposited in the ProteomeXchange Consortium under the dataset identifier PXD074652 (doi: 10.6019/PXD074652).

Regarding the comparison between WT and DKO lysosomes, these data are presented in Figure 5C (silver staining) and Figure 5G (Western blot analysis using 13 distinct antibodies). Furthermore, performing an additional SILAC-based mass spectrometry comparison between WT and DKO lysosomes would involve a cost exceeding \$5,000. Given current federal funding constraints in the US, we hope the reviewer understands our decision to prioritize the existing high-resolution comparisons (KO/WT and DKO/KO). We believe the primary aim of identifying genes that reduce accumulation is sufficiently addressed by the current datasets.

In the methods about this: "Gel bands were excised" - It doesn't come across in the figure that you excised specific bands for analysis. It would be informative to label dominant protein changes you identified on the gel in Fig. 5C (ie 30 kDa protein in WT that is not in LYSET KO or DKO? 300 kDa protein that is in KO and DKO but not in WT)

We apologize for the confusion. Figure 5C shows a representative silver-stained gel used to assess overall protein accumulation in lysosomes from three different cell lines; this gel was not used for mass spectrometry analysis. For the SILAC samples submitted for mass spectrometry, entire gel lanes stained with SYPRO Ruby were excised and processed, rather than individual bands. We have added a detailed description to the Methods section (page 35, lines 710–716).

Methods is inadequate: How did the Mass Spec Facility collect data? What instrument, HPLC column, MS parameters? How were peptides/proteins identified? Where is the MS data deposited? The data reporting here is insufficient. The currently available excel data table should be labeled in the excel data sheet with Supp Table Number, and data should include protein data (protein name, accession number and molecular weight, number of assigned spectra and unique sequences used for protein identification, percent sequence coverage, database search score) and peptide data (peptide sequence data with start and stop residue numbers, observed mass, mass error, scores/expect values for assignment, post-translational modifications, probability determination for modification sites).

Thank you for pointing this out. The mass spectrometry analysis was performed by the Taplin Mass Spectrometry Facility at Harvard Medical School. They provided the experimental details, which we have now included in the Methods section.

All mass spectrometry proteomics data, including peak files, raw files, result files, and Excel tables containing protein-level data (protein ID, gene symbol, peptide counts, heavy/light intensities, and ratios) and peptide-level data (protein ID, gene symbol, search ID, peptide sequence, charge, start/end positions, heavy/light intensities, and ratios), as well as label-free and compiled datasets (e.g., reference, gene symbol, annotation, molecular weight), have been deposited to the ProteomeXchange Consortium via the PRIDE partner repository under the dataset identifier PXD074652 (doi: 10.6019/PXD074652). The dataset is currently private and will be made publicly accessible upon publication of the manuscript. For reviewer access, we provide the project accession PXD074652 and token UhFXnjTpUDx5.

Minor suggestions for improvement:

Line 39: "This mechanism" - authors say repeatedly that mechanism is beyond the scope. I would revise this sentence.

Thank you. We have changed the sentence to "Our findings reveal how lipid metabolism modulates lysosomal homeostasis and provide insights into potential new strategies to combat lysosomal and peroxisomal disorders." (Now line 36)

Line 111: "leading to the dephosphorylation..." could be "dephosphorylation and nuclear translocation" for clarity on why localization is relevant in Fig. S1A

We have revised the text as suggested. (Now line 119) Thank you!

Line 119 and Fig 1F - You discuss this cell line as a triple knockout, but the TKO nomenclature isn't explicitly used in the figure: +/- sgRNA for LYSET. It would help to be consistent.

We have stopped using the term "TKO". Instead, we stated, "we generated TFEB/TFE3 double knockout (DKO) cells and subsequently deleted LYSET in this background". (Lines 127-128)

Figure 2F - typo in "phosphoipids"

We have corrected this typo. Thank you!

Figure S3B - What is the evidence that the starred band is non-specific?

As shown in Fig. 4A, Fig. EV5E, and Fig. EV5F, the ~55 kDa band (indicated by an asterisk) persists after FAR1 knockout. We therefore concluded that it represents a non-specific band.

Figure S3D - triangle/arrowheads don't point to bands on immunoblot.

We have corrected this. Thank you!

Line 196 - I don't think it is accurate to say PEX5 knockdown reduced total levels of GNPAT from Fig. S2C. In general the immunoblots show evidence of altered post-translational modifications more so than total levels. Figures S2F-H show the change in levels for FAR1 and GNPAT (but not so much AGPS). It's not absolutely essential, but I would recommend moving S3F-J before S3B-E in the figure and discussion.

Thank you for the suggestion. We have rearranged the figure panels, switching S3F–J with S3B–E. Post-import cleavage of AGPS by TYSND1 has been previously reported (Mizuno *et al*, 2013; Skowrya & Rapoport, 2022). Consistent with reduced import, we observe a decrease in the lower, cleaved band of AGPS-eGFP in the revised Fig. EV3I.

Regarding GNPAT protein levels, we are uncertain about the behavior of endogenous GNPAT due to the lack of antibody; however, our quantification indicates that EGFP-GNPAT levels are reduced, as shown in Fig. EV3H.

Figure S3F - Quantification of colocalization with FAR1/PMP70 could be shown for completeness

Thank you for the suggestion. However, *PEX19* knockout results in an almost complete loss of peroxisomal signal, as indicated by the marked reduction of both FAR1 and PMP70 in Fig. EV3B. Therefore, we are unable to quantify colocalization between FAR1 and PMP70, as both signals are largely absent following *PEX19* deletion.

Fig S4 D-H is brought up after Fig S5.

Thank you for pointing this out. Organizing such a large dataset into a single manuscript has been challenging. We have decided to retain the original figure order, as Fig. EV5 focuses on conservation across cell lines and species. To maintain clarity, we prefer to keep panels addressing this theme together rather than separating Fig. EV5.

Fig S4G - the X axis could be altered to make it easier to see the relevant data.

Thank you for the suggestion. We have adjusted the X axis of Fig. S4G (now Fig. EV4G). It shows a representative flow cytometry profile from five independent replicates, all of which are quantified and summarized in Fig. EV4H. In our opinion, Fig. EV4H is more important for data presentation, as it clearly demonstrates a statistically significant difference among different sample groups.

Line 259-261 - this speculation about pharmacological relevance should be in the discussion, not the results.

Thank you for the suggestion. We have moved these lines to the Discussion section (lines 455–467).

In fig 4 or S5, the authors could have looked at the degradation capacity of WT and FAR1 KO cells. While they showed no change in lysosome levels in FAR1 KO cells compared to WT, they did not directly look for changes in degradation kinetics.

Thank you for the suggestion. We measured LAPT4A protein levels (a lysosomal degradation substrate) in WT and FAR1 KO cells and did not observe obvious accumulation, in contrast to LYSET KO cells (Supporting Figure 1A).

We also assessed DQ-BSA fluorescence in FAR1 KO cells (Supporting Figure 1B-C). The signal was slightly reduced compared with WT, likely due to increased lysosomal exocytosis in FAR1 KO cells, as a fraction of lysosomal DQ-BSA fluorescence may be released during exocytosis.

Figure 5F- I would appreciate some discussion of the proteins that went up between DKO/LYSET KO. (Gray dots on right). It would have been nice to see the difference between WT and the DKO.

Thank you for the suggestion. Approximately 24 genes were upregulated in Fig. 5F; however, GO term and STRING analyses did not reveal enrichment of any specific pathway, and we did not pursue further analysis. All original mass spectrometry datasets (KO/WT and DKO/KO) have been deposited in a public repository and will be made available upon publication.

As noted above, the high cost of SILAC and mass spectrometry analyses prevented further comparison of WT and DKO lysosomes. Given that the primary aim of this study is to identify genes whose knockout reduces lysosomal accumulation under lysosomal storage disease conditions, additional large-scale analyses are beyond the scope of this work.

Line 291: Fig 5F should also cite a supplementary Table

Thank you. We have added Dataset EV2 here.

Line 329: Lysosomal digestive function should be lysosomal clearance here, since it is not a DQ-BSA assay that directly reads out on digestion.

We have revised the sentence to "Moreover, high doses of HDG result in the accumulation of degradation substrates and a slower clearance rate." (Lines 350-351). Thank you!

Line 352 - The connection between extracellular vesicles and lysosome hyper-exocytosis isn't immediately clear to the reader, who may read extracellular vesicles and think about ciliary ectosomes rather than ILV release with lysosome exocytosis. Suggest "To test this hyper-exocytosis hypothesis, we first examined the secretion of intraluminal vesicles to the extracellular space" to match with Fig 7A ILV notation.

We have revised the sentence as suggested (Lines 374-376). Thank you!

Discussion -

Line 394: Suggest edits to this sentence to make the directionality of impact immediately clear to the reader: Disrupting ether lipids improves lysosomal exocytosis by increasing number or efficiency of lysosome-plasma membrane fusion events, thereby facilitating the clearance of" To put it another way, ether lipids inhibit lysosome-plasma membrane fusion.

We have included the suggestion in Lines 419-422. Thank you!

I would like the authors to add to their discussion the comments about whether increased clearance is due to both increased lysosomal exocytosis and increased lysosomal digestive capacity. The increased digestion is counterintuitive given how much the authors point out that M6P targeting does not increase, so some of their comments about SILAC showing that 14 enzymes are unchanged in the discussion could be clarifying to the reader. What is degrading DQ-BSA in LYSET/FAR1 KO cells?

Thank you ! We have added two paragraphs to the Discussion to address these points (lines 441-453).

I am surprised the authors do not bring up the HDG experiment in the discussion. It is interesting and relevant that excess ether lipids can impair lysosome exocytosis in WT cells. - Fig 6D - F. Could diet increase ether lipid exposure?

Thank you for the suggestion. HDG supplementation has been used in the plasmalogen research community to rescue defects in ether lipid synthesis. We agree that it is highly interesting and relevant that excess HDG can inhibit lysosomal function. However, at this stage, we do not have evidence demonstrating that this effect occurs through inhibition of lysosomal exocytosis, although we consider this a plausible hypothesis.

Thank you so much for all your insightful questions and thoughtful suggestions!

References:

Baroy T, Koster J, Stromme P, Ebberink MS, Misceo D, Ferdinandusse S, Holmgren A, Hughes T, Merckoll E, Westvik J *et al* (2015) A novel type of rhizomelic chondrodysplasia punctata, RCDP5, is caused by loss of the PEX5 long isoform. *Hum Mol Genet* 24: 5845-5854

Buchert R, Tawamie H, Smith C, Uebe S, Innes AM, Al Hallak B, Ekici AB, Sticht H, Schwarze B, Lamont RE *et al* (2014) A peroxisomal disorder of severe intellectual disability, epilepsy, and cataracts due to fatty acyl-CoA reductase 1 deficiency. *Am J Hum Genet* 95: 602-610

Duker AL, Niiler T, Eldridge G, Brereton NH, Braverman NE, Bober MB (2017) Growth charts for individuals with rhizomelic chondrodysplasia punctata. *Am J Med Genet A* 173: 108-113

Mizuno Y, Ninomiya Y, Nakachi Y, Iseki M, Iwasa H, Akita M, Tsukui T, Shimosawa N, Ito C, Toshimori K *et al* (2013) Tysnd1 deficiency in mice interferes with the peroxisomal localization of PTS2 enzymes, causing lipid metabolic abnormalities and male infertility. *PLoS Genet* 9: e1003286

Skowrya ML, Rapoport TA (2022) PEX5 translocation into and out of peroxisomes drives matrix protein import. *Mol Cell* 82: 3209-3225 e3207

Dear Ming,

I am pleased to inform you that your manuscript has been accepted for publication in the EMBO Journal.

Congratulations to you and all involved!

You may qualify for financial assistance for your publication charges - either via a Springer Nature fully open access agreement or an EMBO initiative. Check your eligibility: <https://link.springer.com/journal/44318/how-to-publish-with-us>

Best wishes,

William

William Teale, PhD
Editor
The EMBO Journal
w.teale@embojournal.org

Please note that it is The EMBO Journal policy for the transcript of the editorial process (containing referee reports and your response letters) to be published as an online supplement to each paper. If you should prefer removal of any referee-only figures included in the point-by-point response(s), e.g. because they may still be used for future publication or because they have been reproduced from published work by others, please do let us know immediately via response email.

More information is available here: <https://link.springer.com/partners/embo-press/editorial-policies#Peer%20review>
